

# Consistent biases in Antarctic sea ice concentration simulated by climate models

Lettie A. Roach[1,2], Samuel M. Dean[1], and James A. Renwick[2]

[1]National Institute of Water and Atmospheric Research, 301 Evans Bay Parade, Greta Point, Wellington 6021, New Zealand
[2]School of Geography, Environment and Earth Sciences, Victoria University of Wellington, Wellington 6012, New Zealand

*Correspondence to:* Lettie Roach (lettie.roach@niwa.co.nz)

**Abstract.** The simulation of Antarctic sea ice in global climate models often does not agree with observations. In this study, we examine the compactness of sea ice, as well as the regional distribution of sea ice concentration, in climate models from the latest Coupled Model Intercomparison Project (CMIP5) and in satellite observations. We find substantial differences in concentration values between different sets of satellite observations, requiring careful treatment when comparing to models. As a fraction of total sea ice extent, models simulate too much loose, low-concentration sea ice cover throughout the year, and too little compact, high-concentration cover in the summer. In spite of the differences in physics between models, these tendencies are broadly consistent across the population of 27 CMIP5 simulations, a result not previously highlighted. Targeted model experiments with a coupled ocean — sea ice model show that over-estimation of low-concentration cover is partially determined by choice of constant floe diameter in the lateral melt scheme. This suggests that current sea ice thermodynamics contribute to the inadequate simulation of the low-concentration regime.

## 1 Introduction

The cycle of sea ice growth and melt in the Southern Ocean is one of the largest seasonal signals on Earth. The heterogeneity of the sea ice cover and distribution of open water areas determine regional albedo, the reflectivity of the Earth's surface. This in turn impacts entrainment of irradiative energy into the ocean mixed layer (Asplin et al., 2014) and the atmospheric energy budget (Previdi et al., 2015). Sea ice production, which increases salinity, in areas of open water strongly impacts the rate of Antarctic Bottom Water formation (Goosse et al., 1997), the deepest water mass. Regional sea ice concentration thus plays an important role in the coupled climate system.

Coupled climate model output collated by the World Climate Research Programme (WCRP) under the Coupled Model Intercomparison Project (CMIP) protocol are a valuable resource for understanding Earth's climate system. Over 20 groups worldwide have contributed simulations to the latest project (CMIP5) from their models, many of which are developed independently and include different physics. The sea ice components of these models range in complexity, from single-layer,



ocean-advected, limited-rheology models (e.g. HadCM3; Gordon et al., 2000) to multi-layer, multiple thickness category models with a non-linear viscous plastic rheology and explicit melt pond formation (e.g. NorESM; Bentsen et al., 2013; Hunke et al., 2015). Advances in Earth system modelling have improved simulation of Arctic sea ice compared to the previous intercomparison project (CMIP3) (Stroeve et al., 2012), but have not improved overall in the Antarctic (Mahlstein et al., 2013).

To make assessments like these, most model evaluation studies quantify agreement between sea ice models and observations using sea ice extent, which is simply the area of all grid cells with more than 15 % sea ice concentration. Turner et al. (2013) find a wide range of seasonal cycles and trends in Antarctic sea ice extent across the CMIP5 ensemble. Compared to observations, they find that a majority of models underestimate the minimum sea ice extent in February. Shu et al. (2015) evaluates simulated sea ice volume and thickness as well as sea ice extent, finding that the CMIP5 multi-model ensemble mean sea ice extent is

fairly well simulated, though worse in the Antarctic than in the Arctic, but suggest that the sea ice cover is generally too thin. Zunz et al. (2013) find that all models overestimate interannual variability of Antarctic sea ice extent, particularly in winter. They conclude that no CMIP5 model produces Antarctic sea ice in reasonable agreement with observations over the satellite era.

    Using only sea ice extent means that these model evaluation studies do not take into account any sub-grid scale sea ice

information, or the regional distribution of sea ice. As discussed by Notz (2014) and Ivanova et al. (2016), model simulations with the same sea ice extent could have very different sea ice cover characteristics. Notz (2014) instead examines the frequency distribution of summer Arctic sea ice concentration, finding that around half the CMIP5 models have a 'compact' ice cover ($> 0.4$ of grid cells with more than 90 % sea ice concentration) and the rest have a 'loose' ice cover. Ivanova et al. (2016) present a similar analysis for the Antarctic, but show only the CMIP5 multi-model mean and do not discuss the results in

detail, focusing instead on the alternative metrics they developed.

    In this study we examine model agreement with observations using various simple metrics that account for sea ice concentration values and the regional distribution of sea ice. Our aim is to identify biases in Antarctic sea ice that are common across multiple models. We then carry out targeted model experiments to investigate the role of sea ice model thermodynamics in these biases.

## 2  Methods

### 2.1  CMIP5 Models

A series of experiments from different global climate models were carried out for the Coupled Model Intercomparison Project, Phase 5 (CMIP5), (Taylor et al., 2012). Output is freely available online from the Program for Climate Model Diagnosis and Intercomparison. The historical experiments, which are forced by observed natural and anthropogenic forcings, end in 2005.

To obtain a more contemporary overview, we neglect the years before 2000 and also consider the first nine years of projection experiments from the midrange mitigation emission scenario (RCP4.5), giving fifteen years i.e. 2000-2014. We select the first ensemble member for all models that provide daily sea ice concentration for both the historical and RCP4.5 experiments, resulting in a set of 27 models (see Table 1).





## 2.2 Observations

To account for the large uncertainty in satellite observations (Bunzel et al., 2016), we use three observational data sets which apply different algorithms to convert passive-microwave signals into sea ice concentration. These are the Bootstrap algorithm (Comiso, 1986), the NASA Team algorithm (Cavalieri et al., 1984) and the ASI algorithm (Kaleschke et al., 2001; Spreen et al.,

2008). We do not consider datasets that merge different observation methodologies. Differences between the three selected data sets are large: in the Antarctic, the NASA Team algorithm shows the marginal ice zone (defined as the extent of sea ice with concentration between 15 % and 80 %) to extend over 2 million km$^2$ more than the Bootstrap algorithm (Stroeve et al., 2016). Concentrations from the NASA Team algorithm have been found to be low overall compared to independent observations (Agnew and Howell, 2003; Partington et al., 2003). Steffen and Schweiger (1991) find that this varies across concentration

regimes, with the NASA Team algorithm underestimating concentrations in compact ice cover and overestimating concentrations in areas of loose ice. The NASA Team algorithm is also biased low overall compared to the Bootstrap algorithm ice cover in the Arctic (Notz, 2014), with the difference attributed to their different methods for accounting for melt ponds. Observational uncertainty is higher in summer than the rest of the year, as wet snow and thin ice as well as melt ponds complicate satellite retrievals (Ivanova et al., 2014).

Uncertainty estimates are not released for these data products and we do not seek to evaluate the various observations here. We therefore take a simple objective approach and consider the true sea ice concentration to be the mean of these three data sets. We adopt the range of values from the three sets as a representation of observational uncertainty, following Notz (2015).

## 2.3 Metrics

Following Turner et al. (2013), we interpolate model output and observational data on to a common grid. We choose a $1^o \times 1^o$

regular grid. This resolution is equal to or higher than 14 of the 27 models and lower than all observations. We consider it to be an acceptable midpoint given the large range of model resolutions.

Following convention, sea ice extent is defined as the area of all grid cells with more than 15 % sea ice concentration. Sea ice area is the sum of the area of all grid cells with more than 15 % sea ice concentration multiplied by the sea ice concentration in each grid cell. In this study we consider not only sea ice area, but also develop a new metric based on 'binned sea ice extent,'

which is the area of grid cells whose sea ice concentration falls into one of nine concentration bins, ( (10-20 %], (20-30 %], ... , (90-100 %]). Like Notz (2014) and Ivanova et al. (2016) we include concentrations below 15 %, but unlike these studies we do not include concentrations below 10 %.

To avoid dependence on the total sea ice extent in each model, which is likely determined by ocean surface temperatures, we divide binned sea ice extent by the total extent of ice with concentrations greater than 10 %, giving a 'fractional binned

sea ice extent.' This metric allows us to examine observed and modelled behaviour in different sea ice concentration regimes. It does not penalise models whose spatial distribution of sea ice disagrees with observations, but it does allow us to quantify disagreement with observations on sea ice concentration values while accounting for the observational range.





To account for misplacement of sea ice, we adapt the integrated ice-edge error (IIEE) from Goessling et al. (2016). The IIEE describes the area of grid cells where observations and a model disagree on the presence of sea ice. We instead define an integrated ice area error (IIAE) that describes the area of sea ice on which models and observations disagree. This is the sum of sea ice area overestimated and underestimated,

$$\text{IIAE} = O + U \qquad (1)$$

with

$$O = \int_A \max{(c_m - c_o, 0)} dA \qquad (2)$$

and

$$U = \int_A \max{(c_o - c_m, 0)} dA \qquad (3)$$

where $A$ is the area of interest, $c_m$ is the simulated sea ice concentration and $c_o$ is the observed sea ice concentration. The latter is represented by the mean of the three satellite observations. Following Goessling et al. (2016), the IIAE can be decomposed into the total sea ice area difference between model and observations (absolute area error, AAE) and the difference in sea ice area due to misplacement of sea ice (misplacement area error, MAE),

$$\text{IIAE} = \text{AAE} + \text{MAE} \qquad (4)$$

where

$$\text{AAE} = |O - U| \qquad (5)$$

and

$$\text{MAE} = 2 \cdot \min(O, U) \qquad (6)$$

A disadvantage of the IIAE is that it does not take into account the observational range, using only the observational mean
as the 'true' state. Nevertheless it is a useful metric because it quantifies error in integrated sea ice concentration values as well as quantifying error caused by sea ice appearing in different grid cells than the observations. This is in contrast to difference





in sea ice area, which accounts only for error in integrated sea ice concentration values, and difference in sea ice extent, which accounts only for error in the area of grid cells that have ice.

To look for behaviours which are consistent across all CMIP5 models, we compare the population of all models for the fifteen years from 2000 to 2014 against the population of all observations for the same period. Including all models means
5 that the range is large when models show opposite tendencies; using a multi-model mean would average out this information. Including all days in each season for all years during analysis captures sub-seasonal and inter-annual variability.

Note that we use a 365 day year for all years in the analysis. Any data for February 29th is disregarded. Also note that by 'observational mean' for any metric, we refer to each metric calculated from the mean of daily sea ice concentrations across the three observational data sets.

## 2.4  Coupled ocean-sea ice model

To understand the impact of model parametrizations for sea ice thermodynamics, we carry out perturbed parameter simulations using a coupled ocean — sea ice model. This consists of the ocean model NEMO and the sea ice model CICE5.1 forced with the atmospheric reanalysis JRA-55 (Japan Meteorological Agency, 2013), run on a $1^o$ tripolar grid. CICE is a state-of-the-art sea ice model and is used as the sea ice component for several of the CMIP5 models (Table 1). Below we briefly explain the
15 model's sea ice thermodynamics; further details may be found in Hunke et al. (2015).

CICE describes the evolution of the ice thickness distribution in five discrete categories. A volume of new sea ice growth is calculated from the ocean freezing/melting potential $F_{\mathrm{frz/mlt}}$, with new ice added as area in the smallest thickness category until the open water fraction is closed, after which it grows existing ice thickness. For sea ice melt, the net downward heat flux from the ice into the ocean, $F_{bot}$ is:

$$F_{bot} = -\rho_w c_w c_h u_* (T_w - T_f) \tag{7}$$

where $\rho_w$ and $c_w$ are the density and heat capacity of sea water, $c_h = 0.006$ is the heat transfer coefficient, $u_* = \sqrt{|\boldsymbol{\tau_w}|\rho_w}$ is the friction velocity, $T_w$ is the sea surface temperature and $T_f$ is the ocean freezing temperature, following Maykut and McPhee (1995). The balance of this flux with a conductive flux through the ice determines basal melt.

A fraction of ice is also melted laterally following Steele (1992). If floes have a mean caliper diameter $L$, their perimeter
25 is $p = \pi L$ and their horizontal surface area is $s = \alpha L^2$ (where $\alpha \approx 0.66$ accounts for the non-circularity of floes and was determined empirically by Rothrock and Thorndike 1984). It is assumed that melting occurs uniformly at a rate $w_{lat}$ around the perimeter of each floe, i.e.

$$\frac{\mathrm{d}s}{\mathrm{d}t} = w_{lat} p$$

Therefore the change in diameter is





$$\frac{\mathrm{d}L}{\mathrm{d}t} = \frac{\pi}{2\alpha} w_{lat}$$

For a region containing $n$ floes with only a single diameter $L$, with a total horizontal area $s_{tot}$, the total concentration $A$ is

$$A = \frac{n}{s_{tot}} s(L) = \frac{n}{s_{tot}} \alpha L^2$$

Hence, with $s_{tot}$ and $n$ constant in time and letting the subscript $_o$ denote the initial state,

$$A = A_o \left( \frac{L}{L_o} \right)^2 \tag{8}$$

Differentiating this and inserting $dL/dt$ then gives the change in concentration

$$\frac{\mathrm{d}A}{\mathrm{d}t} = \frac{A\pi}{L\alpha} w_{lat} \tag{9}$$

CICE uses a uniform lateral melt rate of

$$w_{lat} = m_1 (T_w - T - f)^{m_2} \tag{10}$$

which was based on Josberger and Martin (1981), who found a complex boundary layer adjacent to vertical ice walls melting in saltwater in the laboratory, with convective motions following different flow regimes. The region adjacent to the turbulent flow regime showed the largest lateral melt rate, which could be fitted to the above relation. The coefficients $m_1$ and $m_2$ are the best fit to data quoted by Maykut and Perovich (1987), measured in a single static lead in the Canadian Arctic archipelago over a three week period. In order to apply Eq. 9, CICE assumes a single floe diameter of $L = 300$ m throughout the ice pack. This is one of the more sophisticated schemes for lateral melt in the CMIP5 models; often it is not included at all (Table 1).

The experiments described below, which are performed with the coupled NEMO-CICE model, begin in 1979 and end in 2014. The years before 2000 are neglected to allow for model spin-up. Model output from these experiments is analysed on its native grid ($1^o$ tripolar).

## 3 Results

Despite the seemingly diverse sea ice simulations (Fig. 1), the CMIP5 models do show some similar tendencies. While sea ice area at the annual maximum has a large inter-model and inter-annual spread with no clear bias compared to observations, sea ice area at the annual minimum is consistently biased low (Fig. 2). This tendency was also noted by Turner et al. (2013) for sea




ice extent. Integrated ice area error as a fraction of the observational mean sea ice area also suggests that, in general, models perform more poorly at the summer minimum than the winter maximum (Fig. 3(b)).

At the winter maximum, across the population of CMIP5 models and different years, we find that the absolute area error and the misplacement area error contribute equally to the total integrated ice area error. At the summer minimum, the total

5 integrated ice area errors for the CMIP5 models are dominated by absolute area errors (Fig. 3(c)). Such 'total' errors in the integrative measure of hemispheric sea ice area are likely caused by a temperature-biased ocean and/or atmosphere. To remove these total biases and isolate behaviour that may be induced by the sea ice component, we now consider a fractional binned sea ice extent.

Fractional binned sea ice extent is a similar (but not equivalent) metric to the frequency distribution of concentration that

Notz (2014) use to evaluate satellite observations in the Arctic. It can be used to describe observed and simulated behaviour in different concentration regimes. We first describe satellite observations using fractional binned sea ice extent.

In the Antarctic, the ASI observations show similar characteristics to the Bootstrap observations, while the NASA Team observations differ from both. This results in a skewed distribution when considering the observational range from the three data sets relative to the observational mean. We find that the NASA Team algorithm shows a looser ice cover, with a significantly

lower proportion of cover in the 90 %+ concentration bin, than both the Bootstrap and ASI observations (Fig. 4). This result holds when considering (non-fractional) binned sea ice extent as well (not shown). We also find that this difference between data sets persists throughout the year, unlike in the Arctic where frequency of compact sea ice cover shown in the Bootstrap and NASA Team datasets agree better in winter (Notz, 2014). This suggests that it is not just treatment of melt ponds, as suggested by Notz (2014) and which are less important in the Antarctic than Arctic, or of ice types associated with melting that differs

between the two algorithms. Note that the large range in Fig. 4 reflects both inter-annual and sub-seasonal variability in the observations, as well as uncertainty arising from different processing of satellite data.

We find that the lower to upper quartile ranges for fractional binned sea ice extent from the population of all observations and the population of all models, including inter-annual and sub-seasonal information, broadly agree (Fig. 5(a-d)). While most differences between models and observations are significant at the 95 % confidence level (as expected given the population

size), the two populations' lower to upper quartile ranges are not distinct for most concentration bins. This indicates observational limitations as well as suggesting that the sea ice components of CMIP5 models are somewhat successful at simulating the distribution of sea ice concentration.

However, there are some significant differences from observations. To better highlight these, we show the percentage difference from the observational mean in each bin (Fig. 5(e-h)). The populations are the percentage differences from the obser-

30 vational mean for each day and year, and the observational distribution is skewed, so the population from observations is not centred on zero. Fig. 5(e-h) reveals large discrepancies between models and observations in the highest and especially the lowest concentration bins. During summer (DJF), the lower to upper quartile range for 90 %+ sea ice concentration from models and observations do not overlap at all. Models strongly underestimate the fraction of sea ice area with concentration greater than 90 %, that is, their central ice pack is not compact enough. They tend to overestimate the fraction in the 70-90 % bins

to compensate. In all seasons, the models overestimate the fraction of sea ice area in the two lowest concentration bins. The





upper to lower quartiles from observations and models do not overlap at all for the 10-20 % bin in spring, autumn and winter. These results for the highest and lowest bins are robust when considering sea ice concentration bins spaced at 5 % intervals and beginning at 15 %, the cut-off used universally for sea ice extent.

As discussed above, this assessment takes into account observational uncertainty and inter-annual and sub-seasonal variability. That such distinct characteristics arise from a population of 27 models, which contain diverse physics and different sea ice, ocean and atmosphere models, is striking. It suggests that there is some deficiency or missing physical process common to all models.

A plausible explanation could be that models form sea ice that is too thin in the highest bin, which therefore melts more easily. Conversely, low-concentration sea ice may be too thick. However, we found no relation between these concentration biases and average sea ice thickness for the lowest and highest concentration bins (not shown). We therefore turn to lateral, rather than vertical, thermodynamics in the next section.

## 4    Impact of floe size

We hypothesize that the biases in low-concentration Antarctic sea ice are partially determined by lateral floe size. Lateral floe size impacts sea ice concentration through lateral melt only if included at all in the CMIP5 models, (see Table 1). Tsamados et al. (2015) showed that a concentration-dependent lateral melt parametrization significantly impacted the decomposition of sea ice melt processes, resulting in reduced sea ice concentrations around the ice edge in the Arctic. In the Antarctic, heat flux from solar heating of open water areas has been cited as the major cause of sea ice decay (Nihashi and Ohshima, 2001), with this melting potential available for both lateral and bottom melt. Recent studies have also suggested that floe size should also impact sea ice concentration through processes such as floe-floe collisions and lateral growth (Horvat and Tziperman, 2015; Zhang et al., 2015).

As shown in Subsect. 2.4, in CICE the lateral melt flux is independent of floe size, while the change in concentration arising from lateral melt is inversely proportional to a constant floe diameter, $D = 300$ m. In reality, sea ice floes can range in size across orders of magnitude. Several observational studies (e.g. Steer et al. 2008; Paget et al. 2001) find that the number distribution of floe sizes per unit area follows a power law with a negative exponent, suggesting that there can be a large number of small floes.

While concentration is not a proxy for floe size, in general we may expect that low-concentration areas will be made up of smaller sea ice floes than high-concentration areas because they are usually nearer the ice edge. An area of smaller sea ice floes will experience more lateral melt than an area with a larger floe size (Eq. 9). We therefore suggest that CMIP5 models using the Steele (1992) lateral melt parametrization simulate too much low-concentration sea ice because this is made up of floes smaller than 300 m and so should be subject to more lateral melt. At the sea ice edge, lateral melt may occur throughout the year. In areas around the ice edge, which are principally low-concentration, marginal ice zone processes not included in CMIP5 models, such as wave fracture and dynamic floe interactions, may further reduce concentrations. Conversely, in high-concentration areas, floes are likely to be larger than 300 m and therefore should be subject to less lateral melt than the Steele



(1992) parametrization prescribes. This could explain the underestimation of high-concentration sea ice seen in Antarctic summer.

In order to test this hypothesis, we perform three experiments using the coupled ocean — sea ice model described in Subsect. 2.4. The experiments have identical set ups apart from a variation in $L$, the fixed floe diameter. We run experiments using (i) the standard value of $L = 300$ m, (ii) a low value of $L = 1$ m and (iii) a high value of $L = 10,000$ m. Our perturbed parameter values are constant and not realistic, but instead are chosen to investigate and highlight the impact of extreme changes.

Fig. 6 shows the impact of reduced floe size on fractional binned sea ice extent. The fraction of sea ice extent belonging to each concentration bin is significantly different (according to a Student's T-test with $p < 5$ % and with little overlap between the upper and lower quartile ranges) to that from the standard model during DJF (Fig. 6). There is a large reduction in low-concentration (10-20 %) sea ice compared to the standard model, with increases in the 30-70 % concentration bins to compensate. This brings the simulated fractional binned sea ice extent into better agreement with observations. The reduction in low-concentration sea ice is also visible during the fall, MAM (Fig. 7). Note that we have tested only the impact of lateral melt on this bias. Other floe-size dependent processes may reduce the proportion of low-concentration sea ice further. However, we cannot test this without access to sea ice models that include these processes.

The enhanced lateral melt achieved by reducing floe size results in statistically significant reductions in sea ice concentration relative to the standard model, particularly in the summer. As expected, enhanced lateral melt reduces the high bias in concentration near the outer ice edge compared to observations (reduction in blue, Fig. 8(a-b)), but enhances the low bias compared to observations elsewhere (increase in red, Fig. 8(a-b)). We use the integrated ice area error described above to quantify agreement with the mean of the three sets of satellite observations. We find that the difference in overall agreement with observations between the standard model and the small floe simulation is negligible. The absolute area error significantly increases in the small floe simulation, because overall this simulation melts too much ice compared to observations. The misplacement area error, however, is significantly reduced in the small floe simulation. This is partly because there is less ice to be misplaced, but also because increased lateral melt improves the distribution of sea ice around the ice edge, by melting areas where there is too much ice compared to observations (Fig. 8(a-b)).

The impact of increased floe size, on the other hand, is much smaller (Fig. 8(c)). Differences in sea ice concentration between the standard model and the large floe simulation are barely perceptible. Changes in the ice errors relative to the standard model are of the opposite sign compared to the small floe simulation, but these changes are unlikely to be significant. Examining the basal and lateral melt rates, we find that the hemispheric average DJF 2000-2014 mean lateral melt rate accounts for only 5 % of the combined basal and lateral melt rates in the standard model. It accounts for a larger proportion (9 %) of melt in the Arctic. Decreasing floe diameter by two orders of magnitude increases the lateral melt rate to 83 % of the combined basal and lateral melt.This compensation effect of reduced basal melt when lateral melt is increased was also noted by Tsamados et al. (2015) in the Arctic. On the other hand, increasing the floe diameter by two orders of magnitude effectively switches off lateral melt (0.2 % of combined basal and lateral melt). In the latter case, more melting potential is made available for basal melting, which, because Antarctic sea ice is so thin, has the same impact on sea ice concentration as lateral melt. We conclude that there must be alternative reasons for the consistent underestimation of compact summer ice.





Looking at the regional distribution of summer (DJF) seasonal mean sea ice concentration averaged over 2000-2014, high-concentration (90-100 %) ice appears in the observational mean only in the Weddell Sea (Fig. 1). Taking the difference between the observational mean high-concentration ice and sea ice concentration in the CMIP5 model simulations shows that none of the models simulate high enough concentrations in this area (Fig. 9), with the possible exception of ACCESS1-3. They all

underestimate concentrations in the Weddell Sea, the largest region of multi-year ice in Antarctica. The bias is not present in other seasons, suggesting it is related to overestimated melt or break-up processes.

Overestimated melt or break-up could be a result of the sea ice model or a biased warm atmosphere or ocean. While consideration of fractional binned sea ice extent is intended to remove overall biases caused by (for example) a warm ocean, in summer the warm ocean could shift the whole distribution to lower concentrations. Alternatively, or likely in conjunction

with this, regionally important processes may be being misrepresented. Evaluating the ORCA2-LIM coupled ocean-sea ice model, Timmermann et al. (2004) found that overestimation of westerly winds led to an underestimation of sea ice coverage on the eastern side of the Antarctic peninsula, in the Weddell Sea. A similar mechanism may be present in several of the CMIP5 simulations.

## 5  Discussion

In this study, we examine the distribution of sea ice concentration from both models and observations. Firstly, we show that observed sea ice concentration values can differ significantly between three widely-used algorithms for satellite data. This observational uncertainty provides a limit beyond which we cannot further evaluate model agreement with observations. Many sea ice model-observations comparisons use only one satellite dataset assumed to represent the true observed state, an approach which may be sufficient when using sea ice extent, a metric where the various algorithms broadly agree. However, when using

metrics that go beyond sea ice extent, using for example sea ice area or sea ice concentration distributions, model evaluation studies should account for the observational range.

We find that simulation of high-concentration (90 %+) sea ice in models is in better agreement with the NASA Team observations than the observational range including the Bootstrap and ASI observations, in agreement with Ivanova et al. (2016), who only examined the CMIP5 multi-model mean. This could reflect that the NASA Team data set is more frequently

used (e.g. Turner et al. 2013; Shu et al. 2015; Zunz et al. 2013) - although there is some suggestion that it is further from the true sea ice state than the alternatives (Partington et al., 2003). In particular, it may have been used as a target during sea ice model development and tuning, a process which is often poorly documented (Mauritsen et al., 2012; Voosen, 2016). We thus encourage use of multiple sets of satellite observations when tuning sea ice models.

Accounting for the observational range, we find that models overestimate the extent of low-concentration sea ice throughout

the year, while underestimating the extent of high-concentration sea ice in summer. This common behaviour across diverse models with varying physics is a result not previously highlighted and warrants further attention. We note that using the observational range as an uncertainty estimate neglects biases that are common to the three different satellite observations. As mentioned above, satellite observations of sea ice are most uncertain in summer. However, we see the bias in low concentration



ice from CMIP5 models throughout the year, and observed summer high-concentration ice is unlikely to be affected by the melt processes that complicate satellite retrievals. The suggestion that the NASA Team algorithm overestimates low-concentration ice (Steffen and Schweiger, 1991) would further strengthen the contrast between models and observations in this regime.

In Subsect. 2.4 we briefly review typical sea ice model thermodynamics, and in particular the change in concentration induced by lateral melt rate for a region containing floes of a single diameter, which follows Steele (1992). Horvat et al. (2016) finds that development of ocean eddies due to lateral density gradients could induce much larger lateral melt than that suggested from the Steele (1992) geometric model. This would support increasing the lateral melt rate in models, as we have done artificially here through a reduced constant floe size. Heat budget analysis (Nihashi and Ohshima, 2001) and modelling studies (Fichefet and Maqueda, 1997; Ohshima and Nihashi, 2005) suggest that the major cause of Antarctic sea ice decay is atmospheric heat input to open water, which causes bottom and lateral melt. Fichefet and Maqueda (1997) find that sea ice melt by open water plays a larger role in the Antarctic than in the Arctic. We further note that the coefficients in the lateral melt rate used in CICE were measured in the Arctic only (Maykut and Perovich, 1987) and few, if any, observational studies exist on the relative importance of bottom and lateral melt in the Antarctic.

The impacts of enhancing lateral melt via reducing a constant floe size shown here suggest that this process should not be applied in the same way throughout the ice pack. While not all models include such a lateral melt parametrization, the biases at the tails of the concentration distributions from the CMIP5 models point to inclusion of model processes that are not suitable for both high-concentration and low-concentration regimes. A possible conclusion, therefore, is that physics in sea ice models are not heterogeneous enough to represent observed sea ice cover. Including information on the floe size distribution and floe size dependent processes (e.g. Horvat and Tziperman 2015; Zhang et al. 2016; Bennetts et al. 2017) could improve consistency with observations in the metrics presented here.



*Data availability.* Most data from this study are publicly available. See http://cmip-pcmdi.llnl.gov/cmip5/data_portal.html for CMIP5 data, http://icdc.cen.uni-hamburg.de/1/daten/cryosphere/seaiceconcentration-asi-ssmi.html for ASI data and https://nsidc.org/data/docs/noaa/g02202_ice_conc_cdr/ for the Bootstrap and NASA Team data. NEMO-CICE model output is available from the corresponding author upon request.

*Author contributions.* L. Roach and S. Dean designed the analysis and experiments and L. Roach carried them out. L. Roach prepared the
5  manuscript with contributions from all co-authors

*Competing interests.* The authors declare that they have no conflict of interest.

*Acknowledgements.* The authors wish to thank Erik Behrens and Jonny Williams for assistance setting up and running the coupled ocean — sea ice model, as well as Cecilia Bitz for helpful discussions in the early stages of this work. This research was funded via Marsden contract VUW-1408.





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





**Figure 1.** Sea ice concentration shown for the three sets of observations (a-c) and the CMIP5 models (d-ae) for the DJF 2000-2014 mean. Models are shown in order of increasing integrated ice area error relative to the observational mean and this value is shown in brackets in million km$^2$



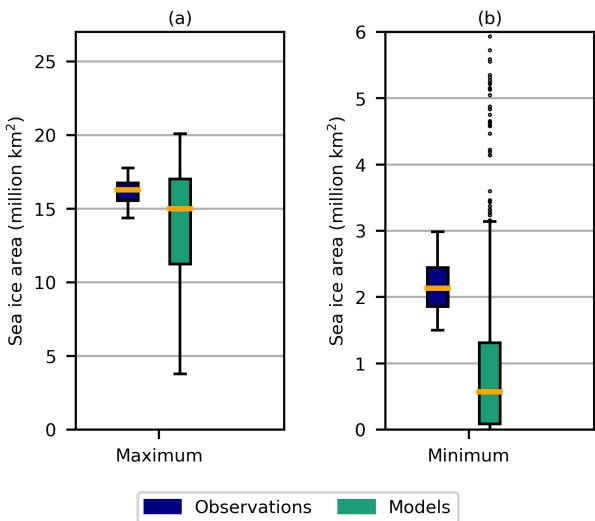

**Figure 2.** (a) The maximum of the sea ice area seasonal cycle and (b) the minimum of the sea ice area seasonal cycle for the population of all observations and the population of models for each year from 2000 to 2014. Boxes extend from the lower to upper quartile values of the data with a line at the median. Whiskers show 1.5 of the interquartile range; beyond this data are considered outliers and plotted as individual points





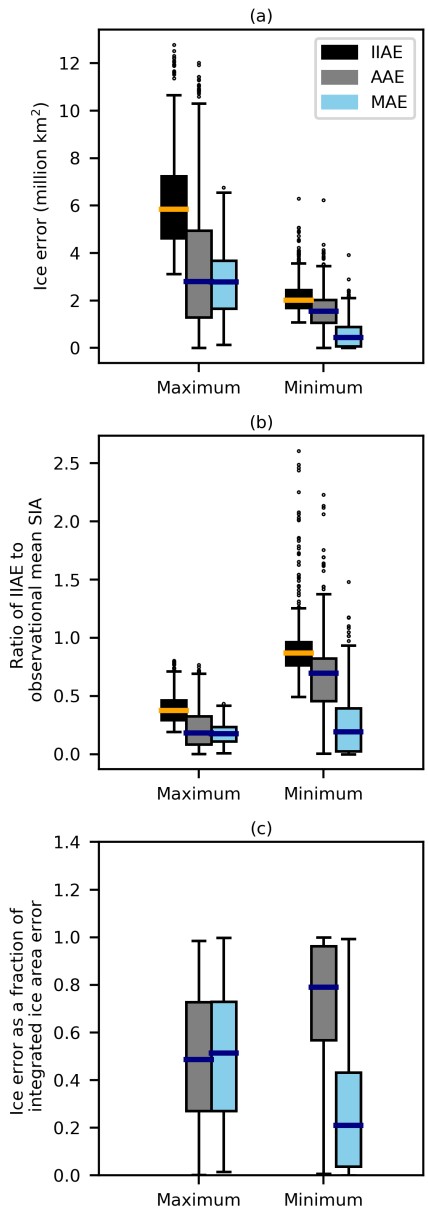

**Figure 3.** (a) The integrated ice area error (IIAE), absolute area error (AAE) and misplacement area error (MAE) relative to the observational mean from the population of CMIP5 models at the day of maximum and minimum sea ice area for each year from 2000 to 2014; (b) as (a) but shown as a fraction of the observational mean sea ice area; (c) as (a) but showing absolute area error and misplacement area error as a fraction of the integrated ice area error for each model and year. Boxplots as in Fig. 2







**Figure 4.** The fraction of sea ice extent in concentration bins for all days in each year from 2000 to 2014 in (a) DJF, (b) MAM, (c) JJA, and (d) SON from the three sets of satellite observations. Boxplots as in Fig. 2



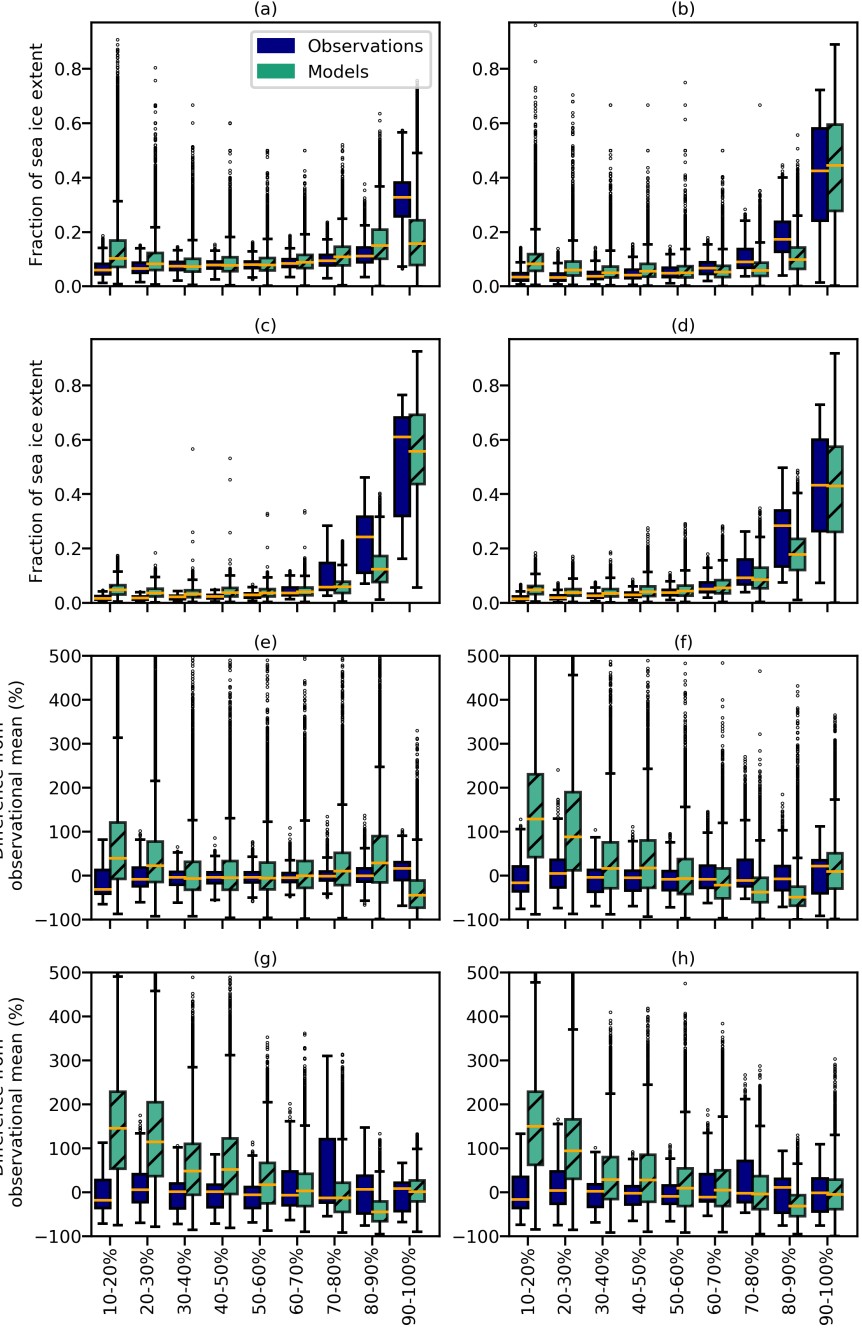

**Figure 5.** (a-d): the fraction of sea ice extent in concentration bins for all days in each year from 2000 to 2014 in (a) DJF, (b) MAM, (c) JJA, and (d) SON for the three sets of satellite observations (*blue*) and the 27 CMIP5 models (*green*). (e-h): as (a-d), but shown as a percentage difference from the observational mean for each day in (e) DJF, (f) MAM, (g) JJA, and (h) SON, for each year from 2000 to 2014. Boxplots as in Fig. 2. Hatching on the model populations denotes that they are statistically different from the corresponding population of observations according to a Student's t-test ($p <$ 5 %)





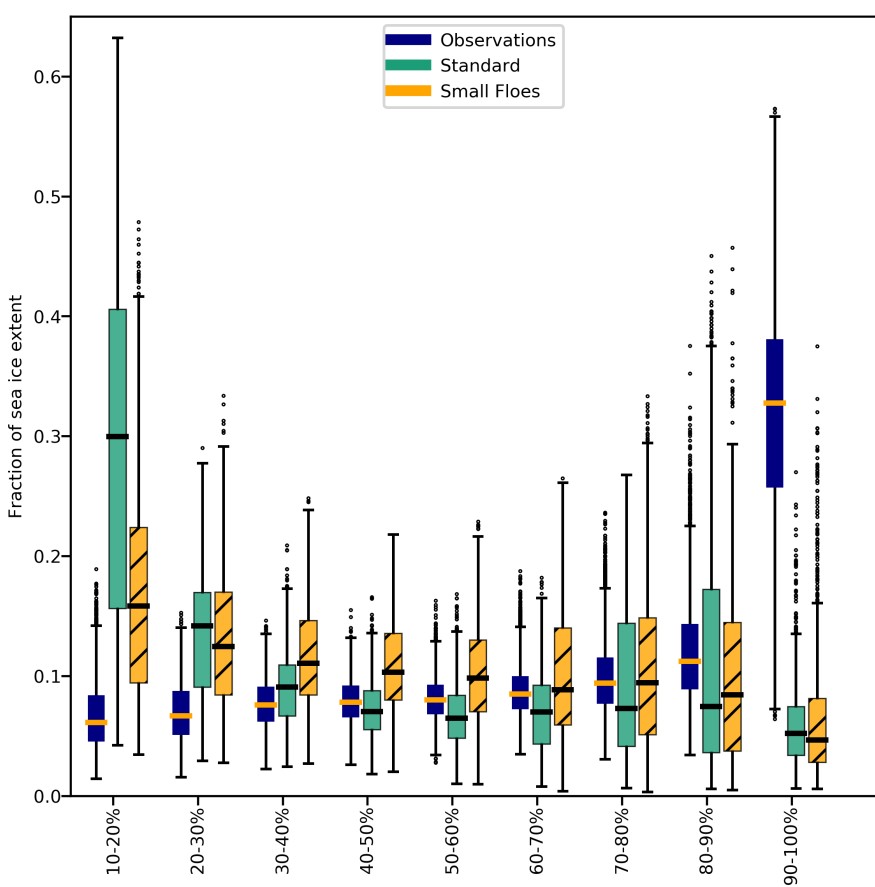

**Figure 6.** The fraction of sea ice extent in concentration bins for all days in each year from 2000 to 2014 in DJF from the three sets of observations, the standard NEMO-CICE ice-ocean model, and the NEMO-CICE ice-ocean model with small floe diameter $L = 1$ m. Boxplots as in Fig. 2. Hatching on the small floe simulation populations denotes that they are statistically different from the corresponding population from the standard model according to a Student's t-test ($p < 5$ %)




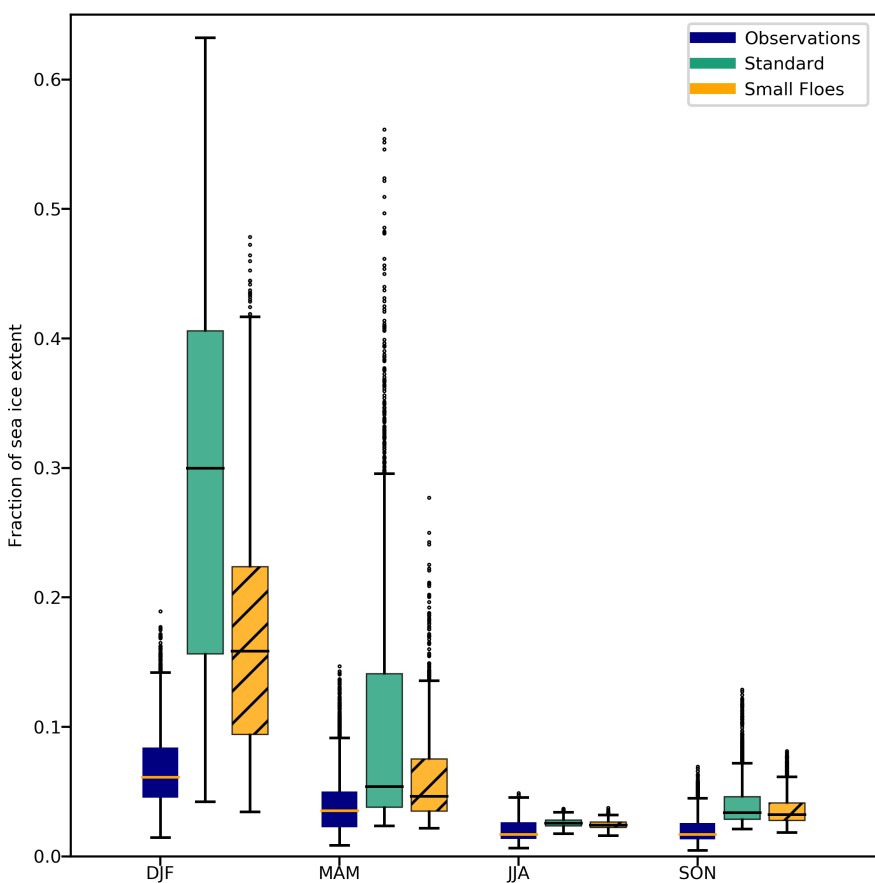

**Figure 7.** The fraction of sea ice extent in the lowest concentration bin ((10-20 %]) for all days in each season for each year from 2000 to 2014 from the three sets of observations, the standard NEMO-CICE ice-ocean model, and the NEMO-CICE ice-ocean model with small floe diameter $L = 1$ m. Boxplots as in Fig. 2. Hatching on the small floe simulation populations denotes that they are statistically different from the corresponding population from the standard model according to a Student's t-test ($p < 5$ %)

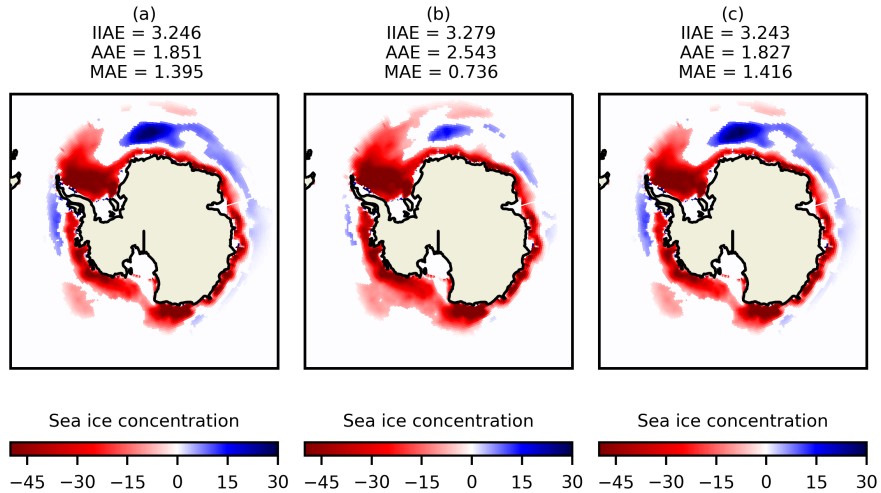

**Figure 8.** Simulation minus observational mean sea ice concentration for DJF 2000-2014 where the simulation has (a) a floe diameter of 300 m (the standard model), (b) a floe diameter of 1 m (small floes), and (c) a floe diameter 10,000 m (large floes). Labels show the integrated ice area error, absolute area error and misplacement area error in million km$^2$. Differences are shown only if they are statistically different according to a Student's t-test over 2000-2014 ($p <$ 5 %)





**Figure 9.** Simulation minus observational mean sea ice concentration for DJF 2000-2014 for each CMIP5 model, where only grid cells with observational mean sea ice concentration is $\geq 90\%$ are considered. Differences are only shown if they are statistically different according to a Student's t-test over 2000-2014 ($p < 5\%$)



**Table 1.** CMIP5 models used in this study

| Short name | Country | Original resolution | Sea ice model | Lateral melt |
|---|---|---|---|---|
| ACCESS1-0 | Australia | $1^o \times 1^o$ tripolar | CICE4.1 | As Subsect. 2.4 |
| ACCESS1-3 | Australia | $1^o \times 1^o$ tripolar | CICE4.1 | As Subsect. 2.4 |
| bcc-csm1-1 | China | $1^o \times (1-\frac{1}{3})^o$ tripolar | SIS | Not included (Li, 2014) |
| bcc-csm1-1-m | China | $1^o \times (1-\frac{1}{3})^o$ tripolar | SIS | Not included (Li, 2014) |
| CanESM2 | Canada | T63 Gaussian * | CanSIM1 | Unknown (reference N/A) |
| CMCC-CM | Italy | ORCA-$2^o$ tripolar | LIM2 | Not included (Rousset et al., 2015) |
| CMCC-CMS | Italy | ORCA-$2^o$ tripolar | LIM2 | Not included (Rousset et al., 2015) |
| CNRM-CM5 | France | ORCA-$1^o$ tripolar | GELATO5 | Thickness-dependent parametrization (Salas Melia, 2002) |
| CSIRO-Mk3-6-0 | Australia | T63 Gaussian * | in-house | Included, but unclear how it impacts SIC** (O'Farrell, 1998) |
| FGOALS-g2 | China | $(1-\frac{1}{2}) \times (1-\frac{1}{2})^o$ tripolar | CSIM5 | As Subsect. 2.4 (Briegleb et al., 2004) |
| GFDL-CM3 | USA | $1^o \times 1^o$ tripolar | SIS | Not included (Winton, 2001) |
| GFDL-ESM2G | USA | $1^o \times 1^o$ tripolar | SIS | Not included (Winton, 2001) |
| GFDL-ESM2M | USA | $1^o \times 1^o$ tripolar | SIS | Not included (Winton, 2001) |
| HADGEM2-CC | UK | $(1-\frac{1}{3})^o \times 1^o$ | CICE-like | Parametrization for SIC <5 % (McLaren et al., 2006) |
| HADGEM2-ES | UK | $(1-\frac{1}{3})^o \times 1^o$ | CICE-like | Parametrization for SIC <5 % (McLaren et al., 2006) |
| inmcm4 | Russia | $1^o \times \frac{1}{2}^o$ | in-house | Empirical parametrization for time evolution of SIC (Yakovlev, 2003) |
| IPSL-CM5A-LR | France | ORCA-$2^o$ tripolar | LIM2 | Not included (Rousset et al., 2015) |
| IPSL-CM5A-MR | France | ORCA-$2^o$ tripolar | LIM2 | Not included (Rousset et al., 2015) |
| IPSL-CM5B-LR | France | ORCA-$2^o$ tripolar | LIM2 | Not included (Rousset et al., 2015) |
| MIROC4h | Japan | $0.28^o \times 0.19^o$ | in-house | Not included (Komuro et al., 2012) |
| MIROC5 | Japan | $1.4^o \times (0.5-1.4)^o$ | in-house | Not included (Komuro et al., 2012) |
| MIROC-ESM-CHEM | Japan | $1.4^o \times 1^o$ | in-house | Not included (Komuro et al., 2012) |
| MIROC-ESM | Japan | $1.4^o \times 1^o$ | in-house | Not included (Komuro et al., 2012) |
| MPI-ESM-LR | Germany | $1.5^o \times 1.5^o$ | in-house | Not included (Notz et al., 2013) |
| MPI-ESM-MR | Germany | $0.4^o \times 0.4^o$ | in-house | Not included (Notz et al., 2013) |
| MRI-CGCM3 | Japan | $1^o \times 0.5^o$ tripolar | in-house | Not included (Tsujino et al., 2010) |
| NorESM1-M | Norway | $1.11^o \times (0.25-0.54)^o$ | CICE4.1 | As Subsect. 2.4 |

* 96 latitudes, 192 longitudes; ** sea ice concentration