# Peer review of "Consistent biases in Antarctic sea ice concentration simulated by climate models"

_The Cryosphere, 2017_

## Referee Comment (RC1) · F. Massonnet (Referee) · 22 Aug 2017

Consistent biases in Antarctic sea ice concentration simulated by climate models

submitted to The Cryosphere by Lettie A. Roach, Samuel M. Dean and James A. Renwick

review by F. Massonnet

This study examines how Antarctic sea ice concentration (SIC) is simulated by modern climate models. The assessment goes beyond traditional metrics of model performance (total area/extent) by looking at how SIC regimes (loose/normal/compact ice) are captured. Multiple observational references are used to account for observational uncertainty in the assessment. The authors find that CMIP5 models overestimate the

fraction of loose ice all year long, while they underestimate the fraction of compact ice in summer. They hypothesize that these systematic biases might be due to the treatment of lateral melting. Running an ocean-sea ice simulation only, they investigate the sensitivity of SIC regimes to the assumed, constant floe size in a state-of-the-art sea ice model.

This work is much welcome and very timely as final developments for CMIP6 are taking place. The simulation of Antarctic sea ice by coupled climate models is arguably poorer than that of Arctic sea ice, hence it is very useful to trace back the possible source of model error. It is always instructive to exhibit systematic biases in large ensembles like CMIP5, because they are indicative of serious concerns in the physics of these models.

The paper essentially consists in two parts, each conveying its own message:

(1) CMIP5 models have consistent biases in the simulation of SIC regimes,

(2) The prescribed ice floe diameter affects the simulated SIC regimes

The hypothesis of the authors is that (1) is largely caused by (2), hence motivating the development of specific lateral melt processes. However, while I appreciate the efforts of the authors to work on parts (1) and (2) individually, my impression is that they are not able to close the loop and directly prove that the deficiency in simulated SIC regimes in CMIP5 is caused by lateral melt parameterizations. In fact, several CMIP5 models do not have such a lateral melting term in their formulation. In addition, there may be many processes, including dynamical ones, that could explain (1). An important paper on the topic (Lecomte et al., 2016; https://doi.org/10.1016/j.ocemod.2016.08.001) should be cited. To test formally whether (2) explains (1), the authors should separate the CMIP5 subset into two groups: those who use a lateral melting term, and those who don't. If the two sub-groups simulate SIC regimes differently in a statistical sense, then this represents a major step forward to confirming the initial hypothesis of the authors. In summary, I'm confident that (2) has a significant effect on the simulation of SIC regimes, but I'm challenging the idea that the CMIP5 ensemble is appropriate to test

this hypothesis, and that (2) is the dominant cause for (1).

I also have several methodological concerns.

First, I appreciate the use of multiple observational datasets in the analyses. This is becoming a standard in the community, and this study is fully in line with that approach. What I find worrying is that the three observations are averaged to make a "super observation". This is a problem when non-linear metrics such as the Integrated Ice Area Error (IIAE) are used, because we loose the sense of uncertainty. If the IIAE was conducted separately on each observational reference first and then reported three times, then the impact of observational uncertainty would be immediately visible. Here, instead, the observational mean is used as a reference giving only one IIAE (e.g., Fig. 3). Information regarding observational uncertainty is lost. For other figures (e.g., Fig. 2) it would be good to display all three observation products individually in order to gauge how much of the variability is due to the product, and how much is due to sub-seasonal and interannual fluctuations.

Second, I sometimes have the impression that the metrics used are overly complicated. The best example is in Fig. 5. There are so many levels of processing that it becomes difficult to understand the meaning of the metrics shown on the figure (see also my comment on Fig. 5 below). For the "binned" diagrams (Figs 4-5-6), why not use only three classes: 0-15%, 15-90%, 90-100%? This would simplify the figures much without removing the useful information that deficiencies in the simulation of SIC happen at the edges of the distribution. In general, I have the impression that I could not replicate the figures if I had the original data.

Third, the authors introduce new metrics without real justification and make inconsistent choices. For example, they adapt the Goessling et al. "Integrated ice edge error" by moving from a "extent-like" definition to an "area-like" definition. This is initially a good idea, since area is a more physical measure than extent. However, the decomposition into an "absolute error" and a "misplacement error" is not as straightforward, as
I'm showing now: consider a model A with uniform SIC of 80% in a number of grid cells, compared to an observational reference with 70% of ice at the same points. Consider also model B, which has half as much ice as A (40% in the grid cells). Even though ice has not been misplaced, the misplacement error term will increase when going from A to B! (While it won't if the Goessling definition was followed). One of the ideas of the Goessling's approach is precisely to use a threshold at 15% to be able to separate total and misplaced errors. The area-like version of that metric looses that property. I also don't understand why the authors compute per-bin sea ice extent, and not sea ice area in Figs. 4-5-6. Sticking to ice area would be a more natural choice given the adaptation made earlier of the IIAE.

Finally, I don't fully understand why the authors used daily data in their analyses. This restricts the number of CMIP5 models available, and adds considerable variability to the metrics. By design, CMIP5 models are not supposed to capture synoptic variability in sea ice extent. Using monthly output would partially average out this variability, and would better allow to exhibit the significant biases of the CMIP5 models as the signal-to-noise ratio would increase. As I'm writing below in a comment, error bars are so large that it is easy to play the devil's advocate and claim that observational references and CMIP5 models are, in the end, not so inconsistent.

Other issues and comments.

p. 2, l. 3: The statement that CMIP5 has improved in the simulation of Arctic sea ice compared to CMIP3 is strong, and perhaps too strong. Rosenblum and Eisenman (2016, http://dx.doi.org/10.1175/JCLI-D-16-0391.s1), for example, suggest that this could be due to the omission of volcanic forcings in several CMIP3 models. Also, it is unclear if the CMIP5 to CMIP3 differences actually reflect improvements, or changes in tuning strategy (e.g. Notz, 2015, http://dx.doi.org/10.1098/rsta.2014.0164). Please nuance this statement.

p. 2, l. 8: evaluates –> evaluate.

p. 2, l. 28: ... Phase 5 (CMIP5; Taylor et al., 2012).

p. 2, l. 30: Years prior to 2000 are neglected in the model evaluation. Have the authors still conducted the evaluation on 1979-2000, for which the CMIP5 output and the observations are readily available? Do conclusions of the study hold? Please elaborate. It would be useful to present the results of such analyses in the supplementary material, to test the stability of the metric developed over time.

p. 2., l. 31: Could the authors conduct their diagnostics on two members of the same climate model, in order to gauge how much internal variability affects the metrics developed in this study?

p. 3., l. 3: Three observational products were used in the study. Since part of the study describes the differences between sea ice concentration in those products, it would be useful to have a few lines describing differences in algorithms between those products.

p. 3., l. 5-7: To what season does the statement on marginal ice zone area difference apply? I think it's winter, please specify.

p. 3., l. 22: "Sea ice area is the sum of the area of all grid cells with more than 15 % sea ice concentration multiplied by the sea ice concentration in each grid cell". Following the conventional NSIDC definition I would have thought that area is just the product of ice concentration by grid cell area, summed over the domain (https://nsidc.org/cryosphere/seaice/data/terminology.html). Why only considering the grid cells with > 15% of ice?

p. 3., l. 25: Why is the bin (0-10%] not in the list?

p. 4., l. 20: "A disadvantage of the IIAE is that it does not take into account the observational range, using only the observational mean as the 'true' state". That's not really a disadvantage of the metric, but rather a methodological issue. The authors could repeat the IIAE taking successively the three observations as references. They would obtain three IIAE's, which would give a sense on the uncertainty associated to

the products. Why didn't the authors go this way? See also my first comment on methodology.

p. 6, l. 17. "[Ocean-sea ice] Model output is analysed on its native grid". Does that mean that the observations were then interpolated onto the NEMO-CICE grid? At page 9, line 18 and in Fig. 8 the NEMO-CICE model is evaluated using the IIAE metric, this means that at some point an interpolation must take place, correct? The model output and the observational reference need to be on the same grid for Eqs (2) and (3) to be evaluated. Why didn't the authors interpolate the NEMO-CICE output on the same target grid as all CMIP5 models, to ensure consistent analysis?

p. 6, l. 20 and Fig. 2. "Sea ice area at the annual minimum is consistently biased low". Here I'm playing the devil's advocate. The blue boxes in Fig. 2 displaythe distri-bution of the three observational references, which are three times the same climate realization plus noise due to the retrieval algorithm. Hence these blue boxes embody time-variability and product variability. By contrast, the green boxes in Fig. 2 contain time-variability internal variabiltiy, and model error. So, the whole question is whether these observational references are incompatible in a statistical sense with the models. Put differently, could the observations be the (N + 1)th CMIP model? Judging from Fig. 2b, the observations lie in the range [1.5 * IQ_75%, IQ_75%] and they could be one of the CMIP models. Or couldn't they? A more quantitative test would be welcome.

p. 7, l. 5-8. The sentence "Such total errors..." is unclear. First, there are many more possible reasons than just ocean/atmosphere temperature biases to explain dif-ferences in total area (which is captured by the AAE). Second, it's not clear why looking at the bias per concentration bin would help isolate the role of the sea ice component.

p. 7, l. 10. Notz (2014) "uses" or "used" but not "use".

p. 7, l. 22-35. The two paragraphs deliver somewhat contradictory messages. The first one finishes by "that the sea ice components of CMIP5 models are somewhat successful at simulating the distribution of sea ice concentration" while the second

paragraph says "large discrepancies between models and observations in the highest and especially the lowest concentration bins". This would need better rephrasing, saying for example in the first paragraph that the "big pictures" are consistent but that this is mostly thanks to cancellation of errors, as explained in the second paragraph.

p. 10, l. 4-6 and Fig. 9. The authors conclude that the systematic underestimation of highly concentrated ice in the Weddell Sea is related to melt or break-up processes. Why is the possibility of a systematic misrepresentation of dynamics ruled out? It could be that all models have deficiencies in capturing the Weddell gyre dynamics. It could be that models are neutral to divergent while observations are in convergent motion. I haven't tested this hypothesis myself, but I don't have enough information from the results of the paper to rule out properly this alternative hypothesis. An exploration of how the models simulate Antarctic ice drift could be helpful in that respect.

p. 10, l. 29: 2. Three observational products are considered in this study. While I appreciate this effort, it looks sometimes like the authors assume that observational errors are random and that the mean of all three products is representative of the truth. It could be that the three observational products have a systematic bias with respect to the truth, which could explain model-obs mismatch on top of model error. The authors don't seem to explore this possibility in the assessment. For example it is known that most algorithms underestimate sea ice concentration as ice becomes very thin. Could this explain the model-obs differences, in particular differences in binned sea ice extent? It is also known that wet snow has a brightness temperature that makes sea ice concentration retrievals higher than they should be. Could this have an impact? More discussion on observational systematic errors would be welcome, in order to place the CMIP results in perspective.

Fig. 1. Interestingly, it is possible that two CMIP5 models with similar IIAEs (e.g., MIROC5 (4.6 Mkm2) and FGOALS-g2 (4.85 Mkm2)) have drastically different sea ice concentration patterns (one with not enough ice and one with way too much ice; panels ac and ad). In the same vein, two models with similar patterns (e.g. CMCC-CMS and

HadGEM2-CC) may have very different IIAEs. This is because of the definition of IIAE which penalizes over- and underestimation in the same way. The authors should comment on that aspect (which I see as a weakness of that metric). Although there is no definition of what a "good" metric is, we could expect that it satisfies properties of continuity in some sense: two models close to each other should have similar metric values.

Fig. 5. This is one example where I would have difficulties in reproducing the result. If I follow correctly, from Fig. 5 caption and from the text: (1) Grid cells are binned according to their concentration (2) The total sea ice extent is computed in the three observational references, in each CMIP5 model, for each day of each year. (3) The extent per bin is normalized by the total extent (for Fig. 5 panels a-d) (4) The normalized extent of each CMIP5 model is compared to the normalized extent of each observation (or to the mean of them?) to give a fractional deviation. I doubt that, out of 10 readers, more than one can replicate Fig. 5 exactly. The authors should detail their approach in a supplementary material, or simplify the metrics.

---

## Short Comment (SC1) · 12 Sep 2017

I found this an interesting read and it is a good addition to the literature, both giving a more comprehensive view of biases and to give a plausible hypothesis. I have three queries/suggestions:

Figure 1 and throughout: I'm unsure of the relevance of DJF; although the traditional meteorological austral summer season, it's arguably not particularly relevant for sea ice, particularly since you do not link analyses to atmospheric variables. However I recognise it's not obvious what the best season would be. I'd suggest showing the minimum or maximum, or if to use DJF, please give some justification (in particular why summer not winter) and mention any sensitivities to season, if found.

[Figure]

Figure 2: This combines spread in information from different years and from different observational data sets. In particular the conclusion in the main text that there is 'no clear bias' at maximum is a little confusing as climatologies are not shown (I would think of 'biases' as referring to climatologies), and the discussion of this figure in the text is very brief; half a sentence or so. Also panel a) appears to be missing outliers? I suggest separating the panels into separate boxplots, particularly for observations, clarifying the multi-model vs multi-yr distinction (if possible), checking the figure caption, and expanding the discussion of this figure a little (it need not be much)

Methodological note: Please say how the regridding is performed (the method and the package used). I have certainly seen cases myself and at meetings (sorry I cannot bring a citation!) which suggest that it can affect results particularly since you are concerned with distributions rather than aggregate measures. Such methodological details are rarely stated in papers about CMIP5, but for reproducibility it they should be!

---

## Referee Comment (RC2) · Anonymous Referee #2 · 23 Sep 2017

This paper was very interesting to read, and is certainly very well written. The result that all CMIP5 models show a too loose sea ice cover around Antarctica is novel and important, as it points to a consistent bias and potentially missing physics. The authors propose that this missing physics might be the treatment of lateral melt in sea ice models used in CMIP5. They perform experiments with one model to evaluate the impact of lateral melt choices. Overall, I think this study is very timely and should be published, after some revisions.

Page 2, Line 30: Why limit the analysis to 2000-2014, when longer observational and model timeseries are available? Longer time series would make the analysis more robust.

Page 2, line 32: Why is daily sea ice concentration used here? This should be ex-

plained, as many more models provide monthly than daily output, and it looks like the authors proceed to average the daily output to seasonal averages.

Page 3, Line 15-17: While I agree that one needs to consider the observational uncertainty, I am not convinced that averaging several products is the best way to do that. First of all, they could all have consistent biases, and hence their range still would not account for the observational uncertainty. Secondly, one of them might be a lot better than the others, and so the combined data might be further from the truth than the best one. So while I am not suggesting that the authors perform an evaluation of the three observations, which is best done by the creators of these data sets, I would encourage the authors to add a sentence or two here to highlight the potential shortcomings of this approach they are using.

Page 3, Line 19: Why was the sea ice output re-gridded, rather than analyzed on the model grids? This can introduce additional errors that have nothing to do with the physics of the model. So there needs to be a good reason to re-grid the model output, otherwise the analysis should be re-done on the original grids. And if the authors have a good reason to do the re-gridding, please include information on how exactly the regridding was done, so it can be replicated by others.

Page 3, Line 27: Why are concentrations below 10% not included? Others included them, so please explain why you would not. For loose sea ice, wouldn't it be important to look at below 10%?

Page 8, line 15, Table 1: Since the authors have the information on whether and how lateral melt is included in the CMIP5 models, do they find any difference between models that include it or not? That would provide an important argument for the hypothesis of the authors that the too loose sea ice concentration is a result of deficiencies in lateral melt.

Page 10, line 24-28: Please remove this entire paragraph. It is pure speculation what modeling centers look at during model development, and this speculation does not add

anything to the arguments or results presented in the paper.

Page 10, Line 29: The observational range is not necessarily fully counted for, as discussed earlier. This should be reflected here.

---

## Author Comment (AC1) · 19 Oct 2017

F. Massonnet (Referee)
francois.massonnet@uclouvain.be

| | Reviewer Comment | Author Response | Author intended action |
|---|---|---|---|
| 1 | "The paper essentially consists in two parts, each conveying its own message: (1) CMIP5 models have consistent biases in the simulation of SIC regimes, (2) The prescribed ice floe diameter affects the simulated SIC regimes The hypothesis of the authors is that (1) is largely caused by (2), hence motivating the development of specific lateral melt processes. However, while I appreciate the efforts of the authors to work on parts (1) and (2) individually, my impression is that they are not able to close the loop and directly prove that the deficiency in simulated SIC regimes in CMIP5 is caused by lateral melt parameterizations." | Our hypothesis is that an under-estimation of lateral melt may be a contributor to the biases noted in the low-concentration regime, but cannot account for biases in the high concentration regime. We have undertaken modelling simulations that support this hypothesis. It was not our intention to suggest that the poor quantification of lateral melt in models is the only possible cause of the biases presented. It almost certainly isn't. We have perturbed that lateral melt in one model sea ice model to show that we can affect a change of the right sign, but have not explored other plausible causes. The language used in the manuscript may have not made this clear enough. We therefore intend to check all wording to clarify that we are suggesting a possible influence, rather than claiming a dominant cause. | Check all wording to clarify that we are suggesting a possible influence, rather than claiming a dominant cause eg. P1, L8 and P8, L13: replace 'partially determined' with 'partially influenced.' Include citations referencing other possible explanations (see author intended action #3). See author response #2. |
| 2 | "In fact, several CMIP5 models do not have such a lateral melting term in their formulation." "To test formally whether (2) explains (1), the authors should separate the CMIP5 subset into two groups: those who use a lateral melting term, and those who don't. If the two sub-groups simulate SIC regimes differently in a statistical sense, then this represents a major step forward to confirming the initial hypothesis of the authors." | We thank the reviewer for this suggestion and we will begin our analysis in the second part of the paper by including such a figure. We did not expect to see a very large difference between models which do not include lateral melt and models which do, given that no model includes a representation of floe diameter. All models must parametrize the impact of lateral melt in some way - otherwise there would be no change in sea ice concentration due to melt – but it may be included explicitly or implicitly. However, using Table 1 to divide the models into two groups (and using the additional models that we can incorporate by using monthly data), we find significant differences in sea ice concentration regimes between the two groups. There is a clear tendency to overestimate the fraction of low concentration ice in models without the explicit lateral melt term. The inclusion of this analysis, together with our modelling simulations which artificially enhance lateral melt, strengthens our hypothesis. | Include a figure showing the sea ice concentration distribution from models with and without an explicit lateral melt term and discuss the outcome. |

| 3 | "In addition, there may be many processes, including dynamical ones, that could explain (1). An important paper on the topic (Lecomte et al., 2016; https://doi.org/10.1016/j.ocemod.2016.08.001) should be cited." | We completely agree with this point and will cite the recommended paper. The recommended paper is consistent with our suggestion that model physics (including dynamical processes) are not currently suitable for the low-concentration regime. | P9, L12: Modify final two sentences to '*Note that we have tested only the impact of lateral melt on this bias. A number of other physical processes, including dynamical ones, may also contribute. Lecomte et al. (2016) find systematic wind-driven biases in sea ice drift speed and direction at the exterior of the Antarctic ice pack. Errors in surface winds could contribute to poor simulation of low-concentration sea ice. However, we find a very strong over-estimation in low-concentration sea ice in the NEMO-CICE model, which is forced by a reanalysis atmosphere and so should not have very unrealistic winds. The dynamical response of sea ice to winds at the edge of the ice may be poorly represented, as we would expect sea ice dynamics to be floe-size dependent. Alternative rheologies (such as a granular rheology (Feltham et al., 2005)) may be better suited to this domain. Concentrations could also be reduced by mechanical interactions between floes. However, we cannot test the impact of such floe-size dependent processes without access to sea ice models that include them.*'

P11, L17. Insert '*Given the possible contribution of dynamic processes to model biases in the sea ice concentration distribution, a full exploration of sea ice dynamics for all CMIP5 models using the sea ice concentration budget decomposition of Uotila et al. (2014) would be welcome.*' |
|---|---|---|---|
| 4 | "In summary, I'm confident that (2) has a significant effect on the simulation of SIC regimes, but I'm challenging the idea that the CMIP5 ensemble is appropriate to test this hypothesis, and that (2) is the dominant cause for (1)." | See author response and intended actions #1 and #3.

The CMIP5 ensemble is the only way we can test this hypothesis until models with floe size information are available. The results presented here motivates developing such models so that the hypothesis can be tested more robustly | See author intended actions #1 and #3. |
| 5 | First, I appreciate the use of multiple observational datasets in the analyses. This is becoming a standard in the community, and this study is fully in line with that approach. What I find worrying is that the three observations are averaged to make a "super observation". This is a problem when non-linear metrics such as the Integrated Ice Area Error (IIAE) are used, because we lose the sense of uncertainty. If the IIAE was conducted separately on each observational reference first and then reported three times, then the impact of observational uncertainty would be immediately visible. Here, instead, the observational mean is used as a reference giving only one IIAE (e.g., Fig. 3). | Although there is precedent for averaging observations and using their mean as a reference, as the true value is not known (eg. Ivanova et al., 2014), we agree that averaging three observations is unsatisfactory. This occurred in the first draft only in the calculation of the integrated ice area error (IIAE) and in Fig. 8 to show a concentration difference. P3, L16 was misleading in suggesting that we did this throughout the analysis. Calculating the IIAE for each observational data set is a good suggestion and we are happy to do this.

Calculating three separate IIAEs means that we cannot show original Fig. 1 in order of increasing IIAE, as the ordering differs slightly according to the observational product used. We could instead show the original Fig. 1 plots ordered alphabetically and | Remove P3, L16.

Remove IIAE from original Fig. 1

Add an additional figure showing the IIAE for each model relative to each observational data set

Show ice errors relative to each observational product separately in original Fig. 3

Show the sea ice concentration difference relative to each observational product separately in original Fig. 8 |

| | | |
|---|---|---|
| | Information regarding observational uncertainty is lost. | add an additional figure which plots the IIAE for each model from each observational product. This may allow us to separate models into 'good', 'middle' and 'poor' categories, without an overall ordering.

For the original Fig. 3, we propose showing the ice errors for models relative to each observational product separately. Preliminary analysis suggests that this makes very little difference to the conclusions drawn from this figure.

For the original Fig. 8, we propose showing the SIC difference relative to each observational product. Again, this makes very little difference to the conclusions drawn from this figure.

For the original Fig. 9, we propose showing the SIC difference relative to one of the observational products. Preliminary analysis suggests that this makes very little difference to the conclusions drawn from this figure - all three observational products show that very few models simulate high enough concentrations in the Weddell Sea. | Show the sea ice concentration difference relative to one observational product in original Fig. 9 |
| 6 | For other figures (e.g., Fig. 2) it would be good to display all three observation products individually in order to gauge how much of the variability is due to the product, and how much is due to sub-seasonal and interannual fluctuations. | We show observational products individually in the original Fig. 4, to show variability due to the product separately from time variability for the normalized SIC distribution. These show that variability in observational products is more significant than interannual fluctuations in the high-concentration regime throughout the year,

This a good suggestion for the original Fig. 2. We propose adding additional panels, with one panel showing boxplots for each model and observational product individually. Preliminary versions of these plots show that the lower to upper quartile ranges for each model (representing only inter-annual variability) are generally smaller than inter-model differences. They also show that the variability due to differences in observational product in sea ice area is more significant at the winter maximum than the summer minimum. The adjacent panel would show the population of all models and the population all observations, as in the original draft. | Add additional panels to Fig. 2 to show all observational products and models individually, as well as the synthesized populations, and discuss the results in the text |
| 7 | Second, I sometimes have the impression that the metrics used are overly complicated. The best example is in Fig. 5. There are so many levels of processing that it becomes difficult to understand | Since the Discussion paper was submitted, from personal communication with D. Notz we have found that our 'binned fractional sea ice extent' is calculated in exactly the same way as the 'normalized sea ice concentration distribution' in Notz (2015). | Replace 'fractional binned sea ice extent' with 'normalized sea ice concentration distribution' throughout text and figures. |

| | | | |
|---|---|---|---|
| | the meaning of the metrics shown on the figure (see also my comment on Fig. 5 below). For the "binned" diagrams (Figs 4-5-6), why not use only three classes: 0-15%, 15-90%, 90-100%? This would simplify the figures much without removing the useful information that deficiencies in the simulation of SIC happen at the edges of the distribution. In general, I have the impression that I could not replicate the figures if I had the original data. | When we read Notz (2015), we were unsure how he normalized the distribution, but it is normalized by grid cell area, as we have done. We therefore intend to adopt the name 'normalized sea ice concentration' from Notz (2015), instead of 'fractional binned sea ice extent.' We hope that is more intuitive for the reader than our original name for this metric.

We would prefer to keep the bins at 10% spacing as this is consistent with Notz (2015) and gives more of a sense of the whole distribution.

Also see response to comments #20 and #30 below. | Refer to Notz (2015) in the Metrics section and say that we use the same approach.

See also author response #20 and #30 below. |
| 8 | Third, the authors introduce new metrics without real justification and make inconsistent choices. For example, they adapt the Goessling et al. "Integrated ice edge error" by moving from a "extent-like" definition to an "area-like" definition. This is initially a good idea, since area is a more physical measure than extent. However, the decomposition into an "absolute error" and a "misplacement error" is not as straightforward, as I'm showing now: consider a model A with uniform SIC of 80% in a number of grid cells, compared to an observational reference with 70% of ice at the same points. Consider also model B, which has half as much ice as A (40% in the grid cells). Even though ice has not been misplaced, the misplacement error term will increase when going from A to B! (While it won't if the Goessling definition was followed). One of the ideas of the Goessling's approach is precisely to use a threshold at 15% to be able to separate total and misplaced errors. The area-like version of that metric looses that property. | We agree that the misplacement area error does not have a physical meaning and are happy to remove it from the paper. This does not affect any of the conclusions of our work. We also agree that the integrated ice area error is a good physical measure, so will continue to use this.

We believe that presenting the decomposition of an integrated ice extent error into misplacement extent error and absolute extent error using the Goessling et al. (2016) approach is useful here, and propose showing them in the updated version of Fig. 3. | Remove misplacement area error from paper.

Adapt Fig. 3 to show (a) The integrated ice area error (IIAE), (b) the integrated ice extent error (IIEE), (c) the absolute extent error as a fraction of the IIEE, and (d) the misplacement extent error as a fraction of the IIEE. (These will be shown relative to each observational product separately, see author intended action #5). |
| 9 | I also don't understand why the authors compute per-bin sea ice extent, and not sea ice area in Figs. 4-5-6. Sticking to ice area would be a more natural choice given the adaptation made earlier of the IIAE. | See response to comment #7 above. We believe that consistency with Notz (2015) is preferable, as it allows comparison with his results for the Arctic. | |
| 10 | Finally, I don't fully understand why the authors used daily data in their analyses. This restricts the number of CMIP5 models available, and adds | We used daily data as some aspects of marginal ice zone behaviour, which is more variable than compact ice, may be visible in the daily data and not in the monthly data. Sea ice models do simulate | Conduct analysis using CMIP5 monthly output and monthly output from the Bootstrap and NASA Team observations (via the Climate Data Record archive). ASI-SSMI observations are only available as daily |

| | | |
|---|---|---|
| | considerable variability to the metrics. By design, CMIP5 models are not supposed to capture synoptic variability in sea ice extent. Using monthly output would partially average out this variability, and would better allow to exhibit the significant biases of the CMIP5 models as the signal-to-noise ratio would increase. | synoptic scale variability in sea ice extent, but it is certainly not a focus of development within CMIP5 class models

We agree with the reviewer's point that using monthly data rather than daily data has the advantage of allowing us to include more models in our inter-comparison. It also allows us to extend the time period of the analysis. We will switch the paper to using monthly mean data for the CMIP5 models. Initial analysis suggests that using monthly data and more models makes no substantial difference to the conclusions drawn from our model-observation comparisons. This strengthens the robustness of results. | output; we will average the sea ice concentration fields each month (using Climate Data Operators, `cdo monavg`).

Update these methodological details in the text |
| 11 | As I'm writing below in a comment, error bars are so large that it is easy to play the devil's advocate and claim that observational references and CMIP5 models are, in the end, not so inconsistent. | We do not show error bars in this paper. We present the data as box plots, indicating the, median, the interquartile range (IQR) and whiskers indicating 1.5 times the lowest datum still within 1.5 IQR of the lower quartile, and the highest datum still within 1.5 IQR. Box plots have the advantage of being non-parametric and can indicate the degree of dispersion and skewness in the data. We have currently suggested that where we see lower quartile – upper quartile ranges that do not overlap, the populations being compared are significantly different.

There are a number of statistical tests that could be applied to examine the differences. We have applied t-tests, but this is a not a particularly useful test as it only considers whether two distributions have different means. Using the test of overlapping confidence intervals as the reviewer hints at would require the assumption of normality and only pass if 95% of the data were different in the two samples. For normalized SIC distributions, which are bounded on the interval [0,1], this is a very difficult test.

After further thought on how to robustly assess inconsistency, we propose including the Kolmogorov–Smirnov test as a robust statistical test of whether the distributions are different. The K-S test is sensitive to differences in both location and shape of the empirical cumulative distribution functions of the two samples.

Preliminary analysis shows that all model – observation comparisons fail the K-S test at the 95% confidence level. However, the p-value obtained from the K-S test, which represents the confidence that the two populations come from the same distribution, is a useful tool to quantify the degree of disagreement. For example, in the original Fig. 5 (a), the p-values from the K-S test | Include further discussion of statistical tests in the text.

Include results from the Kolmogorov–Smirnov test for all two-population comparisons, eg. as annotated text on original Figures 2 and 5, and on the comparison between standard NEMO-CICE and NEMO-CICE with a reduced floe diameter on original Figures 6 and 7. Discuss these results in the text. |

| | | comparing models to observations in each bin are lowest for the 90-100% and the 10-20% bins. This allows us to make the objective conclusion that model-observation difference is most significant for low-concentration and high-concentration sea ice.

Finally, it's worth noting that our main aim, and what we find most interesting about the paper, is the identification of consistent behaviours amongst the CMIP5 models, ie. tendencies in one particular direction – rather than any claim of simple model-observation disagreement. | |
|---|---|---|---|
| 12 | p. 2, l. 3: The statement that CMIP5 has improved in the simulation of Arctic sea ice compared to CMIP3 is strong, and perhaps too strong. Rosenblum and Eisenman (2016, http://dx.doi.org/10.1175/JCLI-D-16-0391.s1), for example, suggest that this could be due to the omission of volcanic forcings in several CMIP3 models. Also, it is unclear if the CMIP5 to CMIP3 differences actually reflect improvements, or changes in tuning strategy (e.g. Notz, 2015, http://dx.doi.org/10.1098/rsta.2014.0164). Please nuance this statement. | We agree with this comment | Replace '*Advances in Earth system modelling have improved simulation of Arctic sea ice compared to the previous intercomparison project (CMIP3) (Stroeve et al., 2012)*' with '*Advances in Earth system modelling have somewhat improved simulation of Arctic sea ice compared to the previous intercomparison project (CMIP3) (Stroeve et al., 2012), although this may reflect changes in forcings (Rosenblum & Eisenman, 2016) or tuning strategy (Notz, 2015) rather than changes in model physics*.' |
| 13 | p. 2, l. 8: evaluates –> evaluate. | Agree | p. 2, l. 8: evaluates –> evaluate. |
| 14 | p. 2, l. 28: … Phase 5 (CMIP5; Taylor et al., 2012). | Agree | p. 2, l. 28: … Phase 5 (CMIP5; Taylor et al., 2012). |
| 15 | p. 2, l. 30: Years prior to 2000 are neglected in the model evaluation. Have the authors still conducted the evaluation on 1979-2000, for which the CMIP5 output and the observations are readily available? Do conclusions of the study hold? Please elaborate. It would be useful to present the results of such analyses in the supplementary material, to test the stability of the metric developed over time. | Our use of daily data meant that were limited (in memory) in our analysis. Use of monthly data means that we can consider a longer time series, and is indeed a more sensible solution. The ASI-SSMI observations begin in 1992, so we plan to do analysis over 1992-2014. Initial analysis suggests that including a longer timeframe of data does not alter the conclusions. | Conduct analysis using model output and observational data from 1992-2014.

Swap the order of Subsections 2.1 and 2.2 in Methods. Add a sentence to stay that ASI-SSMI observations begin in 1992, so we conduct analysis over 1992-2014. |
| 16 | p. 2., l. 31: Could the authors conduct their diagnostics on two members of the same climate model, in order to gauge how much internal variability affects the metrics developed in this study? | It is correct that we have only used one ensemble member from each model at this point. We do not believe this is problematic for the results of the paper. But we agree that the reviewer has proposed an interesting question, and as part of revising the manuscript we will do an analysis of ensemble members from one model to see how much spread is attributable to internal variability. We could use the K-S test to quantify this difference. | Will undertake analysis and comment on, or include, result. |
| 17 | p. 3., l. 3: Three observational products were used in the study. Since part of the study describes the differences between sea ice concentration in those products, it would be useful to have a few lines | Agree | Replace P3, L1-14 with: '*Passive microwave radiometers deployed on satellites measure the brightness temperature of the Earth's surface, and can be used to calculate sea ice concentration. Various observational data sets apply different algorithms to convert passive-* |

| | | |
|---|---|---|
| describing differences in algorithms between those products. | | *microwave signals into sea ice concentration, reflecting the uncertainty in satellite observations (Bunzel et al., 2016). As summarized by Ivanova et al. (2014), differences between algorithms are caused by 1. choice of radiometer channels; 2. tie-points, which are the brightness temperatures used to identify different surfaces; 3. sensitivities to changes in physical temperature of the surface; and 4. weather filters, which correct for atmospheric effects falsely indicating the presence of sea ice.*

*To account for some of this product uncertainty, we use three observational data sets: the Bootstrap algorithm (Comiso, 1986), the NASA Team algorithm (Cavalieri et al., 1984) and the ASI algorithm (Kaleschke et al., 2001; Spreen et al., 2008). We do not consider datasets that merge different observation methodologies. Bootstrap uses cluster analysis of brightness temperatures from two channels (19 GHz and 37 GHz vertical polarization in the Antarctic), applies an ocean mask and is available from 1979 at a resolution of 25 km. NASA Team uses ratios of brightness temperatures (which tends to cancel out physical temperature effects) from three channels (19 GHz in the vertical and horizontal, 37 GHz in the vertical), removes weather contamination based on certain spectral gradient ratios and is available from 1979 at a resolution of 25 km. The ASI algorithm uses the difference in brightness temperatures between horizontal and vertical polarization at 85 GHz, uses lower frequency channels at lower resolution to filter atmospheric effects (which are more apparent at 85 GHz than lower frequencies), and is available from 1992 at a resolution of 12 km. We choose to conduct our analysis over 1992-2014.*

*Differences between the three selected data sets are large: in the Antarctic, the NASA Team algorithm shows the marginal ice zone (defined as the extent of sea ice with concentration between 15 % and 80 %) to extend over 2 million km more than the Bootstrap algorithm (Stroeve et al., 2016). NASA Team is more sensitive to clouds and wind over open water than the Bootstrap mode (Anderson et al., 2006), while the high-frequency ASI algorithm is also sensitive to such atmospheric effects (Spreen et al., 2008). Bootstrap is more sensitive to physical temperature changes than NASA Team, and may underestimate concentrations at low temperatures, such as near the Antarctic coast (Comiso et al. 1997). For low concentrations, atmospheric effects, which generally lead to falsely increased sea ice, become increasingly important (Anderson et al., 2006). The weather filters/ocean masks used to correct these differ between the different algorithms.'* |

| 18 | p. 3., l. 5-7: To what season does the statement on marginal ice zone area difference apply? I think it's winter, please specify. | It's winter, September and October specifically | Insert 'in the winter months' in P3, L7 |
|---|---|---|---|
| 19 | p. 3., l. 22: "Sea ice area is the sum of the area of all grid cells with more than 15 % sea ice concentration multiplied by the sea ice concentration in each grid cell". Following the conventional NSIDC definition I would have thought that area is just the product of ice concentration by grid cell area, summed over the domain (https://nsidc.org/cryosphere/seaice/data/terminology.html). Why only considering the grid cells with > 15% of ice? | Other locations on the NSIDC website show the definition we used in the original draft. For example see: https://nsidc.org/data/docs/noaa/g02135_seaice_index/#comp_area where it states, 'The monthly average sea ice area calculation is performed through simple pixel-by-pixel arithmetic of multiplying the daily concentration by the size of the grid cell[1], for all grid cells which satisfy the 15 percent threshold and then averaging them together for a month'. Also consider: http://nsidc.org/arcticseaicenews/faq/#area_extent which states "Area takes the percentages of sea ice within data cells and adds them up to report how much of the Arctic is covered by ice; area typically uses a threshold of 15%." We note that Notz (2015) and Turner et al. (2017) do not use a 15% cutoff, while Ivanova (2016) does use a 15% cutoff.

We conclude that either definition is acceptable, as long as it is stated clearly. We also must be consistent - a 15% cutoff should be used for the IIAE if it used for the SIA.

As we use the IIEE and associated MEE and AEE, which by the Goessling definition use a 15% cutoff, we think that the most consistent approach is to use the 15% cutoff. | |
| 20 | p. 3., l. 25: Why is the bin (0-10%] not in the list? | As shown in Ivanova et al. (2016) (Fig. 2d), the CMIP5 multi-model mean and the NASA Team observations have a high fraction of ice below 10% sea ice concentration in the summer. We find that the fraction of <10%-concentration ice varies in the models from 0.005 to 1.0 (when models are essentially ice-free) in the summer. It consists of up to around a third of the ice in other seasons for some models.

Including these very low concentrations therefore heavily skews the normalized SIC distribution towards low concentrations. It obscures behaviour at higher concentrations. Our aim is to look for consistent model behaviour; with such variance between different models and between different observations at very low concentrations, it's difficult to conclude anything about model tendencies. | Update Fig. 1 to show 0.1-10% sea ice concentration.

Explain that some models (refer to Fig.1 and Ivanova et al., 2016) have very large numbers of cells with very small concentrations (0.1-10%), noting that the fraction varies greatly between models, and substantially between the three observational products. Including these very low concentrations heavily skews the normalized distribution, so we exclude them from the SIC distributions. |
| 21 | p. 4., l. 20: "A disadvantage of the IIAE is that it does not take into account the observational range, using | See response to comment #5. | See response to comment #5. |

| | | | |
|---|---|---|---|
| | only the observational mean as the 'true' state". That's not really a disadvantage of the metric, but rather a methodological issue. The authors could repeat the IIAE taking successively the three observations as references. They would obtain three IIAE's, which would give a sense on the uncertainty associated to the products. Why didn't the authors go this way? See also my first comment on methodology. | | |
| 22 | p. 6, l. 17. "[Ocean-sea ice] Model output is analysed on its native grid". Does that mean that the observations were then interpolated onto the NEMO-CICE grid? At page 9, line 18 and in Fig. 8 the NEMO-CICE model is evaluated using the IIAE metric, this means that at some point an interpolation must take place, correct? The model output and the observational reference need to be on the same grid for Eqs (2) and (3) to be evaluated. Why didn't the authors interpolate the NEMO-CICE output on the same target grid as all CMIP5 models, to ensure consistent analysis? | Following comments by other reviewers (anonymous review #2 and C Holmes), as well discussion with others in the community, we have thought more carefully about the interpolation. We believe it is preferable to avoid interpolation as much as possible.

This is possible for sea ice area and the normalized sea ice concentration distributions, which we thus propose to calculate on original grids. Preliminary analysis suggests that model-observation differences in the normalized sea ice concentration distributions at low concentrations are slightly reduced when conducting the analysis on the native grids. We intend to state that the normalized sea ice concentration distributions show some sensitivity to grid interpolation in the Metrics subsection.

Integrated ice errors and sea ice concentration differences between models and observations must be calculated on the same grid. In this case, we propose interpolating onto a regular 1 degree grid using Climate Data Operators bilinear interpolation function and state this in the manuscript. This may cause some smoothing of the ice edge. This will have a negligible effect on integrated ice errors and sea ice concentration differences at high concentrations (original Fig. 9). It may impact sea ice concentration differences at low concentrations (original Fig. 8). We shall investigate whether using bilinear or nearest-neighbour interpolation results in differences in the original Fig. 8. | Calculate sea ice area and normalized sea ice concentration distributions on original grids and state this in the manuscript. State in Subsec. 2.3 that there is some sensitivity to grid interpolation for the SIC distributions.

Calculate integrated ice area/extent errors and sea ice concentration differences after interpolation onto a regular 1 degree grid using an appropriate Climate Data Operators (https://code.mpimet.mpg.de/projects/cdo/) interpolation function and state this in the manuscript. |
| 23 | p. 6, l. 20 and Fig. 2. "Sea ice area at the annual minimum is consistently biased low". Here I'm playing the devil's advocate. The blue boxes in Fig. 2 display the distribution of the three observational references, which are three times the same climate realization plus noise due to the retrieval algorithm. Hence these blue boxes embody time-variability and product variability. By contrast, the green boxes in | See response to comment #11. In summary, to more robustly assess inconsistency of distributions we will include the Kolmogorov–Smirnov test as a robust statistical test of whether the distributions are different. The K-S test is sensitive to differences in both location and shape of the empirical cumulative distribution functions of the two samples. | See author intended action #11.

See author intended action #40.

See author intended action #6 |

| | | | |
|---|---|---|---|
| | Fig. 2 contain time-variability internal variability, and model error. So, the whole question is whether these observational references are incompatible in a statistical sense with the models. Put differently, could the observations be the (N + 1)th CMIP model? Judging from Fig. 2b, the observations lie in the range [1.5 * IQ_75%, IQ_75%] and they could be one of the CMIP models. Or couldn't they? A more quantitative test would be welcome. | Also see response to comment #6. Separating boxplots would allow us to discuss the contributions of model/observational product variability and time variability | |
| 24 | p. 7, l. 5-8. The sentence "Such total errors..." is unclear. First, there are many more possible reasons than just ocean/atmosphere temperature biases to explain differences in total area (which is captured by the AAE). Second, it's not clear why looking at the bias per concentration bin would help isolate the role of the sea ice component. | The case we wanted to make is that sea ice extent will be different in a model with a normal ocean compared to one that is 1 degree warmer everywhere. However, both models could still stimulate an appropriate normalized sea ice concentration distribution. Therefore we would argue that a normalized sea ice concentration distribution depends less on overall ocean/atmosphere temperature biases than sea ice extent. | Replace P7, L5-8 with: '*We now consider sea ice concentration distributions from observations and models, which provide a more detailed assessment than hemisphere-integrated measures. A normalized sea ice concentration distribution may help isolate the role of the sea ice component, as models with a constant temperature bias in the atmosphere or ocean, resulting in a biased sea ice area or extent, may still simulate the relative fraction of different concentration regimes successfully.*' |
| 25 | p. 7, l. 10. Notz (2014) "uses" or "used" but not "use". | Agree | Correct to Notz (2014) "uses" |
| 26 | p. 7, l. 22-35. The two paragraphs deliver somewhat contradictory messages. The first one finishes by "that the sea ice components of CMIP5 models are somewhat successful at simulating the distribution of sea ice concentration" while the second paragraph says "large discrepancies between models and observations in the highest and especially the lowest concentration bins". This would need better rephrasing, saying for example in the first paragraph that the "big pictures" are consistent but that this is mostly thanks to cancellation of errors, as explained in the second paragraph. | We agree with the reviewer and will reword accordingly. The use of the K-S test to quantify the differences between the two population is useful here. As explained above, the p-value from the K-S test, which represents the confidence that the two populations come from the same distribution, is highest for the 90-100% and 10-20% bins in DJF. | Reword and include the outcome of the K-S test in the presentation of these results |
| 27 | p. 10, l. 4-6 and Fig. 9. The authors conclude that the systematic underestimation of highly concentrated ice in the Weddell Sea is related to melt or break-up processes. Why is the possibility of a systematic misrepresentation of dynamics ruled out? It could be that all models have deficiencies in capturing the Weddell gyre dynamics. It could be that models are neutral to divergent while observations are in convergent motion. I haven't tested this hypothesis myself, but I don't have enough information from the results of the paper to rule out properly this alternative hypothesis. An exploration of how the | We agree that dynamic processes are a possible cause of the underestimation of highly concentrated ice. The findings of Lecomte et al. (2016) are particularly relevant here. They suggest that models with high ice drift speeds in coastal areas simulate a faster sea ice retreat. These high drift speeds may be influenced by sea ice rheology as well as wind speeds. | Discuss the possible contribution of sea ice dynamics to this bias, with reference to Lecomte et al. (2016). Also see author intended action #3 |

| | | | |
|---|---|---|---|
| | models simulate Antarctic ice drift could be helpful in that respect. | | |
| 28 | p. 10, l. 29: 2. Three observational products are considered in this study. While I appreciate this effort, it looks sometimes like the authors assume that observational errors are random and that the mean of all three products is representative of the truth. It could be that the three observational products have a systematic bias with respect to truth, which could explain model-obs mismatch on top of model error.

The authors don't seem to explore this possibility in the assessment. For example it is known that most algorithms underestimate sea ice concentration as ice becomes very thin. Could this explain the model-obs differences, in particular differences in binned sea ice extent? It is also known that wet snow has a brightness temperature that makes sea ice concentration retrievals higher than they should be. Could this have an impact? More discussion on observational systematic errors would be welcome, in order to place the CMIP results in perspective. | As discussed above, we plan to do as suggested and avoid use of any observational mean in this study.

We agree that including different observational products will give some estimation of error arising from differences in processing satellite data, but there is still the possibility of systematic errors common to all three observational products. In the Discussion, we did briefly discuss systematic errors in the observations:
*'Accounting for the observational range, we find that models overestimate the extent of low-concentration sea ice throughout the year, while underestimating the extent of high-concentration sea ice in summer. This common behaviour across diverse models with varying physics is a result not previously highlighted and warrants further attention. We note that using the observational range as an uncertainty estimate neglects biases that are common to the three different satellite observations. As mentioned above, satellite observations of sea ice are most uncertain in summer. However, we see the bias in low concentration ice from CMIP5 models throughout the year, and observed summer high-concentration ice is unlikely to be affected by the melt processes that complicate satellite retrievals. The suggestion that the NASA Team algorithm overestimates low-concentration ice (Steffen & Schweigher, 1991) would further strengthen the contrast between models and observations in this regime.'*

We agree that this is worthy of more comprehensive discussion within the paper and will expand on this discussion point. | Add the following paragraph to subsection 2.2:
*Besides structural uncertainty in observational algorithms, systematic biases common to all three products are possible. Lack of validation data (Ivanova et al., 2014) mean it is difficult to quantify this, but accuracy is understood to be lower in the presence of melt ponds or other surface melt effects (Ivanova et al., 2014), which may act to lower retrieved concentrations; large fractions of thin ice (Cavalieri, 1995); and stormy conditions near low concentrations (Anderson et al., 2006). Transitions between ice type can cause differences in emissivity (Grenfell and Comiso, 1986), but because models do not simulate ice types such as grease ice, this issue should not impact model-observation comparisons.*

Add to the discussion: *As mentioned above, sea ice concentrations are considered to be most uncertain during melt conditions, for large fractions of thin ice and at low concentrations during storms. In the context of the results from the model-observation comparison for normalized sea ice concentration distributions, we suggest that the impact of uncertainty of melt conditions is limited as the high bias in low-concentration ice from CMIP5 models is visible throughout the year. The low bias in high-concentration ice during the melt season would be strengthened if observations were underestimating ice concentrations in this season. Inclusion of both NASA Team and Bootstrap algorithms, with the former tending to cancel out physical temperature effects, will sample some of this uncertainty. The underestimation of sea ice concentrations in areas of thin ice (<35 cm) (Ivanova et al., 2015) may cause a bias at any concentration in the observed normalized sea ice concentration distribution from observations, with the possibility of a positive bias in the very lowest concentrations. Stormy conditions near the ice edge lead to false sea ice concentrations near the ice edge; weather filters may accurately remove these, leave them uncorrected (Anderson et al., 2006) or erroneously remove real sea ice. The latter may underestimate low concentrations (personal communication, S. Kern). Spreen et al. (2008) suggest the filter method used in ASI observations may result in a positive bias in the marginal ice zone, and Steffen & Schweiger (1991) found that the NASA Team algorithm overestimates low-concentration ice when compared to Landsat imagery. Considering all this evidence we suggest that the magnitude or sign of any systematic biases in satellite radiometer observations is unclear when comparing with climate models. This is particularly true for low concentrations. Here the* |

| | | | |
|---|---|---|---|
| | | | *use of different approaches to weather filters within the different algorithms may assist in sampling observational uncertainty. Development of sea ice satellite emulators, which use climate model output to calculate brightness temperatures (eg. Tonboe et al., 2011), may help to reduce uncertainty when comparing models to observations in the future.* |
| 29 | Fig. 1. Interestingly, it is possible that two CMIP5 models with similar IIAEs (e.g., MIROC5 (4.6 Mkm2) and FGOALS-g2 (4.85 Mkm2)) have drastically different sea ice concentration patterns (one with not enough ice and one with way too much ice; panels ac and ad). In the same vein, two models with similar patterns (e.g. CMCC-CMS and HadGEM2-CC) may have very different IIAEs. This is because of the definition of IIAE which penalizes over- and underestimation in the same way. The authors should comment on that aspect (which I see as a weakness of that metric). Although there is no definition of what a "good" metric is, we could expect that it satisfies properties of continuity in some sense: two models close to each other should have similar metric values. | Is there are reason to favour over-estimation or under-estimation? We don't see that one is better than the other, so we don't see the lack of distinction between the two in the IIAE to be an issue. | Explicitly state in the text that the IIAE does not favour over-estimation or under-estimation |
| 30 | Fig. 5. This is one example where I would have difficulties in reproducing the result. If I follow correctly, from Fig. 5 caption and from the text: (1) Grid cells are binned according to their concentration (2) The total sea ice extent is computed in the three observational references, in each CMIP5 model, for each day of each year. (3) The extent per bin is normalized by the total extent (for Fig. 5 panels a-d) (4) The normalized extent of each CMIP5 model is compared to the normalized extent of each observation (or to the mean of them?) to give a fractional deviation. I doubt that, out of 10 readers, more than one can replicate Fig. 5 exactly. The authors should detail their approach in a supplementary material, or simplify the metrics. | See response to comment #7 above. Our calculation of fractional binned sea ice extent/normalized sea ice concentration distribution is the same as Notz (2015). We propose explicitly staying the steps involved in the Methods section.

We agree that Figs (e-h) are confusing and are happy to take a different approach. The aim of Figs(e-h) in the first draft was to show the biases independent of scale, but we are happy to simply remove (e-h) from the manuscript. | See author intended response #7 above.

Add further detail in Methods to explicitly explain the steps used to calculate the normalized sea ice concentration distribution. '*The sea ice concentration distribution for each model or observational product is calculated by binning grid cells according to their concentration at a 10%-spacing. The distribution is then normalized by the area of grid cells.*'

Remove Figs(e-h) |

Interactive comment on "Consistent biases in
Antarctic sea ice concentration simulated by
climate models" by Lettie A. Roach et al.
Anonymous Referee #2

| | Reviewer Comment | Author Response | Author intended action |
|---|---|---|---|
| 31 | Page 2, Line 30: Why limit the analysis to 2000-2014, when longer observational and model timeseries are available? Longer time series would make the analysis more robust. | This comment and the comment below are connected – our use of daily data meant that we were limited (in memory) in our analysis. Use of monthly data means that we can consider a longer time series, and this is indeed a more sensible solution. The ASI-SSMI observations begin in 1992, so we plan to do analysis over 1992-2014.

See author response to comment #15 above | See author intended action #15 |
| 32 | Page 2, line 32: Why is daily sea ice concentration used here? This should be explained, as many more models provide monthly than daily output, and it looks like the authors proceed to average the daily output to seasonal averages. | See author response to comment #10 above | See author intended action #10 above |
| 33 | Page 3, Line 15-17: While I agree that one needs to consider the observational uncertainty, I am not convinced that averaging several products is the best way to do that.

First of all, they could all have consistent biases, and hence their range still would not account for the observational uncertainty. Secondly, one of them might be a lot better than the others, and so the combined data might be further from the truth than the best one. So while I am not suggesting that the authors perform an evaluation of the three observations, which is best done by the creators of these data sets, I would encourage the authors to add a sentence or two here to highlight the potential shortcomings of this approach they are using. | We agree with your points. Please see author response to comment #5 above regarding averaging observational products.

We do combine the three sets of observations in original Fig.s 5-7 for the concentration distributions. It is not clear to us from the literature that any of the three datasets is better than the others. Evaluation of the products is indeed beyond the scope of this manuscript.

Further, Ivanova (2014) states that 'we cannot establish an absolute ranking of the performance of the algorithms because of the lack of good validation data,' and recommends constructing an ensemble of different observational products. | See author intended action #5

P3, L15: Replace the final paragraph in original subsection 2.2 with: '*In this study, for some of the analysis we consider the three observational data sets individually. In order to compare the sea ice concentration distribution from the set of models against observations, we create an ensemble of the ASI, Bootstrap and NASA Team observational products. Combining the observational products in this way does have limitations, as different algorithms are likely to perform better for certain sea ice conditions and seasons. However, it is not clear from the literature where exactly the strengths of the various algorithms lie, and evaluation of the different algorithms is beyond the scope of this manuscript. The difficulty in ranking various observational algorithms is noted by Ivanova et al. (2014), due to a lack of validation data. They recommend constructing an ensemble of different observational products.*' |
| 34 | Page 3, Line 19: Why was the sea ice output re-gridded, rather than analyzed on the model grids? This can introduce additional errors that have nothing to do with the physics of the model. So there needs to be a good reason to re-grid the model output, otherwise the analysis should be re-done on the original grids. And if the authors have a good reason to do the re-gridding, please include information on how exactly the regridding was done, so it can be replicated by others. | See author response to comment #22 above | See author intended action #22 above |

| | | | |
|---|---|---|---|
| 35 | Page 3, Line 27: Why are concentrations below 10% not included? Others included them, so please explain why you would not. For loose sea ice, wouldn't it be important to look at below 10%? | See author response to comment #20 above | See author intended action #20 above |
| 36 | Page 8, line 15, Table 1: Since the authors have the information on whether and how lateral melt is included in the CMIP5 models, do they find any difference between models that include it or not? That would provide an important argument for the hypothesis of the authors that the too loose sea ice concentration is a result of deficiencies in lateral melt. | See author response to comment #2 above | See author intended action #2 above |
| 37 | Page 10, line 24-28: Please remove this entire paragraph. It is pure speculation what modeling centers look at during model development, and this speculation does not add anything to the arguments or results presented in the paper. | Agreed. We are happy to remove this. | Remove lines 24-28 |
| 38 | Page 10, Line 29: The observational range is not necessarily fully counted for, as discussed earlier. This should be reflected here. | See author response to comment #28 above | See author intended action #28 above

Replace P10, L29 'Accounting for the observational range' with 'Accounting for the range in three observational products.' |

Interactive comment on "Consistent biases in
Antarctic sea ice concentration simulated by
climate models" by Lettie A. Roach et al.
C. Holmes
calmes@bas.ac.uk

| | Reviewer Comment | Author Response | Author intended action |
|---|---|---|---|
| 39 | Figure 1 and throughout:  I'm unsure of the relevance of DJF; although the traditional meteorological austral summer season, it's arguably not particularly relevant for sea ice, particularly since you do not link analyses to atmospheric variables.  However I recognise it's not obvious what the best season would be.   I'd suggest showing the minimum or maximum, or if to use DJF, please give some justification (in particular why summer not winter) and mention any sensitivities to season, if found. | Our interest in summer stems from Fig. 2 (SIA), where we looked at sea ice area from models versus observations and concluded that the minimum showed more disagreement with observations than the maximum. This is further supported by the analysis in Fig.3 (IIAE). We therefore chose to examine the months leading up to the summer minimum (DJF) in more detail.  We chose to show DJF, MAM, JJA, SON in original Fig. 5 as we wanted to include data from all months.

The normalized SIC distribution for DJF shows largest differences from observations at the high and low ends of the distribution. We propose looking at the low (10-20%) and high (90-100%) | Explore the seasonality of results (for original Fig. 5) or suitably justify our interest in a particular time period (for original Fig. 1 and results from the lateral melt experiments) |

| | | | |
|---|---|---|---|
| | | concentrations bins throughout the year, as there are some sensitivities to season.

In response to the other reviews, we are significantly updating the figures. Wherever relevant, we will explore the seasonality of processes or suitably justify our interest in a particular time period. For example, for the second part where we investigate the impact of lateral melt, we could calculate the month(s) where the impact of lateral melt is greatest, and show results for this time period | |
| 40 | Figure 2: This combines spread in information from different years and from different observational data sets. In particular the conclusion in the main text that there is 'no clear bias' at maximum is a little confusing as climatologies are not shown (I would think of 'biases' as referring to climatologies), and the discussion of this figure in the text is very brief; half a sentence or so. Also panel a) appears to be missing outliers? I suggest separating the panels into separate boxplots, particularly for observations, clarifying the multi-model vs multi-yr distinction (if possible), checking the figure caption, and expanding the discussion of this figure a little (it need not be much) | Fig. 2 shows that the interquartile range of the CMIP5 models overlaps that of the observations for the sea ice area maximum, but it does not for the sea ice area minimum. We conclude that there is a tendency for models to underestimate the sea ice area minimum, but there is not such a significant tendency at the sea ice area maximum. This conclusion can be quantified using the K-S test, as discussed above.

We agree with the suggestion of separating out the boxplots, see author response #6.

The data in Fig. 2a does not have outliers when the whiskers are set to 1.5 of the interquartile range. | P6, L20 Replace '*While sea ice area at the annual maximum has a large inter-model and inter-annual spread with no clear bias compared to observations, sea ice area at the annual minimum is consistently biased low.*' with discussion of the separated boxplots and use of the K-S test to quantify the degree of difference between models and observations. Replace 'clear bias' with 'significant tendency.'

See author intended action #6 |
| 41 | Methodological note: Please say how the regridding is performed (the method and the package used). I have certainly seen cases myself and at meetings (sorry I cannot bring a citation!) which suggest that it can affect results particularly since you are concerned with distributions rather than aggregate measures. Such methodological details are rarely stated in papers about CMIP5, but for reproducibility it they should be! | The comment on the impact of regridding is correct. See author response to comment #22 above | See author intended action #22 |

---

## Author Response (AR1)

F. Massonnet (Referee)
francois.massonnet@uclouvain.be

| | Reviewer Comment | Author Response | Author completed action |
|---|---|---|---|
| 1 | The paper essentially consists in two parts, each conveying its own message: (1) CMIP5 models have consistent biases in the simulation of SIC regimes, (2) The prescribed ice floe diameter affects the simulated SIC regimes The hypothesis of the authors is that (1) is largely caused by (2), hence motivating the development of specific lateral melt processes. However, while I appreciate the efforts of the authors to work on parts (1) and (2) individually, my impression is that they are not able to close the loop and directly prove that the deficiency in simulated SIC regimes in CMIP5 is caused by lateral melt parameterizations." | Our hypothesis is that an under-estimation of lateral melt may be a contributor to the biases noted in the low-concentration regime, but cannot account for biases in the high concentration regime. We have undertaken modelling simulations that support this hypothesis. It was not our intention to suggest that the poor quantification of lateral melt in models is the only possible cause of the biases presented. There are certainly other possibilities. We have perturbed that lateral melt in one sea ice model to show that we can effect a change of the right sign, but have not explored any other plausible causes. Following author action #2, the evidence supporting our hypothesis is now stronger. The language used in the original manuscript may have been too strong and our revisions now clarify that we are suggesting a possible influence, rather than claiming a single cause. | P1, L7 [current draft P1, L9] Replaced '*Targeted model experiments with a coupled ocean — sea ice model show that over-estimation of low-concentration cover is partially determined by choice of constant floe diameter in the lateral melt scheme.*' with '*Targeted model experiments with a coupled ocean --- sea ice model show that choice of constant floe diameter in the lateral melt scheme can also impact representation of loose ice.*'

 P8, L13 [current draft P9, L29] Replaced 'partially determined' with 'partially influenced'

 P8, L6 [current draft P9, L22] Replaced '*It suggests that there is some deficiency or missing physical process common to all models* ' with '*It suggests that there is some deficiency or missing physical process common to many models.*

 Added '*in many models*' [current draft P1, L12]

 See author actions #2 and #3. |
| 2 | "In fact, several CMIP5 models do not have such a lateral melting term in their formulation." "To test formally whether (2) explains (1), the authors should separate the CMIP5 subset into two groups: those who use a lateral melting term, and those who don't. If the two sub-groups simulate SIC regimes differently in a statistical sense, then this | We thank the reviewer for this suggestion and we now begin our analysis in the second part of the paper by including such a figure. We did not expect to see a very large difference between models which do not include lateral melt and models which do, given that no model includes a representation of floe diameter. All models must parametrize the impact of lateral melt in some way - | [current draft P1, L8] Added '*Separating models with and without an explicit lateral melt term, we find that inclusion of lateral melt can partially account for over-estimation of low-concentration cover.*'

 [current draft, P9, L30] Added '*Separating models with and without an explicit lateral melt term (Table 1), we find a significant difference between the two groups. Models with explicit lateral melt show a* |

| | | | |
|---|---|---|---|
| | represents a major step forward to confirming the initial hypothesis of the authors." | otherwise there would be no change in sea ice concentration due to melt – but it may be included explicitly or implicitly.

However, using Table 1 to divide the models into two groups (and using the additional models that we can incorporate by using monthly data), we find significant differences in sea ice concentration regimes between the two groups. There is a clear tendency to overestimate the fraction of low concentration ice in models without the explicit lateral melt term.

The inclusion of this analysis, together with our modelling simulations which artificially enhance lateral melt, strengthens our hypothesis. | *greatly reduced fraction of low-concentration ice in from March to July compared to models without, in good agreement with the observations (Fig. 7). Lateral melt can occur all year at the ice edge, where low concentrations occur.*
    *Fig. 7 demonstrates that lateral melt significantly impacts the normalized sea ice concentration distribution during autumn. However, lateral melt as it is currently included in CMIP5 models still results in a tendency towards overestimation of low-concentration sea ice in other months, and some models with an explicit lateral melt term (including the ocean --- sea ice model NEMO-CICE) still simulate too large a fraction of loose ice.*
    *We therefore proceed by examining whether changes to the lateral melt scheme may also impact the simulation of sea ice. The current representation of lateral melt in CMIP5 models is heavily parametrized (Table 1) with the formulation described in Subsect. 2.4 being the most complex parametrization available in the CMIP5 models.'*

[current draft P13, L19] Added '*Categorising models according to whether they explicitly represent lateral melting, which is the only thermodynamic sea ice process that reduces concentrations in models, we find a strong impact of this process on low-concentration sea ice.'*

[current draft, Fig. 7] New figure, with caption: '*The 10-20% bin from the normalized sea ice concentration distribution for each month, where boxes contain all years from 1992 to 2014 from (blue) the the three sets of satellite observations, (purple) CMIP5 models that include an explicit lateral melt term and (grey) CMIP5 models that do not (from Table 1). Boxplots as in Fig. 1. Annotated text is the p-value calculated from a Kolmogorov-Smirnov test, which represents the confidence that the two populations come from the same distribution'* |
| 3 | "In addition, there may be many processes, including dynamical ones, that could explain (1). An important paper on the topic (Lecomte et al., 2016; https://doi.org/10.1016/j.ocemod.2016.08.001) should be cited." | We completely agree with this point and now cite the recommended paper. The recommended paper is consistent with our suggestion that model physics (including dynamical processes) are not currently suitable for the low-concentration regime. | P9, L12 [current draft P11, L18] Modified final two sentences to '*Besides lateral melt, a number of other physical processes, including dynamical ones, may also contribute to an overestimation of low-concentration ice. Lecomte et al. (2016) find systematic wind-driven biases in sea ice drift speed and direction at the exterior of the Antarctic ice pack. Errors in surface winds could contribute to poor simulation of low-concentration sea ice. However, we find a very strong over-estimation in low-concentration sea ice in the NEMO-CICE model, which is forced by a reanalysis atmosphere and so should not have very unrealistic winds. The dynamical response of sea ice to winds at the edge of the ice may be poorly represented, as we would expect sea ice dynamics to be floe-size dependent. Alternative rheologies (such as a granular rheology, eg. Feltham, 2005) may be better suited to this domain. Concentrations could also be reduced by mechanical* |

| | | | |
|---|---|---|---|
| | | | *interactions between floes. However, we cannot test the impact of such floe-size dependent processes without access to sea ice models that include them.'*

P11, L17 [current draft P13, L35] Inserted *'Given the possible contribution of dynamic processes to model biases in the sea ice concentration distribution, a full exploration of sea ice dynamics for all CMIP5 models using the sea ice concentration budget decomposition of Uotila et al. (2014) would be welcome.'* |
| 4 | "In summary, I'm confident that (2) has a significant effect on the simulation of SIC regimes, but I'm challenging the idea that the CMIP5 ensemble is appropriate to test this hypothesis, and that (2) is the dominant cause for (1)." | See author response and intended actions #1 and #3.

The CMIP5 ensemble is the only way we can test this hypothesis until models with floe size information are available. The results presented here motivates developing such models so that the hypothesis can be tested more robustly | See author actions #1 and #3. |
| 5 | First, I appreciate the use of multiple observational datasets in the analyses. This is becoming a standard in the community, and this study is fully in line with that approach. What I find worrying is that the three observations are averaged to make a "super observation". This is a problem when non-linear metrics such as the Integrated Ice Area Error (IIAE) are used, because we lose the sense of uncertainty. If the IIAE was conducted separately on each observational reference first and then reported three times, then the impact of observational uncertainty would be immediately visible. Here, instead, the observational mean is used as a reference giving only one IIAE (e.g., Fig. 3). Information regarding observational uncertainty is lost. | Although there is precedent for averaging observations and using their mean as a reference, as the true value is not known (eg. Ivanova et al., 2014), we agree that averaging three observations is unsatisfactory. This occurred in the first draft only in the calculation of the integrated ice area error (IIAE) and in Fig. 8 to show a concentration difference. P3, L16 was misleading in suggesting that we did this throughout the analysis. Calculating the IIAE for each observational data set is a good suggestion and we now do this in the revised manuscript.

Calculating three separate IIAEs means that we cannot show original Fig. 1 in order of increasing IIAE, as the ordering differs slightly according to the observational product used. We instead show the original Fig. 1 plots ordered alphabetically, and mark on the figure which models are ranked as high- and low-performing, based on the IIAE which was calculated separately for each observational product.

For the original Fig. 3, we now show the ice errors for models relative to each observational product separately, which makes very little difference to the conclusions drawn from this figure.

For the original Fig. 8, we now show the SIC difference relative to one observational product. We found that using different observational products made no difference to the conclusions drawn from this figure. | Removed P3, L16; P4, L19 and P5, L7-9 which reference observational means.

We now show ice errors relative to each observational product separately in original Fig. 3. This is [current draft, Fig. 2], with caption: *'Various ice errors for the population of CMIP5 models for all years from 1992 to 2014. Errors are shown relative to (red) the ASI-SSMI satellite observations, (grey) the Bootstrap satellite observations and (light blue) the NASA-Team observations for the months where the maximum and minimum of the seasonal cycle occur of sea ice area (a) or of sea ice extent (b-d) occur. The errors shown are the integrated ice area error (a), the integrated ice extent error (b), the absolute extent error divided by the integrated extent error (c) and the misplacement extent error divided by the integrated extent error (d). Boxplots are as in Fig. 1'*
We have added some discussion of observational variance for this figure in the text: [current draft P7, L31] *'Results are similar using the integrated ice extent error (Fig. 2(b)), although the use of extent rather than area reduces the variation between observational references.'*

We have removed the IIAE values from original Fig. 1. Instead of adding an additional figure, we have added different bounding boxes to the original Fig. 1 to denote well-performing and poorly-performing models, calculated using the IIAE, where the groupings are independent of the observational product used. This is [current draft, Fig. 2] with, *'Models marked with a bold (dashed) bounding box have high-ranked (low-ranked) integrated ice area errors regardless of observational product used'* inserted into the caption. At [current draft P8, L6], added *'An objective way to quantify model-observation* |

| | | | |
|---|---|---|---|
| | | For the original Fig. 9, we show the SIC difference relative to one of the observational products. We choose the observational product that is in the middle of the three products considered when it comes to the fraction of high (90%+) concentration ice. | *disagreement is to use the integrated ice area error, which describes the area of sea ice on which models and observations disagree. Due to observational variability, we calculate this relative to each observational product individually. The variation in observations means that we cannot rank the models in an overall order, but we can construct two groups of well-performing models and of poorly-performing models whose members do not change when using different observational products. These are marked on Fig. 3.'*

We have opted to show the sea ice concentration difference relative to one observational product separately only in the original Fig. 8, to reduce the number of subplots, as we find the results are very similar regardless of observational product. This is [current draft, Fig. 9], where the caption states that the difference is relative to the Bootstrap observations. [current draft P11, L11] now states *'The same qualitative picture is obtained from all three observational products.'* and the preceding discussion of this figure makes clear that is relative to the Bootstrap observations.

Show the sea ice concentration difference relative to one observational product in original Fig. 9 [current draft, Fig. 10]. We select the ASI-SSMI observations, as they lie in between Bootstrap and NASA Team in terms of the fraction of 90%+ concentration ice in DJF (see [current draft, Fig. 4a]). The caption for [current draft, Fig. 10] states that the difference is relative to the ASI-SSMI observations. [current draft, P12, L5] now states *'Taking the difference between the high-concentration ice in each observational and sea ice concentration in the CMIP5 model simulations shows that very few of the models simulate high enough concentrations in this area. Fig. 10 shows the difference between the ASI-SSMI observations and the CMIP5 models; differences are slightly enhanced using Bootstrap and less pronounced when using NASA Team.'* |
| 6 | For other figures (e.g., Fig. 2) it would be good to display all three observation products individually in order to gauge how much of the variability is due to the product, and how much is due to sub-seasonal and interannual fluctuations. | We show observational products individually in the original Fig. 4, to show variability due to the product separately from time variability for the normalized SIC distribution. These show that variability in observational products is more significant than inter-annual fluctuations in the high-concentration regime throughout the year.

We now display all models and all observational products individually in the original Fig. 2. | We have added additional panels to Fig. 2 to show all observational products and models individually, as well as the synthesized populations. This is [current draft, Fig. 1], with caption *'Sea ice area for the months where the maximum (a-b) and minimum (c-d) of the seasonal cycle occur. Populations include data from all years from 1992 to 2014 with boxplots for (a,c) the three observational products (ASI-SSMI, Bootstrap and NASA Team) and all CMIP5 models listed in Table 1 individually; and (b,d) for the ensemble of observational products and the CMIP5 model ensemble. Boxes extend from the lower to upper quartile values of the data with a line at the median. Whiskers show 1.5 of the interquartile range; beyond this data are considered outliers and* |

| | | |
|---|---|---|
| | | *plotted as individual points. The text labels in (b,d) is the p-value calculated from a Kolmogorov-Smirnov test, which represents the confidence that the two populations come from the same distribution'*

This is discussed in the text at [current draft P7, L13] '*Fig. 1 shows sea ice area at the annual maximum and minimum from models and observations. Examining observations and models shown individually (Fig. 1(a) and 1(c)), we find that the interquartile range arising from inter-annual fluctuations over 15 1992-2014 is generally smaller than inter-model differences.*
  *Fig. 1(b) and 1(d) group the models and observations into two populations for comparison. At the annual maximum (Fig 1(b)), the interquartile range from the ensemble of observations for 1992-2014 is contained within the ensemble of models from the same period, with the medians of the two populations in good agreement. There is no clear model tendency compared to observations for the sea ice area maximum. At the minimum (Fig. 1(d)), the interquartile ranges from models and observations show less overlap than the maximum, with the median from the model ensemble significantly lower than the median from the observational ensemble, suggesting a broadly consistent underestimation of sea ice area at the annual minimum by the CMIP5 models. This tendency was also noted by Turner et al. (2013) for sea ice extent. There are outliers, which show an overestimation of sea ice area, notably CSIRO-Mk-3-6-0 and the CESM models. The Kolmogorov-Smirnov test quantitatively shows that both the maximum and minimum sea ice area model-observation comparisons are significantly different, but the difference between models and observations is larger at the summer minimum than at the winter maximum (Fig.s 1(b) and 1(d)).*'* |
| 7 | Second, I sometimes have the impression that the metrics used are overly complicated. The best example is in Fig. 5. There are so many levels of processing that it becomes difficult to understand the meaning of the metrics shown on the figure (see also my comment on Fig. 5 below). For the "binned" diagrams (Figs 4-5-6), why not use only three classes: 0-15%, 15-90%, 90-100%? This would simplify the figures much without removing the useful information that deficiencies in the simulation of SIC happen at the edges of the distribution. In general, I have the impression that I could not replicate the figures if I had the original data. | Since the Discussion paper was submitted, from personal communication with D. Notz we have found that our 'binned fractional sea ice extent' is calculated in exactly the same way as the 'normalized sea ice concentration distribution' in Notz (2015). When we read Notz (2015), we were unsure how he normalized the distribution, but it is normalized by grid cell area, as we have done. We therefore adopt the name 'normalized sea ice concentration' from Notz (2015), instead of 'fractional binned sea ice extent.' We hope that is more intuitive for the reader than our original name for this metric.

We would prefer to keep the bins at 10% spacing as this is consistent with Notz (2015) and gives more of a sense of the whole distribution. | Replaced 'fractional binned sea ice extent' with 'normalized sea ice concentration distribution' throughout text and figures.

Replaced P3, L24-26 with [current draft P5, L7] '*In this study we also consider sea ice concentration distributions, as in Notz (2014) and Ivanova et al. (2016). The sea ice concentration distribution for each model or observational product is calculated by binning grid cells according to their concentration at a 10 %-spacing. The distribution is then normalized by the area of grid cells. We follow the same calculation steps as Notz (2014).*'

See also author response #20 and #30 below. |

| | | Also see response to comments #20 and #30 below. | |
|---|---|---|---|
| 8 | Third, the authors introduce new metrics without real justification and make inconsistent choices. For example, they adapt the Goessling et al. "Integrated ice edge error" by moving from a "extent-like" definition to an "area-like" definition. This is initially a good idea, since area is a more physical measure than extent. However, the decomposition into an "absolute error" and a "misplacement error" is not as straightforward, as I'm showing now: consider a model A with uniform SIC of 80% in a number of grid cells, compared to an observational reference with 70% of ice at the same points. Consider also model B, which has half as much ice as A (40% in the grid cells). Even though ice has not been misplaced, the misplacement error term will increase when going from A to B! (While it won't if the Goessling definition was followed). One of the ideas of the Goessling's approach is precisely to use a threshold at 15% to be able to separate total and misplaced errors. The area-like version of that metric looses that property. | We agree that the misplacement area error does not have a physical meaning and have removed it from the paper. This does not affect any of the conclusions of our work. We also agree that the integrated ice area error is a good physical measure, so will continue to use this.

We believe that presenting the decomposition of an integrated ice extent error into misplacement extent error and absolute extent error using the Goessling et al. (2016) approach is useful here, and show them in the updated version of Fig. 3. | Removed misplacement area error from paper.

Fig. 3 has been adapted to show the misplacement and absolute extent errors, rather than the misplacement and absolute area errors. This is [current draft Fig. 2], with caption: '*Various ice errors for the population of CMIP5 models for all years from 1992 to 2014. Errors are shown relative to (red) the ASI-SSMI satellite observations, (grey) the Bootstrap satellite observations and (light blue) the NASA-Team observations for the months where the maximum and minimum of the seasonal cycle occur of sea ice area (a) or of sea ice extent (b-d) occur. The errors shown are the integrated ice area error (a), the integrated ice extent error (b), the absolute extent error divided by the integrated extent error (c) and the misplacement extent error divided by the integrated extent error (d). Boxplots are as in Fig. 1.*' Discussion of this figure in the text now reads [current draft P7, L38] '*The poorer performance of models at the summer minimum is supported by the integrated ice area error (Fig. 2(a)). The integrated ice area error has a model median value of around 2 million km2 at the sea ice area minimum and around 5.5 million 30 km2 at the sea ice area maximum, despite a much larger amplitude in model mean sea ice area values (around 15 million km2 and 1 million km2 respectively). Results are similar using the integrated ice extent error (Fig. 2(b)), although the use of extent rather than area reduces the variation between observational references. At the winter maximum, across the population of CMIP5 models and different years, we find that the absolute extent error and the misplacement extent error contribute approximately equally to the total integrated ice extent error (Fig. 2(c-d)). At the summer minimum, the integrated ice extent errors for the CMIP5 models have a slightly larger contribution from absolute extent errors than from misplacement area errors (Fig. 2(c-d)).*' This replaces P7, L1-5 in the original draft.

Fig. 8 [current draft Fig. 9] shows the integrated extent errors rather than the integrated area errors, and the text at [current draft P11, L10-14] has been updated accordingly (the conclusions drawn from this figure do not change)

Replaced P4, L1-18 with '*To account for misplacement of sea ice, we use the integrated ice-edge error (IIEE) from Goessling et al. (2016). The IIEE describes the area of grid cells where observations and a model disagree on the presence of sea ice with concentration greater* |

| | | | |
|---|---|---|---|
| | | | *than 15 %. It can be decomposed into the total sea ice extent difference between model and observations (absolute extent error, AEE) and the difference in sea ice extent due to misplacement of sea ice (misplacement extent error, MEE). See Goessling et al. (2016) for further details.* |
| | | | *   Here, we also define an integrated ice area error (IIAE) that describes the area of sea ice on which models and observations disagree. The ice area on which models and observations disagree is likely to be more physically relevant than the area of grid cells on which models and observations disagree. The IIAE is the sum of sea ice area overestimated and underestimated, [equations] '* |
| 9 | I also don't understand why the authors compute per-bin sea ice extent, and not sea ice area in Figs. 4-5-6. Sticking to ice area would be a more natural choice given the adaptation made earlier of the IIAE. | See response to comment #7 above. We believe that consistency with Notz (2015) is preferable, as it allows comparison with his results for the Arctic. | |
| 10 | Finally, I don't fully understand why the authors used daily data in their analyses. This restricts the number of CMIP5 models available, and adds considerable variability to the metrics. By design, CMIP5 models are not supposed to capture synoptic variability in sea ice extent. Using monthly output would partially average out this variability, and would better allow to exhibit the significant biases of the CMIP5 models as the signal-to-noise ratio would increase. | We used daily data as some aspects of marginal ice zone behaviour, which is more variable than compact ice, may be visible in the daily data and not in the monthly data. Sea ice models do simulate synoptic scale variability in sea ice extent, but it is certainly not a focus of development within CMIP5 class models.\n\nWe agree with the reviewer's point that using monthly data rather than daily data has the advantage of allowing us to include more models in our inter-comparison. It also allows us to extend the time period of the analysis. We have switched the paper to using monthly mean data for the CMIP5 models. We found that using monthly data and more models makes no substantial difference to the conclusions drawn from our model-observation comparisons. This strengthens the robustness of our results. | We now conduct all analysis using monthly data. This is reflected in all figure captions and in the text.\n\nWe use CMIP5 monthly output and monthly output from the Bootstrap and NASA Team observations (via the Climate Data Record archive). ASI-SSMI observations are only available as daily output; we will average the sea ice concentration fields each month. We have updated these methodological details in the text. [current draft P3, L22] '*Bootstrap and NASA Team data are available as monthly output; ASI-SSMI data is only available as daily output so the concentration fields are averaged for each month*' |
| 11 | As I'm writing below in a comment, error bars are so large that it is easy to play the devil's advocate and claim that observational references and CMIP5 models are, in the end, not so inconsistent. | We do not show error bars in this paper. We present the data as box plots, indicating the, median, the interquartile range (IQR) and whiskers indicating 1.5 times the lowest datum still within 1.5 IQR of the lower quartile, and the highest datum still within 1.5 IQR. Box plots have the advantage of being non-parametric and can indicate the degree of dispersion and skewness in the data. In our draft manuscript we suggested that where we see lower quartile – upper quartile ranges that do not overlap, the populations being compared are significantly different. | We have included further discussion of statistical tests in the text, inserting at [current draft P5, L17] '*To quantify the agreement between two populations, we use the two-sample Kolmogorov-Smirnov test. This compares the empirical distribution functions of each sample, and takes into account both the location and shape of the distributions. In contrast, a Student's t-test would only examine whether the means of the distributions agree. The p-value obtained from the Kolmogorov-Smirnov test represents the confidence that the two populations come from the same distribution.*' |

| | | | |
|---|---|---|---|
| | | There are a number of statistical tests that could be applied to examine the differences. We applied t-tests, but this is a not a particularly useful test, as it only considers whether two distributions have different means. Using the test of overlapping confidence intervals as the reviewer hints at would require the assumption of normality and only pass if 95% of the data were different in the two samples.  For normalized SIC distributions, which are bounded on the interval [0,1], this is a very difficult test.

After further thought on how to robustly assess inconsistency, we have opted to include the  Kolmogorov–Smirnov test as a robust statistical test of whether the distributions are different. The K-S test is sensitive to differences in both location and shape of the empirical cumulative distribution functions of the two samples.

We found that practically all model – observation comparisons fail the K-S test at the 95% confidence level. However, the p-value obtained from the K-S test, which represents the confidence that the two populations come from the same distribution, is a useful tool to quantify the degree of disagreement. For example, in the original Fig. 5 (a), the p-values from the K-S test comparing models to observations in each bin are lowest for the 90-100% and the 10-20% bins. This allows us to make the objective conclusion that model-observation difference is most significant for low-concentration and high-concentration sea ice.

Finally, it's worth noting that our main aim, and what we find most interesting about the paper, is the identification of consistent behaviours amongst the CMIP5 models, ie. tendencies in one particular direction – rather than any claim of simple model-observation disagreement. | We have removed t-tests on two-population comparisons for the normalized SIC distributions from the figures and text. We now include results from the Kolmogorov–Smirnov test for all two-population comparisons as annotated text on [current draft Fig. 1 and Fig.s 5-8]. These are described in the captions. They are used to quantify our conclusions about the figures in the text:

[current draft P7, L23] *'The Kolmogorov-Smirnov test quantitatively shows that both the maximum and minimum sea ice area model-observation comparisons are significantly different, but the difference between models and observations is larger at the summer minimum than at the winter maximum (Fig.s 1(b) and 1(d)).'*

[current draft P9, L11] *'The two-sample Kolmogorov-Smirnov test can be used to quantify the degree of disagreement between models and observations. The confidence level that the ensemble of observations and ensemble of models were drawn from the same population has the smallest values for the 90 - 100 % and 10-20 % in DJF, the 70 - 90 % concentrations in MAM, the 10-30 % concentrations in JJA and the 80-90 % and 10-20 % concentrations in SON.'* |
| 12 | p. 2, l. 3: The statement that CMIP5 has improved in the simulation of Arctic sea ice compared to CMIP3 is strong, and perhaps too strong. Rosenblum and Eisenman (2016, http://dx.doi.org/10.1175/JCLI-D-16-0391.s1), for example, suggest that this could be due to the omission of volcanic forcings in several CMIP3 models. Also, it is unclear if the CMIP5 to CMIP3 differences actually reflect improvements, or changes in tuning strategy (e.g. Notz, 2015, http://dx.doi.org/10.1098/rsta.2014.0164). Please nuance this statement. | Agree | Replaced *'Advances in Earth system modelling have improved simulation of Arctic sea ice compared to the previous intercomparison project (CMIP3) (Stroeve et al., 2012)'* with *'Advances in Earth system modelling have somewhat improved simulation of Arctic sea ice compared to the previous intercomparison project (CMIP3) (Stroeve et al., 2012), although this may reflect changes in forcings (Rosenblum & Eisenman, 2016) or tuning strategy (Notz, 2015) rather than changes in model physics.'* |
| 13 | p. 2, l. 8:  evaluates –> evaluate. | Agree | p. 2, l. 8:  evaluates –> evaluate. |
| 14 | p. 2, l. 28: … Phase 5 (CMIP5; Taylor et al., 2012). | Agree | p. 2, l. 28: … Phase 5 (CMIP5; Taylor et al., 2012). |

| 15 | p. 2, l. 30: Years prior to 2000 are neglected in the model evaluation. Have the authors still conducted the evaluation on 1979-2000, for which the CMIP5 output and the observations are readily available? Do conclusions of the study hold? Please elaborate. It would be useful to present the results of such analyses in the supplementary material, to test the stability of the metric developed over time. | Our use of daily data meant that were limited (in memory) in our analysis. Use of monthly data means that we can consider a longer time series, and is indeed a more sensible solution. The ASI-SSMI observations begin in 1992, so we now conduct analysis over 1992-2014. Including a longer timeframe of data does not significantly alter the conclusions. | We now conduct analysis using model output and observational data from 1992-2014. This is reflected in figure captions and in the text.

Replaced P2, L30-31 with [current draft P3, L2] '*Due to the availability of observations (see below), we conduct analysis using 1992-2014.*'

[current draft P3, L15] added '*Bootstrap uses cluster analysis of brightness temperatures from two channels (19 GHz and 37 GHz vertical polarization in the Antarctic), applies an ocean mask and is available from 1979 at a resolution of 25 km. NASA Team uses ratios of brightness temperatures (which tends to cancel out physical temperature effects) from three channels (19 GHz in the vertical and horizontal, 37 GHz in the vertical), removes weather contamination based on certain spectral gradient ratios and is available from 1979 at a resolution of 25 km. The ASI algorithm uses the difference in brightness temperatures between horizontal and vertical polarization at 85 GHz, uses lower frequency channels at lower resolution to filter atmospheric effects (which are more apparent at 85 GHz than lower frequencies), and is available from 1992 at a resolution of 12 km. We choose to conduct our analysis over 1992-2014.*'

We confirmed that using 1992-2014 is still appropriate for the NEMO-CICE simulations and is not impacted by model spin up. [current draft P7, L7] states '*The experiments described below, which are performed with the coupled NEMO-CICE model, begin in 1979 and end in 2014. The years before 1992 are neglected to allow for model spin-up. Time series of annual maximum sea ice extent show that this takes around ten years to stabilize.*' |
| 16 | p. 2., l. 31: Could the authors conduct their diagnostics on two members of the same climate model, in order to gauge how much internal variability affects the metrics developed in this study? | Our analysis uses output from 1992-2014, thus sampling 23 years of internal variability. Including additional ensemble members for each model would be equivalent to extending the analysis period. Models with ten ensemble members would give 230 years of output, capturing decadal and centennial variability, as well as inter-annual variability.

To examine the impact of this additional variability on the fraction of low concentration ice, we calculated the normalized concentration distribution for ten ensemble members from three models (CanCM4, EC-EARTH and GFDL-CM2p1). We compared the range of values of the fraction of low concentration ice from (i) the first ensemble member for 1992-2014 and (ii) the range from ten ensemble members for 1992-2014, giving a 230 year record. We | |

| | | found that the 23 year record from one ensemble member sampled a large proportion of the range from the extended 230 year record (77%, 62% and 61% from each model respectively, averaged over each month). The portion of the range from the 230 year record that was not sampled by the 23 year record variability alone tended to be at larger ice fractions, which would somewhat strengthen the model-observation contrast for low-concentration ice.

Our analysis uses a large, multi-model ensemble over multiple years and so we consider a large range of variability. The above analysis confirms that the conclusions of our manuscript would not be changed in any meaningful way by including additional members from some models, and so we have opted not to include the above analysis in the manuscript. | |
|---|---|---|---|
| 17 | p. 3., l. 3: Three observational products were used in the study. Since part of the study describes the differences between sea ice concentration in those products, it would be useful to have a few lines describing differences in algorithms between those products. | Agree | Replace P3, L1-14 with [current draft P3, L6]: *'Passive microwave radiometers deployed on satellites measure the brightness temperature of the Earth's surface, and can be used to infer sea ice concentration. There can be large differences between satellite observations (Bunzel et al., 2016), as various observational data sets apply different algorithms to convert passive-microwave signals into sea ice concentration. As summarized by Ivanova et al. (2014), differences between algorithms are caused by 1. choice of radiometer channels; 2. tie-points, which are the brightness temperatures used to identify different surfaces; 3. sensitivities to changes in physical temperature of the surface; and 4. weather filters, which correct for atmospheric effects falsely indicating the presence of sea ice.*
   *To account for some of this product uncertainty, we use three observational data sets: the Bootstrap algorithm (Comiso, 1986), the NASA Team algorithm (Cavalieri et al., 1984) and the ASI algorithm (Kaleschke et al., 2001; Spreen et al., 2008). We do not consider datasets that merge different observation methodologies. Bootstrap uses cluster analysis of brightness temperatures from two channels (19 GHz and 37 GHz vertical polarization in the Antarctic), applies an ocean mask and is available from 1979 at a resolution of 25 km. NASA Team uses ratios of brightness temperatures (which tends to cancel out physical temperature effects) from three channels (19 GHz in the vertical and horizontal, 37 GHz in the vertical), removes weather contamination based on certain spectral gradient ratios and is available from 1979 at a resolution of 25 km. The ASI algorithm uses the difference in brightness temperatures between horizontal and vertical polarization at 85 GHz, uses lower frequency channels at lower resolution to filter atmospheric effects (which are more apparent at 85* |

| | | | GHz than lower frequencies), and is available from 1992 at a resolution of 12 km. We choose to conduct our analysis over 1992-2014. Bootstrap and NASA Team data are available as monthly output; ASI-SSMI data is only available as daily output so the concentration fields are averaged for each month.

   Differences between the three selected data sets are large: in the Antarctic, the NASA Team algorithm shows the marginal ice zone (defined as the extent of sea ice with concentration between 15 % and 80 %) to extend over 2 million km more than the Bootstrap algorithm (Stroeve et al., 2016).  NASA Team is more sensitive to clouds and wind over open water than the Bootstrap mode (Anderson et al., 2006), while the high-frequency ASI algorithm is also sensitive to such atmospheric effects (Spreen et al., 2008). Bootstrap is more sensitive to physical temperature changes than NASA Team, and may underestimate concentrations at low temperatures, such as near the Antarctic coast (Comiso et al. 1997). For low concentrations, atmospheric effects, which generally lead to falsely increased sea ice, become increasingly important (Anderson et al., 2006). The weather filters/ocean masks used to correct these differ between the different algorithms.' |
|---|---|---|---|
| 18 | p. 3., l. 5-7: To what season does the statement on marginal ice zone area difference apply? I think it's winter, please specify. | It's winter, September and October specifically | Inserted 'in the winter months' in P3, L7 |
| 19 | p. 3., l. 22: "Sea ice area is the sum of the area of all grid cells with more than 15 % sea ice concentration multiplied by the sea ice concentration in each grid cell". Following the conventional NSIDC definition I would have thought that area is just the product of ice concentration by grid cell area, summed over the domain (https://nsidc.org/cryosphere/seaice/data/terminology.html). Why only considering the grid cells with > 15% of ice? | Other locations on the NSIDC website show the definition we used in the original draft. For example see: https://nsidc.org/data/docs/noaa/g02135_seaice_index/#comp_area where it states, 'The monthly average sea ice area calculation is performed through simple pixel-by-pixel arithmetic of multiplying the daily concentration by the size of the grid cell[1], for all grid cells which satisfy the 15 percent threshold and then averaging them together for a month'. Also consider: http://nsidc.org/arcticseaicenews/faq/#area_extent which states "Area takes the percentages of sea ice within data cells and adds them up to report how much of the Arctic is covered by ice; area typically uses a threshold of 15%."  We note that Notz (2015) and Turner et al. (2017) do not use a 15% cutoff, while Ivanova (2016) does use a 15% cutoff.

We conclude that either definition is acceptable, as long as it is stated clearly. We also must be consistent - a 15% cutoff should be used for the IIAE if it used for the SIA. | |

| | | As we use the IIEE and associated MEE and AEE, which by the Goessling et al. (2016) definition use a 15% cutoff, we think that the most consistent approach is to use the 15% cutoff. | |
|---|---|---|---|
| 20 | p. 3., l. 25: Why is the bin (0-10%) not in the list? | As shown in Ivanova et al. (2016) (Fig. 2d), the CMIP5 multi-model mean and the NASA Team observations have a high fraction of ice below 10% sea ice concentration in the summer. We find that the fraction of <10%-concentration ice varies in the models from 0.005 to 1.0 (when models are essentially ice-free) in the summer. It consists of up to around a third of the ice in other seasons for some models.

Including these very low concentrations therefore heavily skews the normalized SIC distribution towards very low concentrations. It obscures behaviour at higher concentrations. Our aim is to look for consistent model behaviour; with such variance between different models and between different observations at very low concentrations, it's difficult to conclude anything about model tendencies. | We have added 0.1-10% sea ice concentration to Fig. 1 [current draft Fig. 4].

We have added the following text [current draft P8, L15] *'As shown by Ivanova et al. (2016), the CMIP5 multi-model mean and the NASA Team observations have a high fraction of ice below 10 % sea ice concentration in the summer. We find that the fraction of 0.001 - 10 %-concentration ice varies in the models from 0.005 to 1.0 (when models are essentially ice-free) in the summer (Fig. 3). It consists of up to around a third of the ice in other seasons for some models. Including these very low concentrations heavily skews the normalized sea ice concentration distribution towards low concentrations and it obscures behaviour at higher concentrations. Our aim is to look for consistent model behaviour, so to avoid the large variance between different models and between different observations at very low concentrations, we only consider sea ice concentrations above 10 %.'* |
| 21 | p. 4., l. 20: "A disadvantage of the IIAE is that it does not take into account the observational range, using only the observational mean as the 'true' state". That's not really a disadvantage of the metric, but rather a methodological issue. The authors could repeat the IIAE taking successively the three observations as references. They would obtain three IIAE's, which would give a sense on the uncertainty associated to the products. Why didn't the authors go this way? See also my first comment on methodology. | See response to comment #5. | See author response and action #5. |
| 22 | p. 6, l. 17. "[Ocean-sea ice] Model output is analysed on its native grid". Does that mean that the observations were then interpolated onto the NEMO-CICE grid? At page 9, line 18 and in Fig. 8 the NEMO-CICE model is evaluated using the IIAE metric, this means that at some point an interpolation must take place, correct? The model output and the observational reference need to be on the same grid for Eqs (2) and (3) to be evaluated. Why didn't the authors interpolate the NEMO-CICE output on the | Following comments by other reviewers (anonymous review #2 and C Holmes), as well discussion with others in the community, we have thought more carefully about the interpolation. We believe it is preferable to avoid interpolation as much as possible.

This is possible for sea ice area and the normalized sea ice concentration distributions, which we thus propose to calculate on original grids. We found that model-observation differences in the normalized sea ice concentration distributions at low concentrations are slightly reduced when conducting the analysis on the native grids. We now conduct this analysis on native grids | The analysis has been altered so that sea ice area and the normalized sea ice concentration distributions are calculated on original grids. Integrated ice errors for CMIP5 models and sea ice concentration differences in Fig. 9 [current draft, Fig. 10] are calculated by interpolating all data onto a regular 1 degree grid. In order to calculate sea ice concentration differences between the NEMO-CICE model and the Bootstrap observations in Fig. 8 [current draft, Fig. 9], we interpolate the Bootstrap observations onto the NEMO-CICE grid. This is stated in the figure caption.

We have removed P3, L19-21. |

| | | |
|---|---|---|
| same target grid as all CMIP5 models, to ensure consistent analysis? | and state that the normalized sea ice concentration distributions show some sensitivity to grid interpolation in the Metrics subsection.

Integrated ice errors and sea ice concentration differences between models and observations must be calculated on the same grid. In this case, we interpolate onto a common grid using Climate Data Operators bilinear interpolation function and state this in the manuscript. For CMIP5-observation comparisons, we interpolate onto a regular 1 degree grid. For NEMO-CICE -observation comparisons, we interpolate the observations onto the NEMO-CICE grid.

Bilinear interpolation may cause some smoothing of the ice edge. We find that using bilinear and nearest-neighbour interpolation give the same results in the updated Fig. 9 (original Fig. 8), in updated Fig. 10 (original Fig. 9) and in the grouping of models into high- and low-performing using the IIAE shown in the updated Fig. 3. | [current draft P5, L21] now states '*We found that sea ice concentration distributions show some sensitivity to grid interpolation and therefore calculate sea ice concentration distributions, as well as sea ice area, on the native model and observation grids. The integrated ice errors and differences in sea ice concentration fields between models and observations must be calculated on the same grid. In these cases, we follow Turner et al. (2013) and interpolate model output and observational data on to a common grid using the bilinear remapping function from CDO (2015). For the CMIP5 integrated ice errors and sea ice concentration differences, we choose a 1o x 1o regular grid, which is a resolution equal to or higher than 20 of the 40 models and lower than all observations. We consider it to be an acceptable midpoint given the large range of model resolutions.*'

Note this change, together with the extended time period and the use of monthly rather than daily data, affects Fig.s 4 and 5, and so the discussion of these figures in the text differs to the original draft. The big picture conclusions, however, remain the same.

[current draft P7, L9] now states '*Model output from the NEMO-CICE experiments is analysed on its native grid (1o tripolar). Comparisons between NEMO-CICE simulations and observations (integrated ice errors and sea ice concentration differences) are computed by interpolating observations on to the same 1o tripolar grid using CDO (2015)*' |
| 23 | p. 6, l. 20 and Fig. 2. "Sea ice area at the annual minimum is consistently biased low". Here I'm playing the devil's advocate. The blue boxes in Fig. 2 display the distribution of the three observational references, which are three times the same climate realization plus noise due to the retrieval algorithm. Hence these blue boxes embody time-variability and product variability. By contrast, the green boxes in Fig. 2 contain time-variability internal variability, and model error. So, the whole question is whether these observational references are incompatible in a statistical sense with the models. Put differently, could the observations be the (N + 1)th CMIP model? Judging from Fig. 2b, the observations lie in the range [1.5 * IQ_75%, IQ_75%] and they could be one of the CMIP models. Or couldn't they? A more quantitative test would be welcome. | See response to comment #11. In summary, to more robustly assess inconsistency of distributions we include the Kolmogorov–Smirnov test as a robust statistical test of whether the distributions are different. The K-S test is sensitive to differences in both location and shape of the empirical cumulative distribution functions of the two samples.

Also see response to comment #6. Separating boxplots now allows us to discuss the contributions of model/observational product variability and time variability. | See author intended action #11.

See author intended action #40.

See author intended action #6 |

| 24 | p. 7, l. 5-8. The sentence "Such total errors..." is unclear. First, there are many more possible reasons than just ocean/atmosphere temperature biases to explain differences in total area (which is captured by the AAE). Second, it's not clear why looking at the bias per concentration bin would help isolate the role of the sea ice component. | The case we wanted to make is that sea ice extent will be different in a model with a normal ocean compared to one that is biased 1 degree warmer everywhere. However, both models could still stimulate an appropriate normalized sea ice concentration distribution. Therefore we would argue that a normalized sea ice concentration distribution depends less on overall ocean/atmosphere temperature biases than sea ice extent. | Replaced P7, L5-8 with [current draft P8, L11]: '*We now consider sea ice concentration distributions from observations and models, which provide a more detailed assessment than hemisphere-integrated measures. A normalized sea ice concentration distribution may help isolate the role of the sea ice component, as models with a constant temperature bias in the atmosphere or ocean, resulting in a biased sea ice area or extent, may still simulate the relative fraction of different concentration regimes successfully.*' |
| --- | --- | --- | --- |
| 25 | p. 7, l. 10. Notz (2014) "uses" or "used" but not "use". | Agree | Corrected to Notz (2014) "uses" |
| 26 | p. 7, l. 22-35. The two paragraphs deliver somewhat contradictory messages. The first one finishes by "that the sea ice components of CMIP5 models are somewhat successful at simulating the distribution of sea ice concentration" while the second paragraph says "large discrepancies between models and observations in the highest and especially the lowest concentration bins". This would need better rephrasing, saying for example in the first paragraph that the "big pictures" are consistent but that this is mostly thanks to cancellation of errors, as explained in the second paragraph. | We agree with the reviewer and have reworded these paragraphs accordingly. The use of the K-S test to quantify the differences between the two populations is useful here. As explained above, the p-value from the K-S test, which represents the confidence that the two populations come from the same distribution, is highest for the 90-100% and 10-20% bins in DJF. | These paragraphs have been reworded to [current draft P9, L3]: '*Differences between the sea ice concentration distribution from models and observations, including inter-annual and subseasonal information, (Fig. 5) are less distinct than between observational products themselves. This reflects the large range in both models and observations due to systematic uncertainties. The overall decomposition from the CMIP5 models, with a large fraction of compact ice cover and smaller fractions of lower concentrations is somewhat in agreement with observations.*' followed by discussion of where they disagree. |
| 27 | p. 10, l. 4-6 and Fig. 9. The authors conclude that the systematic underestimation of highly concentrated ice in the Weddell Sea is related to melt or break-up processes. Why is the possibility of a systematic misrepresentation of dynamics ruled out? It could be that all models have deficiencies in capturing the Weddell gyre dynamics. It could be that models are neutral to divergent while observations are in convergent motion. I haven't tested this hypothesis myself, but I don't have enough information from the results of the paper to rule out properly this alternative hypothesis. An exploration of how the models simulate Antarctic ice drift could be helpful in that respect. | We agree that dynamic processes are a possible cause of the underestimation of highly concentrated ice. The findings of Lecomte et al. (2016) are particularly relevant here. They suggest that models with high ice drift speeds in coastal areas simulate a faster sea ice retreat. These high drift speeds may be influenced by sea ice rheology as well as wind speeds. | Added '*including misrepresentation of sea ice dynamics.*' to P10, L6 [current draft P12, L10].

Replaced P10, L12 with '*Other CMIP5 models may simulate high drift speeds due to winds or sea ice rheology, which Lecomte et al. (2016) found correlated with a faster sea ice retreat.*' [current draft P12, L16]

Also see author action #3 |
| 28 | p. 10, l. 29: 2. Three observational products are considered in this study. While I appreciate this effort, it looks sometimes like the authors assume that observational errors are random and that the mean of all three products is representative of the truth. It could be that the three observational | As discussed above, we now avoid use of any observational mean in this study as suggested.

We agree that including different observational products will give some estimation of error arising from differences in processing satellite data, but there is still the possibility of systematic errors | Added the following paragraph [current draft P3, L33]:
*Besides structural uncertainty in observational algorithms, systematic biases common to all three products are possible. Lack of validation data (Ivanova et al., 2014) mean it is difficult to quantify this, but accuracy is understood to be lower in the presence of melt ponds or other surface melt effects (Ivanova et al., 2014), which may act to lower* |

products have a systematic bias with respect to the truth, which could explain model-obs mismatch on top of model error.

The authors don't seem to explore this possibility in the assessment. For example it is known that most algorithms underestimate sea ice concentration as ice becomes very thin. Could this explain the model-obs differences, in particular differences in binned sea ice extent? It is also known that wet snow has a brightness temperature that makes sea ice concentration retrievals higher than they should be. Could this have an impact? More discussion on observational systematic errors would be welcome, in order to place the CMIP results in perspective.

common to all three observational products. In the Discussion, we did briefly discuss systematic errors in the observations:
*'Accounting for the observational range, we find that models overestimate the extent of low-concentration sea ice throughout the year, while underestimating the extent of high-concentration sea ice in summer. This common behaviour across diverse models with varying physics is a result not previously highlighted and warrants further attention. We note that using the observational range as an uncertainty estimate neglects biases that are common to the three different satellite observations. As mentioned above, satellite observations of sea ice are most uncertain in summer. However, we see the bias in low concentration ice from CMIP5 models throughout the year, and observed summer high-concentration ice is unlikely to be affected by the melt processes that complicate satellite retrievals. The suggestion that the NASA Team algorithm overestimates low-concentration ice (Steffen & Schweigher, 1991) would further strengthen the contrast between models and observations in this regime.'*

We agree that this is worthy of more comprehensive discussion within the paper and have expanded on this discussion point (see author actions).

*retrieved concentrations; large fractions of thin ice (Ivanova et al., 2015); and stormy conditions near low concentrations (Anderson et al., 2006). Transitions between ice type can cause differences in emissivity (Grenfell and Comiso, 1986), but because models do not simulate ice types such as grease ice, this issue should not impact model-observation comparisons.*

Added the following to the discussion [current draft P12, L33]:
*As mentioned above, sea ice concentrations are considered to be most uncertain during melt conditions, for large fractions of thin ice and at low concentrations during storms. In the context of the results from the model-observation comparison for normalized sea ice concentration distributions, we suggest that the impact of uncertainty of melt conditions is limited as the high bias in low-concentration ice from CMIP5 models is visible throughout the year. The low bias in high-concentration ice during the melt season would be strengthened if observations were underestimating ice concentrations in this season. Inclusion of both NASA Team and Bootstrap algorithms, with the former tending to cancel out physical temperature effects, will sample some of this uncertainty.*
*The underestimation of sea ice concentrations in areas of thin ice (<35 cm) (Ivanova et al., 2015) may cause a bias at any concentration in the observed normalized sea ice concentration distribution from observations, with the possibility of a positive bias in the very lowest concentrations. Stormy conditions near the ice edge lead to false sea ice concentrations near the ice edge; weather filters may accurately remove these, leave them uncorrected (Anderson et al., 2006) or erroneously remove real sea ice. The latter may underestimate low concentrations. Spreen et al. (2008) suggest the filter method used in ASI-SSMI observations may result in a positive bias in the marginal ice zone, and Steffen & Schweiger (1991) found that the NASA Team algorithm overestimates low-concentration ice when compared to Landsat imagery. Considering all this evidence we suggest that the magnitude or sign of any systematic biases in satellite radiometer observations is unclear when comparing with climate models. This is particularly true for low concentrations. Here the use of different approaches to weather filters within the different algorithms may assist in sampling observational uncertainty. Development of sea ice satellite emulators, which use climate model output to calculate brightness temperatures (eg. Tonboe et al., 2011), may help to reduce uncertainty when comparing models to observations in the future.*

| 29 | Fig. 1. Interestingly, it is possible that two CMIP5 models with similar IIAEs (e.g., MIROC5 (4.6 Mkm2) and FGOALS-g2 (4.85 Mkm2)) have drastically | Is there are reason to favour over-estimation or under-estimation? We don't see that one is better than the other, so we don't see the lack of distinction between the two in the IIAE to be an issue. | [current draft P5, L5] added *'The integrated ice errors penalize under-estimation and over-estimation of sea ice equally.'* |

| | | | |
|---|---|---|---|
| | different sea ice concentration patterns (one with not enough ice and one with way too much ice; panels ac and ad). In the same vein, two models with similar patterns (e.g. CMCC-CMS and HadGEM2-CC) may have very different IIAEs. This is because of the definition of IIAE which penalizes over- and underestimation in the same way. The authors should comment on that aspect (which I see as a weakness of that metric). Although there is no definition of what a "good" metric is, we could expect that it satisfies properties of continuity in some sense: two models close to each other should have similar metric values. | | |
| 30 | Fig. 5. This is one example where I would have difficulties in reproducing the result. If I follow correctly, from Fig. 5 caption and from the text: (1) Grid cells are binned according to their concentration (2) The total sea ice extent is computed in the three observational references, in each CMIP5 model, for each day of each year. (3) The extent per bin is normalized by the total extent (for Fig. 5 panels a-d) (4) The normalized extent of each CMIP5 model is compared to the normalized extent of each observation (or to the mean of them?) to give a fractional deviation. I doubt that, out of 10 readers, more than one can replicate Fig. 5 exactly. The authors should detail their approach in a supplementary material, or simplify the metrics. | See response to comment #7 above. Our calculation of fractional binned sea ice extent/normalized sea ice concentration distribution is the same as Notz (2015). We now explicitly state the steps involved in the Methods section.

We agree that Figs (e-h) are confusing and have decided to take a different approach in our revision. The aim of Figs(e-h) in the first draft was to show the biases independent of scale. We have removed (e-h) from the manuscript and added an extra figure [current draft Fig. 6], which shows the fraction of 10-20% ice for each month and more clearly highlights the biases at low concentrations by using a more appropriate scale. | See author action #7 above.

We have removed Figs(e-h). The aim of Figs(e-h) in the first draft was to show the biases independent of scale. To achieve this in the current draft, we have added an extra figure [current draft Fig. 6], which shows the fraction of 10-20% ice for each month and more clearly highlights the biases at low concentrations by using a more appropriate scale. |
| 31 | Page 2, Line 30: Why limit the analysis to 2000-2014, when longer observational and model timeseries are available? Longer time series would make the analysis more robust. | This comment and the comment below are connected – our use of daily data meant that we were limited (in memory) in our analysis. Use of monthly data means that we can consider a longer time series, and this is indeed a more sensible solution. The ASI-SSMI observations begin in 1992, so we now conduct analysis over 1992-2014. | See author intended action #15 |

| | | See author response to comment #15 above | |
|---|---|---|---|
| 32 | Page 2, line 32: Why is daily sea ice concentration used here? This should be explained, as many more models provide monthly than daily output, and it looks like the authors proceed to average the daily output to seasonal averages. | See author response to comment #10 above | See author intended action #10 above |
| 33 | Page 3, Line 15-17: While I agree that one needs to consider the observational uncertainty, I am not convinced that averaging several products is the best way to do that.

First of all, they could all have consistent biases, and hence their range still would not account for the observational uncertainty. Secondly, one of them might be a lot better than the others, and so the combined data might be further from the truth than the best one. So while I am not suggesting that the authors perform an evaluation of the three observations, which is best done by the creators of these data sets, I would encourage the authors to add a sentence or two here to highlight the potential shortcomings of this approach they are using. | We agree with your points. Please see author response to comment #5 above regarding averaging observational products.

We do combine the three sets of observations in original Figs 5-7 for the concentration distributions. It is not clear to us from the literature that any of the three datasets is better than the others. Evaluation of the products is indeed beyond the scope of this manuscript. Further, Ivanova (2014) states that 'we cannot establish an absolute ranking of the performance of the algorithms because of the lack of good validation data,' and recommends constructing an ensemble of different observational products. We have added some justification of our approach to the manuscript. | See author action #5

P3, L15: Replaced this paragraph with: '*In this study, for some of the analysis we consider the three observational data sets individually. In order to compare the sea ice concentration distribution from the set of models against observations, we create an ensemble of the ASI, Bootstrap and NASA Team observational products. Combining the observational products in this way does have limitations, as different algorithms are likely to perform better for certain sea ice conditions and seasons. However, it is not clear from the literature where exactly the strengths of the various algorithms lie, and evaluation of the different algorithms is beyond the scope of this manuscript. The difficulty in ranking various observational algorithms is noted by Ivanova et al. (2014), due to a lack of validation data. They recommend constructing an ensemble of different observational products.*' [current draft P4, L5] |
| 34 | Page 3, Line 19: Why was the sea ice output re-gridded, rather than analyzed on the model grids? This can introduce additional errors that have nothing to do with the physics of the model. So there needs to be a good reason to re-grid the model output, otherwise the analysis should be re-done on the original grids. And if the authors have a good reason to do the re-gridding, please include information on how exactly the regridding was done, so it can be replicated by others. | See author response to comment #22 above | See author action #22 above |
| 35 | Page 3, Line 27: Why are concentrations below 10% not included? Others included them, so please explain why you would not. For loose sea ice, wouldn't it be important to look at below 10%? | See author response to comment #20 above | See author action #20 above |
| 36 | Page 8, line 15, Table 1: Since the authors have the information on whether and how lateral melt is included in the CMIP5 models, do they find any difference between models that include it or not? That would provide an important argument for the | See author response to comment #2 above | See author action #2 above |

| | | | |
|---|---|---|---|
| | hypothesis of the authors that the too loose sea ice concentration is a result of deficiencies in lateral melt. | | |
| 37 | Page 10, line 24-28: Please remove this entire paragraph. It is pure speculation what modeling centers look at during model development, and this speculation does not add anything to the arguments or results presented in the paper. | Agree | We have removed P10, L24-28 |
| 38 | Page 10, Line 29: The observational range is not necessarily fully counted for, as discussed earlier. This should be reflected here. | See author response to comment #28 above | See author action #28 above  Replace P10, L29 *'Accounting for the observational range'* with *'Accounting for the range in three observational products'* [current draft P12, L29] |
| 39 | Figure 1 and throughout:  I'm unsure of the relevance of DJF; although the traditional meteorological austral summer season, it's arguably not particularly relevant for sea ice, particularly since you do not link analyses to atmospheric variables.  However I recognise it's not obvious what the best season would be.   I'd suggest showing the minimum or maximum, or if to use DJF, please give some justification (in particular why summer not winter) and mention any sensitivities to season, if found. | Our interest in summer stems from Fig. 2 (SIA), where we looked at sea ice area from models versus observations and concluded that the minimum showed more disagreement with observations than the maximum. This is further supported by the analysis in Fig.3 (IIAE). We therefore chose to examine the months leading up to the summer minimum (DJF) in more detail.  We chose to show DJF, MAM, JJA, SON in original Fig. 5 as we wanted to include data from all months.  The normalized SIC distribution for DJF shows largest differences from observations at the high and low ends of the distribution. We now show the fraction of 10-20% ice for each month in an additional figure and have added further discussion of the seasonality of the normalized sea ice concentration distribution.  We have switched some analysis to using the annual minimum rather than DJF and justified our choice of seasons in the text (see right). | We have switched the order of the initial figures to make clear that our interest in summer stems from the poorer performance of models at the summer minimum than at the winter maximum for sea ice area (original Fig. 2). This is now Fig. 1. Integrated ice errors at the summer minimum and winter maximum are shown in Fig. 2 (original Fig. 3). We stress *'The poorer performance of models at the summer minimum'* by beginning the second paragraph with this phrase [current draft P7, L27].  We now show the sea ice concentration fields at the summer minimum (with month of minimum computed for each model individually) instead of the DJF mean in Fig. 3 (original Fig. 1). This fits better with the narrative of the first two paragraphs in the results section. At [current draft P8, L3], we state that *'The large inter-model variability in extent and area at the summer minimum can be seen in Fig. 3, where the sea ice concentration fields show diverse behaviour. Variability between observational products is smaller than inter-model differences, but observational differences are visible, particularly at low concentrations.'* |

We would prefer to show data from all months in Figures 4 and 5. [current draft P8, L21] states '*We present all months grouped by meteorological season (December - February (DJF), March-May (MAM), June - July (JJA) and September - November (SON)). This choice separates the melt season (September-February) from the freezing season (March - August), while limiting the number of months included in each season.*'

We have added an extra figure [current draft Fig. 6], which shows the fraction of 10-20% ice for each month and added further discussion of the seasonality of the normalized sea ice concentration distribution. [current draft P9, L17] '*Unlike the other CMIP5 model tendencies, the overestimation of 10-20% ice occurs in every month (Fig. 6), with the CMIP5 model median always outside the interquartile range of the observations*'

[current draft] Fig.s 7 and 8 also show the fraction of 10-20% ice for each month to highlight the seasonality of results. The discussion of Fig. 8 (which was most similar to original Fig. 7] now reads '*Fig. 8 shows the fraction of 10-20% sea ice concentration from observations, the standard NEMO-CICE model and the model with reduced floe size. The standard model has very strong overestimation of low-concentration ice through December to March compared to observations. Impact of reduced floe size on the distribution is limited, with the exception of February where there is a very strong reduction in the fraction of 10-20% concentration ice, bringing it into better agreement with observations.*' [current draft P11, L1]

We have added some discussion of the seasonality of lateral melt. [current draft P11, L6] '*The enhanced lateral melt achieved by reducing floe size results in statistically significant reductions in sea ice concentration relative to the standard model in DJF (Fig. 9b)). December, January and February stand out from the other months in having particularly high total lateral melt rates.*' That DJF has the greatest lateral melt justifies showing DJF in [current draft] Fig. 9 (original draft Fig. 8). To clarify that that there are statistically significant reductions in SIC in DJF relative to the standard model for the small floes simulation; and no statistically significant reductions in SIC in DJF relative to the standard model for the large floes simulation; we have added additional panels to Fig. 9. The caption now reads: '*Sea ice concentration averaged over DJF 1992-2014 for (a) the standard model simulation with a floe diameter of 300 m; (b) a model simulation with a floe diameter of 1 m (small floes) minus (a); (c) a model simulation with a floe diameter of 10,000 m (large floes) minus (a). (d-f) show simulation minus observed Bootstrap sea ice concentration,*

| | | |
|---|---|---|
| | | *where the latter has been interpolated on to the model grid for (d) the standard model simulation, (e) the small floes simulation and (d) the large floes simulation. In (b-f), differences are shown only if they are statistically different according to a Student's t-test over 1992-2014 (p<5 %). Labels on (d-f) show the integrated ice extent error, absolute extent error and misplacement extent error in million km2.'*

 We continue to show DIF in [current draft Fig. 10] (original draft Fig. 9) as this is '*the season where the bias is apparent in Fig. 5'* [current draft P12, L3]. |
| 40 | Figure 2: This combines spread in information from different years and from different observational data sets. In particular the conclusion in the main text that there is 'no clear bias' at maximum is a little confusing as climatologies are not shown (I would think of 'biases' as referring to climatologies), and the discussion of this figure in the text is very brief; half a sentence or so. Also panel a) appears to be missing outliers? I suggest separating the panels into separate boxplots, particularly for observations, clarifying the multi-model vs multi-yr distinction (if possible), checking the figure caption, and expanding the discussion of this figure a little (it need not be much) | Fig. 2 shows that the interquartile range of the CMIP5 models overlaps that of the observations for the sea ice area maximum, but it does not for the sea ice area minimum. We conclude that there is a tendency for models to underestimate the sea ice area minimum, but there is not such a significant tendency at the sea ice area maximum. This conclusion can be quantified using the K-S test, as discussed above.

 We have separated out the boxplots - see author response #6.

 The data in Fig. 2a does not have outliers when the whiskers are set to 1.5 of the interquartile range. | We have added panels to this figure with separate boxplots and expanded discussion of this figure - see author action #6 |
| 41 | Methodological note: Please say how the regridding is performed (the method and the package used). I have certainly seen cases myself and at meetings (sorry I cannot bring a citation!) which suggest that it can affect results particularly since you are concerned with distributions rather than aggregate measures. Such methodological details are rarely stated in papers about CMIP5, but for reproducibility it they should be!" | The comment on the impact of regridding is correct. See author response to comment #22 above | See author action #22 |

[revised manuscript text omitted]

forcings, end in 2005. To obtain a more contemporary overview, we  also consider the first nine

years of projection experiments from the midrange mitigation emission scenario (RCP4.5)

 Due to the availability of observations (see below), we conduct analysis using 1992-2014. We select the first ensemble member

for all models that provide  monthly sea ice concentration for both the historical and RCP4.5 experiments, resulting in a

5   set of  40 models (see Table 1).

**2.2   Observations**

 Passive microwave radiometers

deployed on satellites measure the brightness temperature of the Earth's surface, and can be used to infer sea ice concentration.

10   There can be large differences between satellite observations (Bunzel et al., 2016), as various observational data sets apply

different algorithms to convert passive-microwave signals into sea ice concentration. As summarized by Ivanova et al. (2014),

differences between algorithms are caused by (1) choice of radiometer channels; (2) tie-points, which are the brightness

temperatures used to identify different surfaces; (3) sensitivities to changes in physical temperature of the surface; and (4)

weather filters, which correct for atmospheric effects falsely indicating the presence of sea ice.

15

To account for some of this product uncertainty, we use three observational data sets: the Bootstrap algorithm (Comiso, 1986),

the NASA Team algorithm (Cavalieri et al., 1984) and the ASI algorithm (Kaleschke et al., 2001; Kaleschke et al., 2008). We

do not consider datasets that merge different observation methodologies. Bootstrap uses cluster analysis of brightness temperatures

20   from two channels (19 GHz and 37 GHz vertical polarization in the Antarctic), applies an ocean mask and is available from

1979 at a resolution of 25 km. NASA Team uses ratios of brightness temperatures (which tends to cancel out physical

temperature effects) from three channels (19 GHz in the vertical and horizontal, 37 GHz in the vertical), removes weather

contamination based on certain spectral gradient ratios and is available from 1979 at a resolution of 25 km. The ASI algorithm

uses the difference in brightness temperatures between horizontal and vertical polarization at 85 GHz, uses lower frequency

25   channels at lower resolution to filter atmospheric effects (which are more apparent at 85 GHz than lower frequencies), and is

available from 1992 at a resolution of 12 km. We choose to conduct our analysis over 1992-2014. Bootstrap and NASA Team

data are available as monthly output; ASI-SSMI data is only available as daily output so the concentration fields are averaged

for each month.

30

 Differences between the three selected data sets are large: in the Antarctic, the NASA Team algorithm shows the marginal ice zone (defined as the extent of sea ice with concentration between 15 % and 80 %) to extend over 2 million km more than the Bootstrap algorithm in the winter months (Stroeve et al., 2016). NASA Team is more sensitive to clouds and wind over open water than the Bootstrap mode (Andersen et al., 2006), while the high-frequency ASI algorithm is also sensitive to such atmospheric effects (Spreen et al., 2008). Bootstrap is more sensitive to physical temperature changes than NASA Team, and may underestimate concentrations at low temperatures, such as near the Antarctic coast (Comiso et al., 1997). For low concentrations, atmospheric effects, which generally lead to falsely increased sea ice, become increasingly important (Andersen et al., 2006). The weather filters/ocean masks used to correct these differ between the different algorithms.

Besides structural uncertainty in observational algorithms, systematic biases common to all three products are possible. Lack of validation data (Ivanova et al., 2014) mean it is difficult to quantify this, but accuracy is understood to be lower in the presence of melt ponds or other surface melt effects (Ivanova et al., 2014), which may act to lower retrieved concentrations; large fractions of thin ice (Ivanova et al., 2015); and stormy conditions near low concentrations (Andersen et al., 2006). Transitions between ice type can cause differences in emissivity (Grenfell and Comiso, 1986), but because models do not simulate ice types such as grease ice, this issue should not impact model-observation comparisons.

 In this study, for some of the analysis we consider the three observational data sets individually. In order to compare the sea ice concentration distribution from the set of models against observations, we create an ensemble of the ASI-SSMI, Bootstrap and NASA Team observational products. Combining the observational products in this way does have limitations, as different algorithms are likely to perform better for certain sea ice conditions and seasons. However, it is not clear from the literature where exactly the strengths of the various algorithms lie, and evaluation of the different algorithms is beyond the scope of this manuscript. The difficulty in ranking various observational algorithms is noted by Ivanova et al. (2014), due to a lack of validation data. They recommend constructing an ensemble of different observational products.

**2.3 Metrics**

Following convention, sea ice extent is defined as the area of all grid cells with more than 15 % sea ice concentration. Sea ice area is the sum of the area of all grid cells with more than 15 % sea ice concentration multiplied by the sea ice concentration in each grid cell.

To ~~avoid dependence on the total sea ice extent in each model, which is likely determined by ocean surface temperatures, we divide binned sea ice extent by the total extent of ice with concentrations greater than 10 %, giving a 'fractional binned sea ice extent.' This metric allows us to examine observed and modelled behaviour in different sea ice concentration regimes. It does not penalise models whose spatial distribution of sea ice disagrees with observations, but it does allow us to quantify disagreement with observations on sea ice concentration values while accounting for the observational range.~~

 account for misplacement of sea ice, we  use the integrated ice-edge error (IIEE) from Goessling et al. (2016). The IIEE describes the area of grid cells where observations and a model disagree on the presence of sea ice  with concentration greater than 15 %. It can be decomposed into the total sea ice extent difference between model and observations (absolute extent error, AEE) and the difference in sea ice extent due to misplacement of sea ice (misplacement extent error, MEE). See Goessling et al. (2016) for further details.

Here, we also define an integrated ice area error (IIAE) that describes the area of sea ice on which models and observations disagree.  The ice area on which models and observations disagree is likely to be more physically relevant than the area of grid cells on which models and observations disagree. The IIAE is the sum of sea ice area overestimated and underestimated,

$$\text{IIAE} = O + U \tag{1}$$

with

$$O = \int_A \max\left(c_m - c_o, 0\right) dA \tag{2}$$

and

$$U = \int_A \max\left(c_o - c_m, 0\right) dA \tag{3}$$

where $A$ is the area of interest, $c_m$ is the simulated sea ice concentration and $c_o$ is the observed sea ice concentration.

$$\text{IIAE} = \text{AAE} + \text{MAE}$$

where

$$AAE = |O - U|$$

and

$$MAE = 2 \cdot \min(O, U)$$

5

The integrated ice errors are useful as they quantify error in integrated sea ice concentration values as well as quantifying error caused by sea ice appearing in different grid cells than the observations. This is in contrast to difference in sea ice area, which accounts only for error in integrated sea ice concentration values, and difference in sea ice extent, which accounts only

10 for error in the area of grid cells that have ice. The integrated ice errors penalize under-estimation and over-estimation of sea ice equally.

In this study we also consider sea ice concentration distributions, as in Notz (2014) and Ivanova et al. (2016). The sea ice concentration distribution for each model or observational product is calculated by binning grid cells according to their concentration at a 10 %-spacing. The distribution is then normalized by the area of grid cells. We follow the same calculation

15 steps as Notz (2014). This metric allows us to examine observed and modelled behaviour in different sea ice concentration regimes. It does not penalise models whose spatial distribution of sea ice disagrees with observations, but it does allow us to quantify disagreement with observations on sea ice concentration values while accounting for the observational range.

To look for behaviours which are consistent across all CMIP5 models, we compare the population of all models for the  years 1992-2014 against the population of all observations for the same period. Including all

20 models means that the range is large when models show opposite tendencies; using a multi-model mean would average out this information. Including all  months in each season for all years during analysis captures sub-seasonal and inter-annual variability.

 To quantify the agreement between two

25 populations, we use the  two-sample Kolmogorov-Smirnov test. This compares the empirical distribution functions of each sample, and takes into account both the location and shape of the distributions. In contrast, a Student's t-test would only examine whether the means of the distributions agree. The p-value obtained from the Kolmogorov-Smirnov test represents the confidence that the two populations come from the same distribution.

30 We found that sea ice concentration distributions show some sensitivity to grid interpolation and therefore calculate sea ice concentration distributions, as well as sea ice area, on the native model and observation grids. The integrated ice errors and

differences in sea ice concentration fields between models and observations must be calculated on the same grid. In these cases, we follow Turner et al. (2013) and interpolate model output and observational data on to a common grid using the bilinear remapping function from Climate Data Operators (CDO 2015). For the CMIP5 integrated ice errors and sea ice concentration differences, we choose a $1^o \times 1^o$ regular grid, which is a resolution equal to or higher than 20 of the 40 models and lower than all observations. We consider it to be an acceptable midpoint given the large range of model resolutions.

**2.4 Coupled ocean-sea ice model**

To understand the impact of model parametrizations for sea ice thermodynamics, we carry out perturbed parameter simulations using a coupled ocean — sea ice model. This consists of the ocean model NEMO and the sea ice model CICE5.1 forced with the atmospheric reanalysis JRA-55 (Japan Meteorological Agency, 2013), run on a $1^o$ tripolar grid. CICE is a state-of-the-art sea ice model and is used as the sea ice component for several of the CMIP5 models (Table 1). Below we briefly explain the model's sea ice thermodynamics; further details may be found in Hunke et al. (2015).

CICE describes the evolution of the ice thickness distribution in five discrete categories. A volume of new sea ice growth is calculated from the ocean freezing/melting potential $F_{\text{frz/mlt}}$, with new ice added as area in the smallest thickness category until the open water fraction is closed, after which it grows existing ice thickness. For sea ice melt, the net downward heat flux from the ice into the ocean, $F_{bot}$ is:

$$F_{bot} = -\rho_w c_w c_h u_* (T_w - T_f) \tag{4}$$

where $\rho_w$ and $c_w$ are the density and heat capacity of sea water, $c_h = 0.006$ is the heat transfer coefficient, $u_* = \sqrt{|\boldsymbol{\tau_w}|\rho_w}$ is the friction velocity, $T_w$ is the sea surface temperature and $T_f$ is the ocean freezing temperature, following Maykut and McPhee (1995). The balance of this flux with a conductive flux through the ice determines basal melt.

A fraction of ice is also melted laterally following Steele (1992). If floes have a mean caliper diameter $L$, their perimeter is $p = \pi L$ and their horizontal surface area is $s = \alpha L^2$ (where $\alpha \approx 0.66$ accounts for the non-circularity of floes and was determined empirically by Rothrock and Thorndike 1984). It is assumed that melting occurs uniformly at a rate $w_{lat}$ around the perimeter of each floe, i.e.

$$\frac{\mathrm{d}s}{\mathrm{d}t} = w_{lat}p$$

Therefore the change in diameter is

$$\frac{\mathrm{d}L}{\mathrm{d}t} = \frac{\pi}{2\alpha}w_{lat}$$

For a region containing $n$ floes with only a single diameter $L$, with a total horizontal area $s_{tot}$, the total concentration $A$ is

$$A = \frac{n}{s_{tot}}s(L) = \frac{n}{s_{tot}}\alpha L^2$$

Hence, with $s_{tot}$ and $n$ constant in time and letting the subscript $_o$ denote the initial state,

$$A = A_o\left(\frac{L}{L_o}\right)^2 \tag{5}$$

Differentiating this and inserting $dL/dt$ then gives the change in concentration

$$\frac{\mathrm{d}A}{\mathrm{d}t} = \frac{A\pi}{L\alpha}w_{lat} \tag{6}$$

CICE uses a uniform lateral melt rate of

$$w_{lat} = m_1(T_w - T - f)^{m_2} \tag{7}$$

which was based on Josberger and Martin (1981), who found a complex boundary layer adjacent to vertical ice walls melting in saltwater in the laboratory, with convective motions following different flow regimes. The region adjacent to the turbulent

10 flow regime showed the largest lateral melt rate, which could be fitted to the above relation. The coefficients $m_1$ and $m_2$ are the best fit to data quoted by Maykut and Perovich (1987), measured in a single static lead in the Canadian Arctic archipelago over a three week period. In order to apply Eq. 6, CICE assumes a single floe diameter of $L = 300$ m throughout the ice pack. This is one of the more sophisticated schemes for lateral melt in the CMIP5 models; often it is not included at all (Table 1).

The experiments described below, which are performed with the coupled NEMO-CICE model, begin in 1979 and end in

15 2014. The years before  1992 are neglected to allow for model spin-up. Time series of annual maximum sea ice extent show that this takes around ten years to stabilize. Model output from  the NEMO-CICE experiments is analysed on its native grid ($1^o$ tripolar). Comparisons between NEMO-CICE simulations and observations (integrated ice errors and sea ice concentration differences) are computed by interpolating observations on to the same $1^o$ tripolar grid using CDO (2015).

**3    Results**

20 ~~Despite the seemingly diverse sea ice simulations (Fig. ??), the CMIP5 models do show some similar tendencies. While sea ice area at the annual maximum has a large inter-model and inter-annual spread with no clear bias compared to observations, sea ice area at the annual minimum is consistently biased low (Fig. ??). This tendency was also noted by Turner et al. (2013) for sea ice extent. Integrated ice area error as a fraction of the observational mean sea ice area also suggests that, in general, models perform more poorly at the summer minimum than the winter maximum (Fig. ??(b)).~~

Fig. **??** shows sea ice area at the annual maximum and minimum from models and observations. Examining observations and models shown individually (Fig. **??**(a) and **??**(c)), we find that the interquartile range arising from inter-annual fluctuations over 1992-2014 is generally smaller than inter-model differences.

Fig. **??**(b) and **??**(d) group the models and observations into two populations for comparison. At the annual maximum (Fig **??**(b)), the interquartile range from the ensemble of observations for 1992-2014 is contained within the ensemble of models from the same period, with the medians of the two populations in good agreement. There is no clear model tendency compared to observations for the sea ice area maximum. At the minimum (Fig. **??**(d)), the interquartile ranges from models and observations show less overlap than the maximum, with the median from the model ensemble significantly lower than the median from the observational ensemble, suggesting a broadly consistent underestimation of sea ice area at the annual minimum by the CMIP5 models. This tendency was also noted by Turner et al. (2013) for sea ice extent. There are outliers, which show an overestimation of sea ice area, notably CSIRO-Mk-3-6-0 and the CESM models. The Kolmogorov-Smirnov test quantitatively shows that both the maximum and minimum sea ice area model-observation comparisons are significantly different, but the difference between models and observations is larger at the summer minimum than at the winter maximum (Fig.s **??**(b) and **??**(d)).

The poorer performance of models at the summer minimum is supported by the integrated ice area error (Fig. 2(a)). The integrated ice area error has a model median value of around 2 million km$^2$ at the sea ice area minimum and around 5.5 million km$^2$ at the sea ice area maximum, despite a much larger amplitude in model mean sea ice area values (around 15 million km$^2$ and 1 million km$^2$ respectively). Results are similar using the integrated ice extent error (Fig. 2(b)), although the use of extent rather than area reduces the variation between observational references. At the winter maximum, across the population of CMIP5 models and different years, we find that the absolute extent error and the misplacement extent error contribute approximately equally to the total integrated ice extent error (Fig. 2(c-d)). At the summer minimum, the integrated ice extent errors for the CMIP5 models have a slightly larger contribution from absolute extent errors than from misplacement area errors (Fig. 2(c-d)).

The large inter-model variability in extent and area at the summer minimum can be seen in Fig. 3, where the sea ice concentration fields show diverse behaviour. Variability between observational products is smaller than inter-model differences, but observational differences are visible, particularly at low concentrations. An objective way to quantify model-observation disagreement is to use the integrated ice area error, which describes the area of sea ice on which models and observations disagree. Due to observational variability, we calculate this relative to each observational product individually. The variation in observations means that we cannot rank the models in an overall order, but we can construct two groups of well-performing models and of poorly-performing models whose members do not change when using different observational products. These are marked on Fig. 3.

Such 'total' errors in the integrative measure of hemispheric sea ice area are likely caused by a temperature-biased ocean and/or atmosphere. To remove these total biases and isolate behaviour that may be induced by the sea ice component, we now consider a fractional binned sea ice extent. Fractional binned sea ice extent is a similar (but not equivalent) metric to the frequency distribution of concentration that Notz (2014) use to evaluate satellite observations in the Arctic. It can be used to describe observed and simulated behaviour in different concentration regimes. We first describe satellite observations using fractional binned sea ice extent. We now consider sea ice concentration distributions from observations and models, which provide a more detailed assessment than hemisphere-integrated measures. A normalized sea ice concentration distribution may help isolate the role of the sea ice component, as models with a constant temperature bias in the atmosphere or ocean, resulting in a biased sea ice area or extent, may still simulate the relative fraction of different concentration regimes successfully.

As shown by Ivanova et al. (2016), the CMIP5 multi-model mean and the NASA Team observations have a high fraction of ice below 10 % sea ice concentration in the summer. We find that the fraction of 0.001 - 10 %-concentration ice varies in the models from 0.005 to 1.0 (when models are essentially ice-free) in the summer (Fig. 3). It consists of up to around a third of the ice in other seasons for some models. Including these very low concentrations heavily skews the normalized sea ice concentration distribution towards low concentrations and it obscures behaviour at higher concentrations. Our aim is to look for consistent model behaviour, so to avoid the large variance between different models and between different observations at very low concentrations, we only consider sea ice concentrations above 10 %. We present all months grouped by meteorological season (December - February (DJF), March-May (MAM), June - July (JJA) and September - November (SON)). This choice separates the melt season (September-February) from the freezing season (March - August), while limiting the number of months included in each season.

In the Antarctic, the ASI observations show similar characteristics to the Bootstrap observations, while the NASA Team observations differ from both. This results in a skewed distribution when considering the observational range from the three data sets relative to the observational mean. We find that the NASA Team algorithm shows a looser ice cover, with a significantly lower proportion of cover in the 90 %+ concentration bin, than both the Bootstrap and ASI observations (Fig. **??**). This result holds when considering (non-fractional) binned sea ice extent as well (not shown). We also find that this difference between data sets persists throughout the year, unlike in the Arctic where frequency of compact sea ice cover shown in the Bootstrap and NASA Team datasets agree better in winter (Notz, 2014). This suggests that it is not just treatment of melt ponds, as suggested by Notz (2014) and which are less important in the Antarctic than Arctic, or of ice types associated with melting that differs between the two algorithms. Note that the large range in Fig. **??** reflects both inter-annual and sub-seasonal variability in the observations, as well as uncertainty arising from different processing of satellite data. We first describe satellite observations using the normalized sea ice concentration distribution (Fig. 4). Here, individual boxplots contain both inter-annual and sub-seasonal variability, while the differences between boxplots reflects uncertainty arising from different processing of satellite data. Differences between observational products are largest for compact ice (90 %+) than other concentrations. In general, the ASI-SSMI observations show more similar characteristics to the Bootstrap observations than the NASA Team observations for most of the year, apart from DJF where the opposite is true. This results in a somewhat skewed distribution when considering an ensemble created from three data sets. We find that the NASA Team algorithm shows a looser

ice cover, with a significantly lower proportion of cover in the 90 %+ concentration bin, than both the Bootstrap and ASI-SSMI observations. This result holds when considering an un-normalized sea ice concentration distribution as well (not shown). The fraction in the 70-90 % bins is larger to compensate. We also find that differences between data sets persists throughout the year. This is in contrast to the Arctic, where the frequency of compact sea ice cover shown in the Bootstrap and NASA Team datasets shows largest disagreement in the summer, due to issues with treatment of melt ponds (Notz, 2014). In the Antarctic, observational uncertainty in the frequency of compact sea ice is largest in winter.

We find that the lower to upper quartile ranges for fractional binned sea ice extent from the population of all observations and the population of all models, including inter-annual and sub-seasonal information, broadly agree (Fig. **??**(a-d)). While most differences between models and observations are significant at the 95 % confidence level (as expected given the population size), the two populations' lower to upper quartile ranges are not distinct for most concentration bins. This indicates observational limitations as well as suggesting that the sea ice components of CMIP5 models are somewhat successful at simulating the distribution of sea ice concentration.

However, there are some significant differences from observations. To better highlight these, we show the percentage difference from the observational mean in each bin (Fig. **??**(e-h)). The populations are the percentage differences from the observational mean for each day and year, and the observational distribution is skewed, so the population from observations is not centred on zero. Fig. **??**(e-h) reveals large discrepancies between models and observations in the highest and especially the lowest concentration bins. During summer (DJF), the lower to upper quartile range for 90 %+ sea ice concentration from models and observations do not overlap at all. Models strongly underestimate the fraction of sea ice area with concentration greater than 90 %, that is, their central ice pack is not compact enough. They tend to overestimate the fraction in the 70-90 % bins to compensate. In all seasons, the models overestimate the fraction of sea ice area in the two lowest concentration bins. The upper to lower quartiles from observations and models do not overlap at all for the 10-20 % bin in spring, autumn and winter. These results for the highest and lowest bins are robust when considering sea ice concentration bins spaced at 5 % intervals and beginning at 15 %, the cut-off used universally for sea ice extent.

Differences between the sea ice concentration distribution from models and observations, including inter-annual and sub-seasonal information, (Fig. 5) are less distinct than between observational products themselves. This reflects the large range in both models and observations due to systematic uncertainties. The overall decomposition from the CMIP5 models, with a large fraction of compact ice cover and smaller fractions of lower concentrations is somewhat in agreement with observations. Agreement appears poorest in DJF, where the lower to upper quartile range for 90 %+ sea ice concentration from models and observations overlap very little. Models strongly underestimate the fraction of sea ice area with concentration greater than 90 %, that is, their central ice pack is not compact enough. They tend to overestimate the fraction in the 80-90 % bin and at lower concentrations to compensate. In other seasons, there appears to be a slight tendency to overestimate the fraction of compact ( 90 %+) ice, with a reduction in the 70-90 % bins to compensate. The two-sample Kolmogorov-Smirnov test can be used to quantify the degree of disagreement between models and observations. The confidence level that the ensemble of observations and ensemble of models were drawn from the same population has the smallest values for the 90 - 100 % and 10-20 % in DJF, the 70 - 90 % concentrations in MAM, the 10-30 % concentrations in JJA and the 80-90 % and 10-20 % concentrations in

SON. There is a tendency for models to overestimate the fraction of low-concentration (10 - 20 %) sea ice in all seasons. This overestimation of <20% sea ice compared to observations is robust when considering sea ice concentration bins spaced at 5 % intervals and beginning at 15 %, the cut-off used universally for sea ice extent (not shown). Unlike the other CMIP5 model tendencies, the overestimation of 10-20% ice occurs in every month (Fig. 6), with the CMIP5 model median always outside the interquartile range of the observations.

As discussed above, this assessment takes into account observational uncertainty and inter-annual and sub-seasonal variability. That  distinct tendencies arise from a population of  40 models, which contain diverse physics and different sea ice, ocean and atmosphere models, is striking. It suggests that there is some deficiency or missing physical process common to  many models.

A plausible explanation could be that models form sea ice that is too thin in the highest bin, which therefore melts more easily. Conversely, low-concentration sea ice may be too thick. However, we found no relation between these concentration biases and average sea ice thickness for the lowest and highest concentration bins (not shown). We therefore turn to lateral, rather than vertical, thermodynamics in the next section.

**4 Impact of floe size**

We hypothesize that the biases in low-concentration Antarctic sea ice are partially  influenced by lateral floe size. Lateral floe size impacts sea ice concentration through lateral melt only if included at all in the CMIP5 models  (see Table 1). Separating models with and without an explicit lateral melt term (Table 1), we find a significant difference between the two groups. Models with explicit lateral melt show a greatly reduced fraction of low-concentration ice in from March to July compared to models without, in good agreement with the observations (Fig. 8). Lateral melt can occur all year at the ice edge, where low concentrations occur.

Fig. 8 demonstrates that lateral melt significantly impacts the normalized sea ice concentration distribution during autumn. However, lateral melt as it is currently included in CMIP5 models still results in a tendency towards overestimation of low-concentration sea ice in other months, and some models with an explicit lateral melt term (including the ocean — sea ice model NEMO-CICE) still simulate too large a fraction of loose ice.

We therefore proceed by examining whether changes to the lateral melt scheme may also impact the simulation of sea ice. The current representation of lateral melt in CMIP5 models is heavily parametrized (Table 1), with the formulation described in Subsect. 2.4 being the most complex parametrization available in the CMIP5 models. Tsamados et al. (2015) showed that a more advanced concentration-dependent lateral melt parametrization significantly impacted the decomposition of sea ice melt processes, resulting in reduced sea ice concentrations around the ice edge in the Arctic. In the Antarctic, heat flux from solar heating of open water areas has been cited as the major cause of sea ice decay (Nihashi and Ohshima, 2001), with this melting potential available for both lateral and bottom melt. Recent studies have also suggested that floe size should also impact sea ice concentration through processes such as floe-floe collisions and lateral growth (Horvat and Tziperman, 2015; Zhang et al., 2015).

As shown in Subsect. 2.4, in CICE the lateral melt flux is independent of floe size, while the change in concentration arising from lateral melt is inversely proportional to a constant floe diameter, $D = 300$ m. In reality, sea ice floes can range in size across orders of magnitude. Several observational studies (e.g. Steer et al. 2008; Paget et al. 2001) find that the number distribution of floe sizes per unit area follows a power law with a negative exponent, suggesting that there can be a large number of small floes.

While concentration is not a proxy for floe size, in general we may expect that low-concentration areas will be made up of smaller sea ice floes than high-concentration areas because they are usually nearer the ice edge. An area of smaller sea ice floes will experience more lateral melt than an area with a larger floe size (Eq. 6). We therefore suggest that CMIP5 models using the Steele (1992) lateral melt parametrization simulate too much low-concentration sea ice because this is made up of floes smaller than 300 m and so should be subject to more lateral melt.  In areas around the ice edge, which are principally low-concentration, marginal ice zone processes not included in CMIP5 models, such as wave fracture and dynamic floe interactions, may further reduce concentrations. Conversely, in high-concentration areas, floes are likely to be larger than 300 m and therefore should be subject to less lateral melt than the Steele (1992) parametrization prescribes. This could explain the underestimation of high-concentration sea ice seen in Antarctic summer.

In order to test this hypothesis, we perform three experiments using the coupled ocean — sea ice model (NEMO-CICE) described in Subsect. 2.4. The experiments have identical set ups apart from a variation in $L$, the fixed floe diameter. We run experiments using (i) the standard value of $L = 300$ m, (ii) a low value of $L = 1$ m and (iii) a high value of $L = 10,000$ m. Our perturbed parameter values are constant and not realistic, but instead are chosen to investigate and highlight the impact of extreme changes.

~~Fig. 7 shows the impact of reduced floe size on fractional binned sea ice extent. The fraction of sea ice extent belonging to each concentration bin is significantly different (according to a Student's T-test with $p < 5$ % and with little overlap between the upper and lower quartile ranges) to that from the standard model during DJF (Fig. 7). There is a large reduction in low-concentration (10-20 %) sea ice compared to the standard model, with increases in the 30-70 % concentration bins to compensate. This brings the simulated fractional binned sea ice extent into better agreement with observations. The reduction in low-concentration sea ice is also visible during the fall, MAM (Fig. **??**). Note that we have tested only the impact of lateral melt on this bias. Other floe-size dependent processes may reduce the proportion of low-concentration sea ice further. However, we cannot test this without access to sea ice models that include these processes.~~ Fig. 9 shows the fraction of 10-20% sea ice concentration from observations, the standard NEMO-CICE model and the model with reduced floe size. The standard model has very strong overestimation of low-concentration ice through December to March compared to observations. Impact of reduced floe size on the distribution is limited, with the exception of February where there is a very strong reduction in the fraction of 10-20 % concentration ice, bringing it into better agreement with observations.

The enhanced lateral melt achieved by reducing floe size results in statistically significant reductions in sea ice concentration relative to the standard model  in DJF (Fig. 10(b)). December, January and February stand out from the other months in having particularly high total lateral melt rates. As expected from Fig. 6, enhanced lateral melt reduces

the high bias in concentration near the outer ice edge compared to  Bootstrap observations in DJF (reduction in blue, Fig. 10(d-e)), but enhances the low bias compared to the Bootstrap observations elsewhere (increase in red, Fig. 10(d-e)). We use the integrated ice  extent error described above to quantify agreement with the  Bootstrap observations. The same qualitative picture is obtained from all three observational products. We find that the difference in overall agreement with observations between the standard model and the small floe simulation is negligible. The absolute  extent error significantly increases in the small floe simulation, because overall this simulation melts too much ice compared to observations. The misplacement  extent error, however, is significantly reduced in the small floe simulation. This is partly because there is less ice to be misplaced, but also because increased lateral melt improves the distribution of sea ice around the ice edge, by melting areas where there is too much ice compared to observations (Fig. 10(d-e)).

Besides lateral melt, a number of other physical processes, including dynamical ones, may also contribute to an overestimation of low-concentration ice. Lecomte et al. (2016) find systematic wind-driven biases in sea ice drift speed and direction at the exterior of the Antarctic ice pack. Errors in surface winds could contribute to poor simulation of low-concentration sea ice. However, we find a very strong over-estimation in low-concentration sea ice in the NEMO-CICE model, which is forced by a reanalysis atmosphere and so should not have very unrealistic winds. The dynamical response of sea ice to winds at the edge of the ice may be poorly represented, as we would expect sea ice dynamics to be floe-size dependent. Alternative rheologies (such as a granular rheology, eg. Feltham 2005) may be better suited to this domain. Concentrations could also be reduced by mechanical interactions between floes. However, we cannot test the impact of such floe-size dependent processes without access to sea ice models that include them.

The impact of increased floe size, on the other hand, is much smaller (Fig. 10(c,f)). Differences in sea ice concentration between the standard model and the large floe simulation are barely perceptible. Changes in the ice errors relative to the standard model are of the opposite sign compared to the small floe simulation, but these changes are unlikely to be significant. Examining the basal and lateral melt rates, we find that the hemispheric average DJF  1992-2014 mean lateral melt rate accounts for only 5 % of the combined basal and lateral melt rates in the standard model. It accounts for a larger proportion ( 17 %) of melt in the Arctic. Decreasing floe diameter by two orders of magnitude increases the lateral melt rate to 83 % of the combined basal and lateral melt. This compensation effect of reduced basal melt when lateral melt is increased was also noted by Tsamados et al. (2015) in the Arctic. On the other hand, increasing the floe diameter by two orders of magnitude effectively switches off lateral melt (0.2 % of combined basal and lateral melt). In the latter case, more melting potential is made available for basal melting, which, because Antarctic sea ice is so thin, has the same impact on sea ice concentration as lateral melt. We conclude that there must be alternative reasons for the consistent underestimation of compact summer ice.

Looking at the regional distribution of  DJF (the season where the bias is apparent in Fig. 5) seasonal mean sea ice concentration averaged over 1992-2014, high-concentration (90-100 %) ice appears in the  observations mean only in the Weddell Sea (Fig. 3). Taking the difference between the  high-concentration ice  in each observational product and the sea ice concentration in the CMIP5 model simulations shows that  very few of the models simulate high enough concentrations in this area

. Fig. 11 shows the difference between the ASI-SSMI observations and the CMIP5 models; differences are slightly enhanced using Bootstrap and less pronounced when using NASA Team. This demonstrates a consistent model tendency to underestimate concentrations in the Weddell Sea, the largest region of multi-year ice in Antarctica. The bias is not present in other seasons, suggesting it is related to overestimated melt or break-up processes, including misrepresentation of sea ice dynamics.

5      Overestimated melt or break-up could be a result of the sea ice model or a biased warm atmosphere or ocean. While consideration of  normalized sea ice concentration distribution is intended to remove overall biases caused by (for example) a warm ocean, in summer the warm ocean could shift the whole distribution to lower concentrations. Alternatively, or likely in conjunction with this, regionally important processes may be being misrepresented. Evaluating the ORCA2-LIM coupled ocean-sea ice model, Timmermann et al. (2004) found that overestimation of westerly winds led to an

10   underestimation of sea ice coverage on the eastern side of the Antarctic peninsula, in the Weddell Sea.  Other CMIP5  models may simulate high drift speeds due to winds or sea ice rheology, which Lecomte et al. (2016) found correlated with a faster sea ice retreat.

**5   Discussion**

In this study, we examine the distribution of sea ice concentration from both models and observations. Firstly, we show that

15   observed sea ice concentration values can differ significantly between three widely-used algorithms for satellite data. This observational uncertainty provides a limit beyond which we cannot further evaluate model agreement with observations. Many sea ice model-observations comparisons use only one satellite dataset assumed to represent the true observed state, an approach which may be sufficient when using sea ice extent, a metric where the various algorithms broadly agree. However, when using metrics that go beyond sea ice extent, using for example sea ice area or sea ice concentration distributions, model evaluation

20   studies should account for the observational range.

We find that simulation of high-concentration (90 %+) sea ice in models is in better agreement with the NASA Team observations than the observational range including the Bootstrap and  ASI-SSMI observations, in agreement with Ivanova et al. (2016), who only examined the CMIP5 multi-model mean.

25

Accounting for the  range in three observational products, we find that models overestimate the extent of low-concentration sea ice throughout the year, while underestimating the extent of high-concentration sea ice in summer.

30   This common behaviour across diverse models with varying physics is a result not previously highlighted and warrants further attention.

We note that using the observational range as an uncertainty estimate neglects biases that are common to the three different satellite observations. As mentioned above,

 sea ice concentrations are considered to be most uncertain during melt conditions, for large fractions of thin ice and at low concentrations during storms. In the context of the results from the model-observation comparison for normalized sea ice concentration distributions, we suggest that the impact of uncertainty of melt conditions is limited as the high bias in low-concentration ice from CMIP5 models is visible throughout the year. The low bias in high-concentration ice during the melt season would be strengthened if observations were underestimating ice concentrations in this season. Inclusion of both NASA Team and Bootstrap algorithms, with the former tending to cancel out physical temperature effects, will sample some of this uncertainty.

The underestimation of sea ice concentrations in areas of thin ice ($< 35$ cm) (Ivanova et al., 2015) may cause a bias at any concentration in the observed normalized sea ice concentration distribution from observations, with the possibility of a positive bias in the very lowest concentrations. Stormy conditions near the ice edge lead to false sea ice concentrations near the ice edge; weather filters may accurately remove these, leave them uncorrected (Andersen et al., 2006) or erroneously remove real sea ice. The latter may underestimate low concentrations. Spreen et al. (2008) suggest the filter method used in ASI-SSMI observations may result in a positive bias in the marginal ice zone, and Steffen and Schweiger (1991) found that the NASA Team algorithm overestimates low-concentration ice when compared to Landsat imagery. Considering all this evidence we suggest that the magnitude or sign of any systematic biases in satellite radiometer observations is unclear when comparing with climate models. This is particularly true for low concentrations. Here the use of different approaches to weather filters within the different algorithms may assist in sampling observational uncertainty. Development of sea ice satellite emulators, which use climate model output to calculate brightness temperatures (eg. Tonboe et al. 2011), may help to reduce uncertainty when comparing models to observations in the future.

Categorising models according to whether they explicitly represent lateral melting, which is the only thermodynamic sea ice process that reduces concentrations in models regardless of sea ice thickness, we find a strong impact of this process on low-concentration sea ice. In Subsect. 2.4 we briefly review typical sea ice model thermodynamics, and in particular the change in concentration induced by lateral melt rate for a region containing floes of a single diameter, which follows Steele (1992). Horvat et al. (2016) finds that development of ocean eddies due to lateral density gradients could induce much larger lateral melt than that suggested from the Steele (1992) geometric model. This would support increasing the lateral melt rate in models, as we have done artificially here through a reduced constant floe size. Heat budget analysis (Nihashi and Ohshima, 2001) and modelling studies (Fichefet and Maqueda, 1997; Ohshima and Nihashi, 2005) suggest that the major cause of Antarctic sea ice decay is atmospheric heat input to open water, which causes bottom and lateral melt. Fichefet and Maqueda (1997) find that sea ice melt by open water plays a larger role in the Antarctic than in the Arctic. We further note that the coefficients in the lateral melt rate used in CICE were measured in the Arctic only (Maykut and Perovich, 1987) and few, if any, observational studies exist on the relative importance of bottom and lateral melt in the Antarctic.

The impacts of enhancing lateral melt via reducing a constant floe size shown here suggest that this process should not be applied in the same way throughout the ice pack. While not all models include such a lateral melt parametrization, the biases at the tails of the concentration distributions from the CMIP5 models point to inclusion of model processes that are not suitable for both high-concentration and low-concentration regimes. A possible conclusion, therefore, is that physics in sea ice models

5   are not heterogeneous enough to represent observed sea ice cover. Given the possible contribution of dynamic processes to model biases in the sea ice concentration distribution, a full exploration of sea ice dynamics for all CMIP5 models using the sea ice concentration budget decomposition of Uotila et al. (2014) would be welcome. Including information on the floe size distribution and floe size dependent processes (e.g. Horvat and Tziperman 2015; Zhang et al. 2016; Bennetts et al. 2017) could improve consistency with observations in the metrics presented here.

*Data availability.* Most data from this study are publicly available. See http://cmip-pcmdi.llnl.gov/cmip5/data_portal.html for CMIP5 data, http://icdc.cen.uni-hamburg.de/1/daten/cryosphere/seaiceconcentration-asi-ssmi.html for ASI-SSMI data and https://nsidc.org/data/docs/noaa/ g02202_ice_conc_cdr/ for the Bootstrap and NASA Team data. NEMO-CICE model output is available from the corresponding author upon request.

5    *Author contributions.* L. Roach and S. Dean designed the analysis and experiments and L. Roach carried them out. L. Roach prepared the manuscript with contributions from all co-authors

*Competing interests.* The authors declare that they have no conflict of interest.

*Acknowledgements.* The authors wish to thank Erik Behrens and Jonny Williams for assistance setting up and running the coupled ocean — sea ice model,  Stephen Stuart for assistance with CMIP5 output acquisition and Cecilia Bitz for helpful discussions in the early
10   stages of this work. We are grateful to the reviewers, whose comments significantly improved the manuscript. This research was funded via Marsden contract VUW-1408.

[revised manuscript text omitted]

**Figure 1.** ~~(a) The maximum of the sea ice area seasonal cycle and (b) the minimum of the sea ice area seasonal cycle for the population of all observations and the population of models for each year from 2000 to 2014. Boxes extend from the lower to upper quartile values of the data with a line at the median. Whiskers show 1.5 of the interquartile range; beyond this data are considered outliers and plotted as individual points~~ Sea ice area for the months where the maximum (a-b) and minimum (c-d) of the seasonal cycle occur. Populations include data from all years from 1992 to 2014 with boxplots for (a,c) the three observational products (ASI-SSMI, Bootstrap and NASA Team) and all CMIP5 models listed in Table 1 individually; and (b,d) for the ensemble of observational products and the CMIP5 model ensemble. Boxes extend from the lower to upper quartile values of the data with a line at the median. Whiskers show 1.5 of the interquartile range; beyond this data are considered outliers and plotted as individual points. The text labels in (b,d) is the p-value calculated from a Kolmogorov-Smirnov test, which represents the confidence that the two populations come from the same distribution

[Figure]

**Figure 2.** ~~Various ice errors for the population of CMIP5 models for all years from 1992 to 2014. Errors are shown relative to (red) the ASI-SSMI satellite observations, (grey) the Bootstrap satellite observations and (light blue) the NASA-Team observations for the months where the maximum and minimum of the seasonal cycle occur of sea ice area (a) or of sea ice extent (b-d) occur. The errors shown are the integrated ice area error (a), the integrated ice extent error (b), the absolute extent error divided by the integrated extent error (c) and the misplacement extent error divided by the integrated extent error (d). Boxplots are as in Fig. ??~~ (a) The integrated ice area error (IIAE), absolute area error (AAE) and misplacement area error (MAE) relative to the observational mean from the population of CMIP5 models at the day of maximum and minimum sea ice area for each year from 2000 to 2014; (b) as (a) but shown as a fraction of the observational mean sea ice area; (c) as (a) but showing absolute area error and misplacement area error as a fraction of the integrated ice area error for each model and year. Boxplots as in Fig. 2

[Figure]

**Figure 3.**  Sea ice concentrations (above 0.1 %) for the three sets of observations (a-c) and the CMIP5 models (d-ar) for the month of each model or observation's sea ice area minimum, averaged over 1992-2014. Models marked with a bold (dashed) bounding box have high-ranked (low-ranked) integrated ice area errors regardless of observational product used. Integrated ice area errors consider sea ice concentrations > 15 % for the sea ice field shown

[Figure]

**Figure 4.** The normalized sea ice concentration distribution for all months in each year from 1992 to 2014 in (a) DJF, (b) MAM, (c) JJA, and (d) SON from the three sets of satellite observations. Boxplots as in Fig. **??**

[Figure]

**Figure 5.** ~~(a-d): the fraction of sea ice extent in concentration bins for all days in each year from 2000 to 2014 in (a) DJF, (b) MAM, (c) JJA, and (d) SON for the three sets of satellite observations (*blue*) and the 27 CMIP5 models (*green*). (e-h): as (a-d), but shown as a percentage difference from the observational mean for each day in (e) DJF, (f) MAM, (g) JJA, and (h) SON, for each year from 2000 to 2014. Boxplots as in Fig. 2. Hatching on the model populations denotes that they are statistically different from the corresponding population of observations according to a Student's t-test ($p < 5\,\%$)~~The normalized sea ice concentration distribution for all months in each year from 1992 to 2014 in (a) DJF, (b) MAM, (c) JJA, and (d) SON from the the three sets of satellite observations (*blue*) and the 40 CMIP5 models (*green*). Boxplots as in Fig. **??**. Annotated text is the p-value calculated from a Kolmogorov-Smirnov test, which represents the confidence that the two populations come from the same distribution

[Figure]

**Figure 6.** The 10-20% bin from the normalized sea ice concentration distribution for each month, where boxes contain all years from 1992 to 2014 from (*blue*) the the three sets of satellite observations and (*green*) the 40 CMIP5 models. Boxplots as in Fig. **??**. Annotated text is the p-value calculated from a Kolmogorov-Smirnov test, which represents the confidence that the two populations come from the same distribution

[Figure]

**Figure 7.** ~~The fraction of sea ice extent in concentration bins for all days in each year from 2000 to 2014 in DJF from the three sets of observations, the standard NEMO-CICE ice-ocean model, and the NEMO-CICE ice-ocean model with small floe diameter $L = 1$ m. Boxplots as in Fig. 2. Hatching on the small floe simulation populations denotes that they are statistically different from the corresponding population from the standard model according to a Student's t-test ($p < 5$ %)~~

[Figure]

**Figure 8.** The 10-20% bin from the normalized sea ice concentration distribution for each month, where boxes contain all years from 1992 to 2014 from (*blue*) the the three sets of satellite observations, (*purple*) CMIP5 models that include an explicit lateral melt term and (*grey*) CMIP5 models that do not (from Table 1). Boxplots as in Fig. **??**. Annotated text is the p-value calculated from a Kolmogorov-Smirnov test, which represents the confidence that the two populations come from the same distribution

[Figure]

**Figure 9.** ~~The fraction of sea ice extent in the lowest concentration bin ((10-20 %)) for all days in each season for each year from 2000 to 2014 from the three sets of observations, the standard NEMO-CICE ice-ocean model, and the NEMO-CICE ice-ocean model with small floe diameter $L = 1$ m. Boxplots as in Fig. 2. Hatching on the small floe simulation populations denotes that they are statistically different from the corresponding population from the standard model according to a Student's t-test ($p <$5 %)~~The 10-20% bin from the normalized sea ice concentration distribution for each month, where boxes contain all years from 1992 to 2014 from a NEMO-CICE simulation with (*blue*) the the three sets of satellite observations, (*light blue*) a floe diameter of 300 m (the standard model) and (*orange*) a floe diameter of 1 m. Boxplots as in Fig. **??**. Annotated text is the p-value calculated from a Kolmogorov-Smirnov test, which represents the confidence that the two populations come from the same distribution

[Figure]

**Figure 10.** ~~Simulation minus observational mean sea ice concentration for DJF 2000-2014 where the simulation has (a) a floe diameter of 300 m (the standard model), (b) a floe diameter of 1 m (small floes), and (c) a floe diameter 10,000 m (large floes). Labels show the integrated ice area error, absolute area error and misplacement area error in million km$^2$. Differences are shown only if they are statistically different according to a Student's t-test over 2000-2014 ($p$ <5 %)~~ Sea ice concentration averaged over DJF 1992-2014 for (a) the standard model simulation with a floe diameter of 300 m; (b) a model simulation with a floe diameter of 1 m (small floes) minus (a); (c) a model simulation with a floe diameter of 10,000 m (large floes) minus (a). (d-f) show simulation minus observed Bootstrap sea ice concentration, where the latter has been interpolated on to the model grid for (d) the standard model simulation, (e) the small floes simulation and (d) the large floes simulation. In (b-f), differences are shown only if they are statistically different according to a Student's t-test over 1992-2014 ($p$ <5 %). Labels on (d-f) show the integrated ice extent error, absolute extent error and misplacement extent error in million km$^2$.

**Figure 11.** Simulation minus observed ASI-SSMI sea ice concentration for DJF 1992-2014 for each CMIP5 model, where only grid cells with observational mean sea ice concentration is $\geq$ 90 % are considered. Differences are only shown if they are statistically different according to a Student's t-test over 1992-2014 ($p <$5 %)

**Table 1.** CMIP5 models used in this study. SIC denotes sea ice concentration

| Short name | Country |  Resolution | Sea ice model |  Explicit lateral melt term |
|---|---|---|---|---|
| ACCESS1-0 | Australia | $1^o \times 1^o$ tripolar | CICE4.1 | As Subsect. 2.4 |
| ACCESS1-3 | Australia | $1^o \times 1^o$ tripolar | CICE4.1 | As Subsect. 2.4 |
| bcc-csm1-1 | China | $1^o \times (1-\frac{1}{3})^o$ tripolar | SIS | Not included (Li, 2014) |
| bcc-csm1-1-m | China | $1^o \times (1-\frac{1}{3})^o$ tripolar | SIS | Not included (Li, 2014) |
| CanCM4 | Canada | $1.875^o \times 1.875^o$ T63 Gaussian | CanSIM1 | Unknown (reference N/A) |
| CanESM2 | Canada | $1.875^o \times 1.875^o$ T63 Gaussian *– | CanSIM1 | Unknown (reference N/A) |
| CCSM4 | USA | $1.11^o \times (0.27-0.54)^o$ dipolar | CICE4 | As Subsect. 2.4 |
| CESM1-BGC | USA | $1.11^o \times (0.27-0.54)^o$ dipolar | CICE4 | As Subsect. 2.4 |
| CESM1-CAM5 | USA | $1.11^o \times (0.27-0.54)^o$ dipolar | CICE4 | As Subsect. 2.4 |
| CMCC-CM | Italy | ORCA-$2^o$ tripolar | LIM2 | Not included (Rousset et al., 2015) |
| CMCC-CMS | Italy | ORCA-$2^o$ tripolar | LIM2 | Not included (Rousset et al., 2015) |
| CNRM-CM5 | France | ORCA-$1^o$ tripolar | GELATO5 | Thickness-dependent parametrization (Salas Melia, 2002 |
| CSIRO-Mk3-6-0 | Australia | $1.875^o \times 0.94^o$ T63 Gaussian * | in-house | Included, but unclear how it impacts SIC **(O'Farrell, 1 |
| EC-EARTH | EU | ORCA-$1^o$ tripolar | LIM2 | Not included (Rousset et al., 2015) |
| FGOALS-g2 | China | $(1-\frac{1}{2}) \times (1-\frac{1}{2})^o$ tripolar | CSIM5 | As Subsect. 2.4 (Briegleb et al., 2004) |
| GFDL-CM2p1 | USA | $1^o \times 1^o$ tripolar | SIS | Not included (Winton, 2001) |
| GFDL-CM3 | USA | $1^o \times 1^o$ tripolar | SIS | Not included (Winton, 2001) |
| GFDL-ESM2G | USA | $1^o \times 1^o$ tripolar | SIS | Not included (Winton, 2001) |
| GFDL-ESM2M | USA | $1^o \times 1^o$ tripolar | SIS | Not included (Winton, 2001) |
| GISS-E2-H | USA | $1^o \times 1^o$ tripolar | in-house | Not included (Russell et al., 1995) |
| GISS-E2-H-CC | USA | $1^o \times 1^o$ tripolar | in-house | Not included (Russell et al., 1995) |
| GISS-E2-R | USA | $1^o \times 1.25^o$ | in-house | Not included (Russell et al., 1995) |
| GISS-E2-R-CC | USA | $1^o \times 1.25^o$ | in-house | Not included (Russell et al., 1995) |
| HadCM3 | UK | $1.25^o \times 1.25^o$ | in-house | Not included (Gordon et al., 2000) |
| HadGEM2-AO | South Korea | $1^o \times 1^o$ | CICE-like | Parametrization for SIC $< 5$ % (McLaren et al., 2006) |
| HadGEM2-CC | UK | $(1-\frac{1}{3})^o \times 1^o$ | CICE-like | Parametrization for SIC $< 5$ % (McLaren et al., 2006) |
| HadGEM2-ES | UK | $(1-\frac{1}{3})^o \times 1^o$ | CICE-like | Parametrization for SIC $< 5$ % (McLaren et al., 2006) |
| inmcm4 | Russia | $1^o \times \frac{1}{2}^o$ | in-house | Empirical parametrization  (Yak |
| IPSL-CM5A-LR | France | ORCA-$2^o$ tripolar | LIM2 | Not included (Rousset et al., 2015) |
| IPSL-CM5A-MR | France | ORCA-$2^o$ tripolar | LIM2 | Not included (Rousset et al., 2015) |
| IPSL-CM5B-LR | France | ORCA-$2^o$ tripolar | LIM2 | Not included (Rousset et al., 2015) |
| MIROC4h | Japan | $0.28^o \times 0.19^o$ | in-house | Not included (Komuro et al., 2012) |
| MIROC5 | Japan | $1.4^o \times (0.5-1.4)^o$ | in-house | Not included (Komuro et al., 2012) |
| MIROC-ESM | Japan | $1.4^o \times 1^o$ | in-house | Not included (Komuro et al., 2012) |
| MIROC-ESM-CHEM | Japan | $1.4^o \times 1^o$ | in-house | Not included (Komuro et al., 2012) |
| MPI-ESM-LR | Germany | $1.5^o \times 1.5^o$ | in-house | Not included (Notz et al., 2013) |
| MPI-ESM-MR | Germany | $0.4^o \times 0.4^o$ | in-house | Not included (Notz et al., 2013) |
| MRI-CGCM3 | Japan | $1^o \times 0.5^o$ tripolar | in-house | Not included (Tsujino et al., 2010) |
| NorESM1-M | Norway | $1.11^o \times (0.25-0.54)^o$ | CICE4.1 | As Subsect. 2.4 |
| NorESM1-ME | Norway | $1.11^o \times (0.25-0.54)^o$ | CICE4.1 | As Subsect. 2.4 |